# Fabrication of Novel Polymer Composites from Leather Waste Fibers and Recycled Poly(Ethylene-Vinyl-Acetate) for Value-Added Products

Shubham Sharma [1,2,3], P. Sudhakara [1], Jujhar Singh [2], Sanjay M. R. [4,*] and S. Siengchin [4]

[1] CSIR—Central Leather Research Institute, Regional Centre, Jalandhar 144021, Punjab, India
[2] Department of Mechanical Engineering, IK Gujral Punjab Technical University, Jalandhar-Kapurthala Road, Kapurthala 144603, Punjab, India
[3] Mechanical Engineering Department, University Centre for Research and Development, Chandigarh University, Mohali 140413, Punjab, India
[4] Natural Composites Research Group Lab., Department of Materials and Production Engineering, The Sirindhorn International Thai-German Graduate School of Engineering (TGGS), King Mongkut's University of Technology North Bangkok (KMUTNB), Bangkok 10800, Thailand
* Correspondence: mavinkere.r.s@op.kmutnb.ac.th

**Abstract:** This investigation was focused on evaluating the utilization of Leather-waste, i.e., "Leather Shavings", to develop "Poly(ethylene-vinyl-acetate)" (EVA) based "polymer matrix composites". Composites with the highest ratio of 1:1 were developed using a rolling-mill, which was then subjected to hot-press molding for value-added applications, notably in the "floor-covering", "structural", "footwear", and "transportation domain". The specimens were examined for evaluating the "physico-mechanical characteristics" such as, "Compressive and Tensile, strength, Abrasion-resistance, Density, tear-resistance, hardness, adhesion-strength, compression, and resilience, damping, and water absorption" as per standard advanced testing techniques. Raising the leather-fiber fraction in the composites culminated in considerable enhancement in "physico-mechanical characteristics" including "modulus", and a decline in "tensile-strain" at "fracture-breakage". The thermo-analytic methods, viz. TGA and DSC studies have evidenced that substantial enhancement of thermo-stability (up to 211.1–213.81 °C) has been observed in the newly developed PMCs. Additionally, the DSC study showed that solid leather fibers lose water at an endothermic transition temperature of around 100 °C, are thermo-stable at around 211 degrees centigrade, and begin to degrade at 332.56-degree centigrade for neat recycled EVA samples and begin to degrade collagen at 318.47-degree centigrade for "leather shavings/recycled EVA polymer composite samples", respectively. Additionally, the "glass transition temperature" ($T_g$) of the manufactured composites was determined to be between −16 and 30 °C. Furthermore, SEM and EDAX analysis have been used to investigate the morphological characteristics of the developed composites. Micrograph outcomes have confirmed the excellent "uniformity, compatibility, stability and better-bonding" of leather-fibers within the base matrix. Additionally, the "Attenuated-total-reflection" (ATR-FTIR) was carried out to test the "physicochemical chemical-bonding", "molecular-structure", and "functional-groups" of the "base matrix", and its "composites" further affirm the "recycled EVA matrix" contained additives remain within the polymeric-matrix. An "X-ray diffraction study" was also conducted to identify the "chemical-constituents" or "phases" involved throughout the "crystal-structures" of the base matrix and PMCs. Additionally, AFM analysis has also been utilized to explore the "interfacial adhesion properties" of mechanically tested specimens of fabricated polymeric composite surfaces, their "surface topography mapping", and "phase-imaging analysis" of polymer composites that have leather-shavings fibers.

**Keywords:** leather shavings; recycled EVA; compression molding; rolling; footwear and leather ancillaries(s) applications

## 1. Introduction

A crucial worldwide commodity is leather. With an estimated USD 200 billion in annual global trade value for leather and leather goods, the leather industry is a significant contributor to the economy. However, the wastes ('solid and liquid') of such industries, such as buffing dust sludge etc., regularly pose a threat to the industry [1]. Land disposal is the most common method of treating solid waste [1,2]. Numerous waste-management options work to create value from trash by creating new materials and technology [3–5]. The fibrous structure of the leather is one of its key characteristics. It also offers excellent characteristics such as "high tear resistance, flexibility, thermal insulation, water resistance, and form maintenance". Therefore, adding this leather waste (specifically leather shavings) to any appropriate polymer matrix may result in composites with improved qualities for a variety of applications, including 'footwear, leather ancillaries, automotive, transportation, packaging, etc. Making valuable goods out of unused material would have a synergistic influence on the atmosphere and add value to the wastes produced by the polymer and leather industries, respectively. By using environmentally friendly polymers as matrix materials, numerous researchers have attempted to use leather wastage to make bio-composite materials, including 'films, sheets, fibers, etc.' [6–25].

The development of footwear and clothing made of thermoplastic elastomeric (TPE) material is rapidly accelerating. Numerous factors and criteria have been made possible by TPEs in order to meet consumer expectations [26–29]. EVA is used to make 'inserts/soles and midsoles' throughout the industrial footwear sector, accounting for 18% to 28% of the total. Prior to adding a mixture of Natural and Butadiene, rubbers possess better mechanical characteristics such as "high capabilities, minimum slippage, shrinkage, slipping, and delivers a ferocious compression-set," EVA has "low compression-set, abrasion, and tearing characteristics". These materials have exceptional 'physical and compression' characteristics together with 'least slippage' when compared to EVA/BR and EVA/NR. In the production of "hiking boots, basketball shoes, and virtually every other type of athletic footwear", the EVA midsole is an excellent 'vibration-damping' based cushioning material, which is 'light in weight, resists compression set, easily available in any color, and is made conveniently' [27,28,30].

For the current study, recycled EVA polymer matrix and leather shavings, two substantial leather solid wastes that are produced in huge quantities globally, have been discovered. Leather shavings are considered hazardous solid waste because they contain trivalent chromium, which, in some circumstances, can transform into the poisonous form known as hexavalent chromium. Traditional disposal techniques such as landfilling and incineration can easily produce secondary contamination and can't get rid of the hazardousness. In order to significantly reduce the hazardousness, it may be best to impregnate this leather waste into a recycled EVA polymer matrix. Additionally, the collagenous protein of the skins and hides creates a 3D cross-linking structure with the chromium, resulting in so-called durable leather. When this chrome-tanned leather waste is incorporated as reinforcement in the recycled EVA matrix, the physicochemical strength and thermal resistance of the composite blends will be enhanced. For instance, in the case of vegetable-tanned leather, where the chromium is absent, the degradation temperature and physico-mechanical properties of the composite blend will be inferior to that of chrome-tanned leather.

Finally, the literature review revealed that combining leather solid waste with recycled thermoplastic elastomeric polymers should result in composites with outstanding qualities [3,5,16,18,21,31]. No work related to leather-shaving fibers with recycled EVA polymer matrix composites with a 1:1 ratio has been reported by any researcher. The method of impregnating leather-shaving into a polymer in max. percent-wt. reinforcement of 1:1 is undoubtedly difficult. To enhance the 'homogeneity'/'proper blending' of the polymer and leather, however, in this work, appropriate additives, including lubricants/oils, paraffin and naphthalenes, have been used, resulting in composites with superior qualities for the desired multifunctional "value-addition" applications. From in-depth literary studies, it has been unveiled that no work has been made accessible

on carrying-out experiments regarding the efficient utilization of leather waste (Leather shavings) with max leather fiber-loading of 1:1 in combination with recycled EVA polymer matrix for footwear, transportation applications, etc., ref. [16,32–44]. As a consequence, the recommended suggestions of this research are valuable for the leather as well as polymeric industries in order to significantly alleviate a load of disposal of solid waste to the greatest extent possible. To date, no work has been reported on the utilization of solid leather waste as a reinforcement in the reused "EVA thermoplastic" matrix for the fabrication of leather waste-reinforced polymeric composites. Thus, the suggested concept of such research could be substantially effective and compelling for the leather and polymeric sectors to eradicate the strain of solid-waste management to the greatest extent practicable.

In addition, no literature has been made available in which researchers have tried to modify the processing method of solid leather waste/recycled EVA thermoplastics matrix-based polymeric composites by adding preferable additives, namely paraffin lubricants/oils and naphthalene to enhance the appropriate blending of leather and polymer thus to produce composites with better properties for proposed applications. Limited work is available on the study of mechanical, thermal, structural, morphological, CRT, and cushioning characteristics of leather waste-reinforced recycled/scrap thermoplastic polymer composites. Therefore, opportunities exist to investigate the characteristics of these leather waste reinforcement polymer composites so as to explore their usage for multifunctional applications. It is also intended to see the effect of additives and plasticizers on the physicochemical, mechanical, thermal, volumetric wear-loss, damping, cushioning/shock absorption, compression, resilience, and Morphology analysis on the fabricated polymer composites.

Thus, the study examines the potential use of "leather shaving waste" as reinforcing fibers in "recycled EVA polymeric-matrix" in light of their "intrinsic fibrous-nature", and "renewability characteristics". The intent was to explore the implications of "leather shaving fibers" on the "recycled EVA matrix" based on its "physico-mechanical", "thermal", "morphological", "chemical", "structural crystallization", "elemental mapping", and "topographic image-mapping properties". From an industrial standpoint, "EVA" is widely employed in the production of "inserts", and "soles" in the "footwear sector", accounting for 18% to 28% of total usage. "EVA" is known for its "low compression-set", "abrasion", and "tearing resistance" and its exceptional "physical" and "compression characteristics". EVA is also used as a "cushioning material" in "athletic footwear" owing to its "light weight", "resistance to compression-set", and "vibration-damping characteristics". A composite composed of "recycled-EVA polymer" combined with "leather shaving waste" is undoubtedly an attractive option, as it has the significant merit of being in accordance with the underlying concept of "circularity" throughout the underlying "production chain" for "value-addition applications".

## 2. Experimental

### 2.1. Materials

As far as the "sample location" is concerned, Laxmi Polymers, a company in Bahadurgarh, India, has supplied recycled EVA granules from the footwear industries. After confirming with the Gel Permeation Chromatography (GPC) analysis with the supplier for the "recycled EVA polymer", the average molecular weight of "Recycled polyethylene-vinyl acetate polymer" (RPEVA) is 170,000 g/mol, and the "Melt flow index" (MFI) of RPEVA can range from 0.1 to 45 g/10 min [45]. As shown in (Figure S1a,b), the solid leather wastes (Leather-shaving fibers) were gathered from the 'tanneries' of "Leather Complex", "Jalandhar Leather India Pvt. Limited", 'Punjab'. Solid leather fibers have blended properly with the recycled polymer matrix with the usage of 'fillers' and 'plasticizers' such as "zinc octadecenoate" and "octadecanoic acid" that have been procured from a local source, Impact Agencies, Industrial Estate, Jalandhar, Punjab. In the hot -press, "Poly-vinyl Alcohol" (P.V.A.) and "Teflon" (P.T.F.E.) sheets wrapped around 'steel-molds' are used as "mold release agents" provided by "Sigma Aldrich chemicals" as part of the 'post-processing' and 'finishing operations'.

### 2.2. Fabrication of Composites

Fabrication of Thermoplastic Elastomer Polymer Composites, including Leather Flakes and Recycled EVA

The ratio of recycled EVA polymer and leather shavings in the compositions of leather polymer composites is 1:1. (wt. percent). The current study's methodology is shown in sequential sequence in Figure S2. The following Table S1 presents the formulations of "polymer composite" specimens in weight -ratios based on the maximal "leather-shaving" fiber loading of 1:1.

As per a particle size analyzer, the leather shavings had an average particle size of 500 microns. In accordance with the sampling process is concerned, the two-roll mill (manufactured/exported from the Ravi Engg. Works, New Delhi, India), normally employed for rubber compounding, has been purposely used to mix the components, and the experimentation has been carried -out at local manufacturer, Impact Agencies, Industrial Estate, Jalandhar, Punjab. Optimal huge-shear blending can be achieved by orienting two rolls facing one another at 368.15 K with a user-specified preset, configurable, or customizable gap to facilitate material permeation. The rollers' spacing/gap has been kept at 1 to 1.5 mm [16,39,40,46–48]. Usually, a rear-roll rotates more quickly than a front-roll. In this instance, the front roll rotates at 10 rpm while the back roll rotates at 15 rpm. To develop the necessary flexible -sheet with specifications of 185 mm × 185 mm × 3.5 mm, dies have been fabricated. After compounding, materials were introduced in lower-dies. The heat was applied to the dies, reaching 368.15 K. After cooling in the die for 30 min, the upper die was squeezed against the lower die with an applied -load of 20 tonnes. With the use of the ejector pin, the composite material that had been compounded was removed from the die.

Two roll mills were used to mix the materials for the creation of flexible composites under specific conditions. The uniform blend was transformed to a hot-press molding, Manufacturer: Techno Search Instruments, Model PF M-15 containing preheated molds with 20 cm × 20 cm with variation in thickness of 1, 2, 3, and 5 mm. Fillers and Plasticizers such as "zinc octadecenoate" and "Octadecanoic acid" were also utilized to optimize the processing variables. The necessary flexible visco-elastic damping composite sheets will be produced by hot-press, die-molding the fine particulate-matter blended mix. To squeeze out more water, a hydraulic press was used to push the moist sheet that had developed. As shown in Figure S3a,b, the "leather-fibrous composites" were "dried and hot-pressed" under the 90 to 120-degree centigrade temperature and 77 to 108 psi of pressure applied for 20 to 40 min.

### 2.3. Sample Preparation

The standards for the performance of leather-fiber composite materials are provided by ASTM/ISO/SATRA. It is imperative to understand that 'material characteristics' differs in relation to "specimen preparation", "specimen dimension", "speed", and "environment of testing". A series of experiments are necessary for this research project. Hence, the 3–5 samples from each category were examined. The equipment utilized, the different test techniques, and the specific standards employed in this investigation are all shown in Table S2. As shown in Table S2, the specimens for each test were prepared in accordance with SATRA/ISO/ASTM criteria.

To determine how much force it can withstand the fabricated material using the tensile test according to ASTM/SATRA/ISO standards. By using Instron equipment, the maximum specimen elongation can be determined. The variations of Stress-displacement and load-displacement can be calculated by using tensile test, which determines the tensile modulus. From the tensile test readings, we can come to which material can be used to withstand the load and design it so that it can pass the quality check control of materials. High aspect ratio and young's modulus with Tensile strength, in mixture with thermostability and other physico-mechanical characteristics, make them smart materials and open new thoughts for Smart materials.

The aforementioned artificial SATRA testing standards are preserved in a computerized UTM machine. To determine the material's elongation and strength, the specimen is kept in the UTM at the right spacing and pulled until it breaks at a 60 mm/min speed.

*2.4. Characterizations*

To meet the specimen specifications set out by ASTM, SATRA, and ISO, strips of the base matrix and its composites were prepared into different sizes. The samples were then measured for thickness at a minimum of four randomly chosen places, with the findings averaged. Table S2 lists the specimen dimensions for various mechanical testing in accordance with ASTM/SATRA/ISO standards. The details of the specimen, test technique, test significance, and testing equipment parameters are covered subsequently in the corresponding sections.

Moisture content, PH, chromium trivalent, and chromium content of leather shavings that were chosen specifically for the creation of composites were measured in order to ascertain their physicochemical properties.

2.4.1. pH of Leather-Shavings

To find the pH level, 5 g of the specimen was soaked in 100 mL "distilled water" for about 16–24 h, and then the pH level was directly measured using the SLC 13 standard procedures (SLC, 1996) [28].

2.4.2. Moisture Content of Leather-Shavings

A "physical sieve-shaker" is used to produce the "fine-grade micron-sized particles" of the leather-fibers of less than 650 µm. The materials were heated for about 30 min at 100 °C in an 'oven' to remove the 'moisture'. These particles were initially dried to a moisture level of % in a small dry kiln operating between 95 and 100 °C. At 23 degrees centigrade and 0.5 "relative-humidity", properties were then measured. By measuring "3 mg of the sample in a crucible and then placing it in an oven for 5 to 6 h", the moisture content of the samples was then evaluated using the gravimetric method, SLC-3 (SLC, 1996). The results showed that the moisture content of the samples was $9 \pm 2\%$ [28].

2.4.3. Chromium Trivalent of Leather-Shavings

Chromium trivalent was identified using "SLC 8 in SLC 1996" [28]. A "5 mL of $H_2SO_4$, 10 mL of $HClO_4$, or 15 mL of the blend and 15 mL of $HNO_3$ were added to 2 g of grind chrome shaving". Using a "hot plate heated to a moderate boil, a funnel was positioned on the flask in anticipation of the reaction blend starting to convert into orange". The 'flame' was then reduced until the color change was complete. The mixture was then gently heated for an additional two minutes, quickly cooled in air, and then diluted to about 200 mL. To remove the chlorine interference, the mixture was further heated for about 10 min, allowed to cool, and then 20 mL of 10% potassium iodide and 15 mL of orthophosphoric acid were added to conceal the presence of any iron. This mixture was then allowed to stand for 10 min in a dark area. Using 5 mL of a 1% starch indicator, "the sample was titrated against 0.1 N Sodium thiosulphate until the solution in the flask turned light green (SLC, 1996)" [28].

$$\text{"}Cr_2O_3(\%)\text{"} = \frac{\text{"titration volume} \times 0.00253 \times 100 \times \text{correction factor"}}{\text{"Sample weight"}} \tag{1}$$

where 0.1 N "Sodium thiosulphate $(Na_2S_2O_3.xH_2O)$" = 0.00253 g $Cr_2O_3$

**3. Chemical Properties EVA-Based Composites**

As per "IUC 8 and 18 test standards", the UV-Visible spectrophotometry instrument (Inkarp, Sican 2600 Series model) provides the most appropriate, efficient, and precise methodology for analyzing "the chromium levels in leather savings and "Polymeric matrix composites" (PMC's) [29,30]. Therefore, "the chromium content of the control sample and the leather fibrous composites was determined" as per "IUC 8 and 18 test-standards/ISO

17072-1:2011(E): IULTCS/IUC 27-1:2011(E) test procedure" [29,30]. A 100 mL of the "acid artificial-perspiration acid" (APA) solution was mixed with the precisely weighed 2 g of ground leather board sample. The specimen was then slowly shaken in water at 37 °C $\pm$ 2 °C for 4 h $\pm$ 5 min. This APA solution was then made by blending 5 g of NaCl, 2.2 g of $H_6NaO_6P$, and 0.5 g of L-histidine—monohydrochloride monohydrate. The 'extracted mixture' was filtered with a membrane filter before being filtered initially with filter paper. An appropriate amount of extract was obtained for analysis, and nitric acid at a concentration of 5% (by volume) was added for the direct measurement of the elements. The dilution factor took this contribution into account. To control impurities using the same process, blank samples were run concurrently with the sample. A sample container is used to collect an aliquot of acid sweat, which is then treated as a sample for all tests. The amount of chromium was measured as per the 'IUC 8 test method' with the help of a 'UV-visible spectrophotometry device (Agilent 700 Series)' [30]. Nitric acid was added to a 2 g sample, then 20 mL of a blend of perchloric acid and $H_2SO_4$ in a 70:30 ratio was added. Heat was applied to the mixture until an orange color appeared. After cooling the digested sample, 100 mL of distilled water was introduced and boiled for 10 min to eliminate excess chlorine. An additional 15 mL of orthophosphoric acid was added to prevent iron interference. To avoid "light interference" to the 'iodine', 20 mL of 10% "Potassium iodide" was mixed and preserved in a "dark area for 10 min". Using starch as an indicator, "the sample was titrated against 0.1 N Sodium Thiosulphate until a sky blue color was seen (IUC/SLC, 1996)" [28–30]". By immersing "5 g sample in 100 mL of distilled water and keeping it in an orbital shaking device for 16 to 24 h", the pH of the reinforcing agent and its composites were measured directly using the SLC 13 standard methods (SLC, 1996) [28–30].

## 4. Physico-Mechanical Characteristics

### 4.1. Tensile Strength

Tensile strength is one of the factors that can be evaluated as a generic feature even though it has no direct impact on the properties and expectations of the soling materials. The fracture force/cross sectional-area is referred to as tensile strength. After cross sectional-area test specimen, the tensile tester is filled with both jaws, as shown in Figure S4. The specimen is pulled with force to ascertain. The "percent-age of elongation at break" is also calculated as it is an important parameter to determine the flexibility of the material. The tensile test on composite sheets was conducted as per SATRA TM-137 1995, using a "computer-controlled INSTRON apparatus, model 3369J7257, with an optical-extensometer gauge-length of 50 mm" [49–53]. After that, the tensile tester was fastened to the leather composite sample. Jaw separation was limited to 100 $\pm$ 20 mm per min. The leather composite sample breaking point was recorded. Before beginning the instrument ($L_1$) and when the test specimen broke ($L_2$), the tensile tester's two separating jaws' distances were measured. The %age elongation at break and tensile strength were computed. These properties are crucially used to gauge the durability of "leather and footwear-related items". Four tests were conducted for every developed composite, and their average was taken for analysis.

$$\text{Tensile Strength}\left(\frac{\text{N}}{\text{mm}^2}\right) = \frac{\text{Fracture Force}}{\text{cross} - \text{sectional area}} \tag{2}$$

$$\text{"Elongation at break (\%)} = \frac{L_2 - L_1}{L_1} \times 100\text{"} \tag{3}$$

"where, $L_2$ = final length after fracture", and $L_1$ = initial length of the specimen.

### 4.2. Compressive Strength

The compressive test on composites was carried out in accordance with ASTM D-3410 [54]. The experiments were conducted on "computer-controlled INSTRON equipment, model 3369J7257, with an optical-extensometer gauge-length of 50 mm and a test-speed of 50 mm/min". Two cross heads are used to fix the test specimen in the loading unit,

and the 'drive unit' supplies the force at the desired rate. Depending on the kind of tests, replaceable load cells with capacities of 5 KN and higher are positioned above or below the cross-heads. There are several ways to adjust, control, or vary the 'speed' of the moving cross-head. The final physico-mechanical properties of the composites were calculated by averaging results from several observational and measurement methods.

### 4.3. Tearing Resistance

Tearing resistance reflects the ability of polymer material to resist tearing. For this, tearing test samples were made according to "SATRA TM-218 1999/ISO 20344:2011" [55]. Tests were performed at 50 mm/min speed, and four tests per composite were conducted.

### 4.4. Hardness

"The penetration of a rigid ball into the test pieces under particular conditions" is measured using a device called a hardness tester to determine the hardness of a material. The Shore Using a shore test, the composites' hardness was evaluated by employing a "Shore-A hardness tester model Digitest" using the "SATRA TM-205:2016/ISO 868:2013 test" standards [56].

### 4.5. Density

The density of the material is proportionate to its weight and is represented as "mass per unit of volume". This illustrates "how a material becomes lighter or heavier means denser the material resulting larger its density" and vice versa. To estimate the 'density' of flat materials, uniformly thick test specimens have been placed in a 'circular' or 'square' container. By measuring both the 'thicknesses' and the 'diameter/length', the volume may be computed. The mass of a sample is furthermore evaluated, and "the density is derived by dividing mass by volume". It is hard to ascertain "the volume of a moulded unit-sole dye to its uneven-surface". Nevertheless, the volume of such specimen soles is measured in both 'normal' and 'immersed' conditions. "As per the ASTM-D-792-00, the weight-loss in water indicates the volume of water-replaced, which is identical to 'sole-unit', and the 'density' may then be computed" as per equation given as under:

$$\rho = \frac{w_a \times 0.9975}{w_a - (w_w - w_b)} \tag{4}$$

where $\rho$ = Density of sample, $w_a$ = Wt. of the specimen while suspended in the atmosphere, $w_w$ = Wt. of partially-immersed 'wire-holding' specimen, $w_b$ = Wt. of the specimen when completely immersed in 'distilled-water', as well as the partially-immersed 'wire-holding' the sample, and the 'density' of the 'distilled-water' at 23 °C in gm/cm$^3$.

Samples were acquired in any form with a volume to calculate a density using the equation for volume, mass, and density.

### 4.6. Adhesion Strength

"The adhesion strength of the leather shaving and its PMCs was assessed Using SATRA TM 401 2000 at room temperature" with an aim to determine how well the linings adhered [57].

### 4.7. Compression and Resilience Test (CRT)

The Compression and Resilience Test (CRT) of the PMCs and leather savings was performed on PMCs to evaluate their compressive characteristics. The tests were conducted using "SATRA TM 64:1996 at room temperature" [46].

### 4.8. Abrasion Resistance

"The wear resistance of the leather shavings and its PMCs was evaluated using DuPont-based unidirectional rotary drum abrasion model STM-140 UK tester" in accordance with ISO20871:2001 requirements [58]. At 30 revolutions per minute, composites with the proper

specifications were attached to the coarse wheel. There were 1000 cycles of testing. A rotating cylinder filled with "50-mesh emery abrasive paper" was used to measure the weight loss caused by the abrasive paper. "The specimen-holding limb was propelled by a 1500 gm weight that provided a 2.03943 Kgf contact normal force and moved parallel to the (horizontal) cylinder axes". An alternative method involved cutting leather composite samples with a conventional die, weighing them, and clamping them firmly in an "abrasive disc against emery paper". The machine was now turned on, and measurements of weight loss were made after 500, 1000, 1500, and 2000 rotations. Volumetric wear-loss is therefore calculated based on the reduction in weight of the specimen prior to and after the testing.

### 4.9. Water-Absorption Test

The specimen used in this study is a sheet of composite plate measuring 25 mm by 25 mm. After thoroughly cleaning the composite plate, the sample of the specimen being tested was measured, and the initial weight was retained for reference. The sample was placed in a jar with one liter of fresh water, kept inside, and submerged entirely in water for 12 h. After that, the water sample was taken, and the percentage of water absorption was determined by monitoring the change in sample weight over time.

$$\text{"Percentage water absorption"} = \frac{\text{"Volume (mL) of water absorbed"}}{\text{"Weight (gm)of leather specimens"}} \times 100 \quad (5)$$

## 5. Thermal Properties

"The thermal properties of leather and its composites reflect its stability under high temperature". Given the high polarity of both the leather fibers and the matrix, interactions between the two may influence how the leather fibers melt. Therefore, in this section, thermogravimetry and DSC techniques are employed to investigate "the temperature-dependent degradation of the leather-filled composites as well as the melting behaviour of both neat, base polymer with/without leather fiber".

### 5.1. Thermo-Gravimetric Analysis (TGA)

The thermo-stability testing of base material and their composites were performed with the help of "TGA/DSC analyzer TA instrument, Waters Austria type Q50". "In a $N_2$ atmosphere, a sample of 5 mg was treated in an aluminium pan and heated from 30–800 °C at a rate of 20 °C per minute. As a minimum weight loss of the sample, the temperature in the TGA thermogram that corresponds to a 5% weight loss is used". The quantitative information on "reduction in weight decomposition, and the products created on decomposition" is provided by the TGA thermogram of leather shaving and PMCs.

### 5.2. Differential Scanning Calorimetry (DSC)

The samples were conditioned for 24 h at room temperature after being pre-dried for 10 min at 100 °C in an air-circulating oven. With a thermo-gravimetric analyzer, the thermal stability of leather shavings/recycled EVA composites and base matrix with/without any leather fiber was monitored (TA instruments, Waters Austria, model Q50).

Utilizing differential scanning calorimetry, the thermal characteristics of leather-shavings/recycled EVA polymer composites have been studied (DSC). The DSC2A-00837 (192.168.1.11) was used for the tests, along with standardized aluminum crucible pans and lid coverings. The cell environment has been injected or pumped with nitrogen. The first samples were evaluated at room temperature and then "quickly chilled to a temperature of about −25 °C and then raised up to 400 °C at a rate of 10 °C per minute". They were maintained at this temperature for two minutes. The specimens were then cooled to −20 °C and heated to 400 °C at a rate of 10 °C per minute. Re-crystallization and "melting temperatures" were calculated using "heat-flow indicators", which were utilized to determine the glass-transition temperature. The initial heating diminished the material's manufacturing background ($T_g$).

DSC provides details on the thermal transitions that occur in polymers, including melting, the transition to glass, oxidation, etc. Additionally, shifts or modifications in the crystallization peaks can be used to analyze "the miscibility of the polymeric blend system".

A Q200 from A TA Instruments, Waters Austria, DSC2A-00837 (192.168.1.11) The melting behavior and thermal transitions of "neat recycled-EVA and leather fibers" when blended with recycled EVA polymer were monitored using a "differential scanning calorimeter connected to a 990 thermal analyzer". The material was put into the DSC cell in an aluminum pan that contained around 20 mg of it. As a point of reference, an empty aluminum pan was used. "DSC examination was conducted under nitrogen environment with a heating rate of 10 $°C$ $min^{-1}$ and temperature ranges of $-25$ $°C$ to 400 $°C$".

### 6. Fourier-Transform Infra-Red Spectroscopy (FT-IR)

The samples' interfacial contact has been verified using FTIR (FT/IR-4700typeA Serial Number C016661788). In transmission mode, FTIR has been performed in a humidity-free environment at ambient temperature. For the powdered material, spectra between 4000 and 600 $cm^{-1}$ were recorded. By using ATR, solid samples have been examined (Attenuated Transmission Reflectance).

The chemical properties of the neat and leather shavings/recycled EVA polymer samples were analyzed using the FT/IR-JASCO, model 4700typeA. Using a SHIMAZU IR affinity -IS spectrometer, "the FTIR spectra for collagen hydrolysate were captured with KBr pellet". The samples were reduced to a fine powder, combined with KBr, and then put in an IR cell that was stored in a container with a regular slit. As the IR and air spectra were being captured, the reference was used. "With a resolution of 4 per cm and a scan speed of 2 mm/s in the wavenumber range of 600–4000 $cm^{-1}$ at room temperature, it was captured in the transmittance-vibrational modes". This method generates a constrained compositional profile while measuring the surface composition, bonding, and structure.

### 7. Morphological and Elemental Analysis

Understanding the potential morphological changes in leather fibers during molding at high temperatures, as well as the potential alterations at the interface between the matrix and the fiber, was carried out using a "Scanning Electron Microscopy" (SEM) equipped with "Energy-dispersive analysis of X-ray" (EDAX), "a Phenom World PhenomPro model, and an accelerating voltage of 10 kV". Gold ions were used to sputter-coat samples (1 $cm^2$), acting as a "conducting medium" when they were scanned with a "scanning microscope" made by Phenom World, model PhenomPro. The surface morphology of leather fibers, the x-section morphology of "leather fiber reinforced recycled EVA polymer composites", the "degree of fiber alignment, the degree of adhesion, the fusion of leather fibers with recycled EVA polymer, the uniformity of fiber dispersion, and the extent of fiber polymer adhesion" have all been studied using SEM analysis. To determine the occurrence or existence of chemical-elemental compositions, E-DAX is examining the stoichiometric compounds and percent chemical purity of the specimens.

Without any further additives, untreated leather fibers (Shaving wastes) were included into "the recycled EVA polymer matrix and mixed either at ambient temperature or at 120 $°C$ on a mixing mill that also served as a hot-press moulding machine". The "neat recycled-EVA" polymer with no leather fiber content and the composites made of leather shavings and recycled EVA polymer was utilized for SEM morphological analyses after the mixed samples were kept for a period of 24 h.

### 8. X-ray Diffraction Analysis

An "X-ray Diffraction (XRD) analysis" was conducted to identify the chemical constituents or phases involved throughout the crystal structures of recycled EVA polymer composites. The specimens were powdered with neat Recycled EVA material of 60 shore-hardness and leather-shavings/recycled EVA polymer material of 90.5 shore-hardness on a wide-angle X-ray diffractometer (XRD), made with model name (Bruker AXS, ECO D8 Advance) at the

Chandigarh University, Mohali, India. Such diffractometer makes utilization of the copper (Cu-K) radiation sources that have been driven by 25 mA, and 40 kV. For attenuating an interference peak, K-beta filtering is employed. Divergence and scattering-slits of 0.4-degree were employed, including a receivers-lit of 0.2 mm. The composite components were positioned in a sample container, and the analysis was continued perpetually. The trial evaluation was performed by observing the diffraction-patterns between in the 2-theta ranges from 5-degrees to 90-degrees at a scanning-rate of 2-degrees/min and a scan-step of 0.05-degrees.

### 9. Atomic Force Microscopy (AFM) Analysis

An optical-microscopy, and an intermittent-contact atomic force microscopy (AFM, Make NT-MDT, Ireland, Model name, NTEGRA PRIMA) were used to investigate its molecular structure, leather-shavings/PMC phases; to analyze their cross sectioned area.

AFM has been utilized to study the adsorption of "neat recycled EVA and recycled EVA polymer composites on the leather fibers". The most powerful objective lenses and phase contrast microscope (Leica DM 750) were used for the microscopic morphological research (i.e., $100\times$ lens).

A Cryomicrotome equipped with a diamonds-knife is being used to create test specimens for AFM experiments, and the samples were kept at $-80\,^{\circ}$C. A microscope was used to record "the AFM pictures in tapping mode (TM) under room-temperature". Ambient-air (AFM, Make NT-MDT, Ireland, Model name, NTEGRA PRIMA). "Commercial silicon-tips with a spring-constant of 24–52 N/m, a resonating frequency of 264–339 kHz, and an average radius of the curve throughout the 10–15 nm region was used as probing for the test". This study used the greatest sampled resolution accessible to generate phase-detection images. The recycled EVA thermoplastic polymer matrix will be enhanced by incorporating fillers, plasticizers such as stearic acid, zinc stearate, and other lubricants such as paraffins and naphthalene oils.

AFM has been utilized to determine the size (thickness, width etc.,) of the Interphases and their stiffness relative to the bulk phase of PMCs and their constituents. AFM has been utilized to examine the AFM Roughness Analysis and AFM Grain Analysis of the "neat recycled EVA polymer composites, and leather-shaving fibers reinforced with the recycled EVA polymer composites".

### 10. Results and Discussions

*10.1. Moisture Content, Chromium Trivalent and Chemical Composition*

As a result of the uniform-ragged powdery form-structure, the leather-shavings exhibit a "moisture content" of 5.12%. As a result, drying time for leather shavings waste must be increased, or another "drying method" must be selected further.

Additionally, UV-Visible spectroscopy was used as a quick method of detecting Cr(VI) in the "solid leather wastes/recycled EVA thermo-plastic polymer composites". The absence of Cr(VI) in thermoplastic elastomer-based polymer composites made from recycled materials or leather shavings is demonstrated by the results. Results from Tables S3 and S4 indicate that the tested leather shaving waste-reinforced recycled EVA thermoplastic elastomeric polymer composite samples did not contain any hexavalent chromium.

*10.2. Physicomechanical Characteristics*

The tensile slabs were molded in a "hydraulic press with electrically heated platens operating at 110 $^{\circ}$C and 6 tonnes of pressure" to evaluate physical and mechanical properties. The mold was filled with the compounded sheet, which had a thickness of about 3.54 mm, and the platens were compressed to seal.

The percent leather shavings as fiber concentrations had an indispensable effect on the "physico-mechanical characteristics of the solid leather wastes/recycled polymer composites". The "neat recycled-EVA polymer" tensile strength was discovered to be 12.295 MPa. However, it was discovered that recycled EVA polymer composites using leather shavings as reinforcing fiber had an average tensile strength of 10.615 MPa. This behavior was attributed to compelling interface bonding or adhesion strength and the fact that the fiber

agglomerations are closer together with a 1:1 ratio of reinforcement and bas matrix. The stress may have been transmitted from one aggregate to another as a result of the recycled EVA polymer acting as an "adhesive-agent" between the "coalescence agglomerations". Thus, it is abundantly obvious that the addition of leather shavings as fiber causes enhances ductility and a decrease in the "elongation at break" of the base matrix.

The elasticity of the composite materials can be shown in the % elongation at break. As shown in Figure 1, the "leather shavings/recycled EVA polymer composites" showed lower "percentage elongation at break" of 67.91 percent as compared to the control sample, i.e., "neat recycled-EVA polymer", which is discovered to be much higher at 192.16 percent. This low value of "elongation at break" demonstrate how stiffer the leather fibers become when combined with the recycled EVA polymer matrix. As the "leather loading in the polymer" grows, the failure process becomes more brittle [16,18,35,37,46].

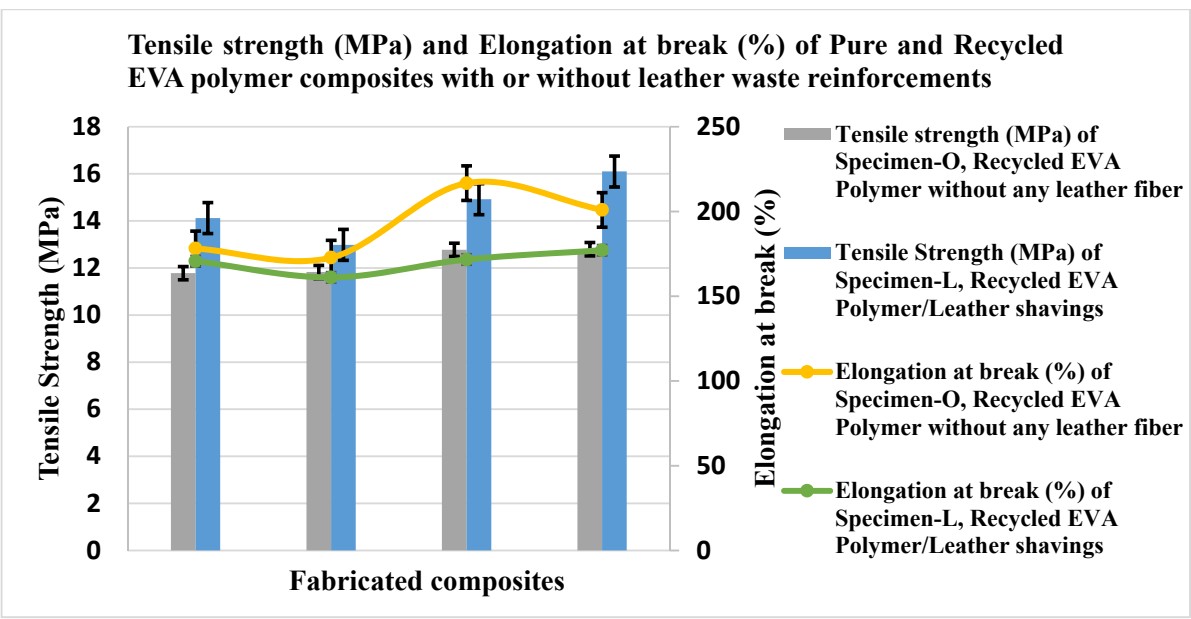

**Figure 1.** Comparison of the "tensile strength" of "neat recycled EVA polymer composites" and "Leather shavings/recycled EVA polymer composites".

The production of "crevasses, deformations, perforations, microscopic cracks enclosing the fillers, and the emergence of void cavities" due to a localized "detachment of the recycled EVA matrices from the leather fabrics" could be blamed for the poor elongation at break values [35,37,39,40,47,48]. As the composites get closer to brittle failure, there is a significant reduction in the elongation at break values. When filler loading is large (1:1), voids develop that compromise the structural integrity of the composites. The matrix and fiber become separated when the voids expand and interact with one another. The reinforcement provided by the leather fibers is outweighed by the increased contact between the nearby spaces. Additionally, each filler particle experiences normal stresses while under a tensile load. The most vulnerable locations are those where these pressures are greatest since that is where "the separation of the recycled EVA matrix from the leather fiber is most likely to begin". The leather fibers may also selectively form a cohesive network between themselves as the loading increases and reach up to 1:1, which causes a weak interfacial bonding with the "recycled EVA matrix" and poor mechanical qualities. Even so, "the leather shavings reinforced recycled EVA polymer composites" still display an exceptional outstanding mechanical strength when compared to the features of these composites with the prior discoveries [16,18,35,37,39,40,46–48].

The related results in context with the distribution of stress from one aggregate to another one have fervently been reported by Ambrósio et al. (2011), where, as the percent of leather-particulates in the composite materials keeps increasing, the tensile-strength gradually increases [37]. The physico-mechanical characteristics of leather-fibres/polymeric-

composites have been ascertained not only from the properties of the polymer-matrix and reinforcing leather-fibers, as well as by the interfacial seen among the matrix and fiber-reinforcement, which would be absolutely crucial for the transmission of inter-facial stress [37]. A sudden drop in the tensile-strength is a widespread occurrence in thermoplastic materials loaded with natural-filler particles; as the filler-content rises, thus also continues to increase the interfacial contact-area, the interfacial-adhesion, and bonding-strength between both the polar-solvents based-filling materials and the matrix-polymer deteriorates/decomposes [59–61].

The coupling inhibitors and compatibilizing-agents enhance the tensile strength of composites by lowering the gap between the tensile strength of "neat polymer-composite matrices" and that of the composites with "unprocessed natural fabrics" [37]. An enhancement in the tensile-strength implies that stress is transmitted from the matrices to the filling additive materials [37].

According to the SEM-fractography analysis, the findings have indicated that only certain cryogenic-fracturing cracks, fissures, and deformations of the leather/poly(vinyl butyral) (PVB) specimens happened via aggregates or coalescence or agglomerations, revealing that the interfacial-interaction transmitted the stresses of poly(vinyl butyral) (PVB)-matrices to the leather-fibers [37]. Even though enhancing "the wt.% of leather-reinforcing in the composite materials diminishes tensile-strength, excellent interfacial-bonding interaction among the PVB-matrix with the vinyl-alcohol-hydroxyl as well as R1-C(=O)-C-R2 functional-groups and the leather-fiber with their hygroscopic-collagenous fibril molecules might well have improved strength properties" [37]. This could lead to a favorable equilibrium and, as a result, an enhancement in physico-mechanical characteristics.

As the wt.% of leather-fibers tends to increase, the gap-distances/spacing between the fabric agglomerations reduces, which can be evident in the SEM-fractography analysis for the composites with max. percent wt. of leather-loading [37]. Due to the obvious superior interfacial-bonding and crosslinking-density mechanisms of poly(vinyl butyral) (PVB) with the leather-fiber aggregates, the PVB may very well have behaved/functioned as an adhesives binding-agent among both of the agglomerations, thereby distributing the stresses of one coalescence conglomeration to the other one [37].

The leather-fiber content had a significant effect on the physic-mechanical properties of leather/poly(vinyl butyral) (PVB) composites. The tensile strength of composites was lower than those of base PVB. Whenever the volume of leather-fibers in the composites has been raised, the tensile strength is considerably enhanced to a substantial extent [37]. This mechanism was attributed to strong interfacial interaction with a firm bonding and the notion that perhaps the fabric-agglomerations get nearer to one another as the wt.% of leather-loading rises. As a result, a PVB might well have functioned as a bonding agent between the agglomerations, transmitting stress from one to the next [37]. The tensile-modulus of the composites increased substantially as the quantity of leather-fiber within composites increased. This phenomenon was supported based on the premise that the leather-fiber agglomerations are rather near to one other, posing the possibility of entanglement [37]. Because of the elasticity of the leather-fibers and the matrices of plasticized-PVB, the leather-fiber agglomerations might even get nearer and undergo tangling or entrapment as the wt.% of leather-loading increases [37]. The entanglement agglomerations encompassed by the PVB-matrices have the potential to significantly improve the elasticity, modulus, and strength of the composites and, thus, the young's modulus [37].

If the matrix and the fiber have sufficient adhesion, the interfacial strength is often improved. The internal resistive power to come apart the sample is controlled by the "stress-concentration" across the region of the "nano-additives or nano-filling agent" if the "interfacial strength" is greater than the matrix "cohesion force". In these circumstances, it is common to see an increase in modulus, a sharp decrease in elongation at break, and an increase in tear strength. Since the variation of tensile strength relies on the loading, the kind of fiber, the makeup of the matrix, and other factors, it normally does not show any particular trend. When the fiber concentration is below the threshold volume percentage, composite materials can have tensile strengths that

are even lower than those of the unfilled matrix [62]. However, as the fiber concentration rises above the "critical volume fraction", mechanical properties get better.

The comparable findings have been observed by Nanni et al., wherein, Poly(12-aminododecanoic acid lactam) and thermoplastic poly-urethane-predicated compositions revealed desired behavior (about +47 percent modulus and +40 percent creep-resistance) even though not-optimum fabric-matrices interfacial-adhesion, as well as inadequate defibration of leather-fiber wastes, substantially decreased strength along with elongation-at-break [38]. Eventually, the thermoplastic elastomeric-based specimen behaved the poorest leading to a complete lack/absence of compatibility interaction and affinity between both the polar-molecules-based-leather-waste fibers as well as the non-polar based-polymer-matrices [38].

Leather fiber agglomerates could be brought closer together and demonstrate an unbreakable link thanks to the flexibility and toughness of the "leather fibers and the plastic recycled EVA matrix", as depicted in Figure S5a,b. Leather shavings could also be entangled as much as possible according to these properties. According to Ambrosio et al., the linked collective "covered in the recycled EVA matrix could significantly improve the stiffness, elasticity, and stability of the composites", resulting in an elastic module (2011) [38].

Dodwell (1989) claims that "expensive and light footwear could have been made from "leather-boards having a tensile strength of 4.0 MPa, fashionable, aesthetically pleasing, and practical footwear could have been made from leather-boards having tensile strength of 5.5 MPa" [63].

A wide range of behavior possibilities is also shown in Figure S5a,b, ranging from "hard, brittle" to "ductile", with a yield point similar to that of a "thermoplastic polymer" when subjected to mechanical agitations (stresses) [64].

The matrix had already effectively received the stiffness of the reinforcement. According to Covington (2009), the higher-modulus at a 10% waste-content could be explained by the HDPE90/Chrome tanned wastes' amorphous form after being chilled from melt [47].

The "neat recycled-EVA" polymer sample's tensile-fractured micrographs are shown in Figure 2a,b. The surface is lined and filled with "bulged, twisted, and elongated materials", which appears to be a sign of such "lateral-pressure bearing strength, significant deformation capacities as might be anticipated for relatively flexible recycled EVA co-polymer". It also supports the "enormous strains as measured through experimental studies with a tensile strength of 12.295 MPa and percent-elongation at break of 192.17 percent as the comparable values by Stael et al. (2005)" [39]. Figure 2a,b has illustrated the fractography surface-morphology analysis, which evidently unveiled that the surface is encompassed by stretch material, which seems to be an indication of the significant distortion abilities that might be envisaged with the exceptionally elastic and resilient EVA polymer, so it appears to confirm the huge strains analytically reported. Comparable findings have been reported in prior studies conducted [16,21,26,35,37,39,46–48,65].

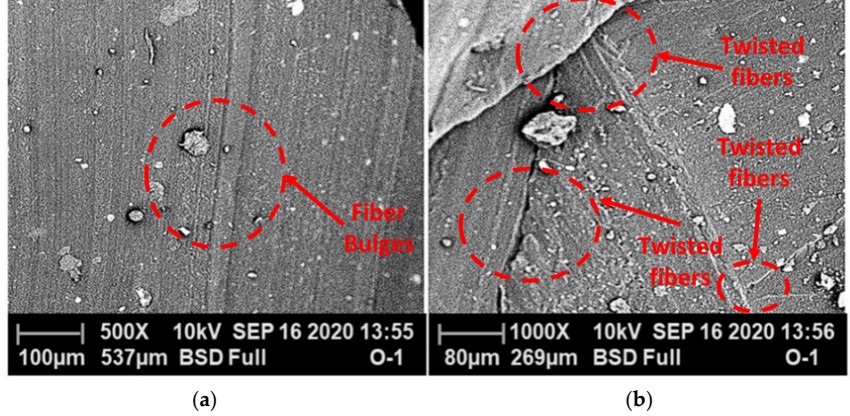

(**a**)  (**b**)

**Figure 2.** "SEM tensile-fractured micrographs of Neat recycled EVA polymer without leather fibers" at a magnification of (**a**) 500× and; (**b**) 1000× Adapted from the reference [45].

Figures 3a,b and 4a,b shows certain perforations, fractures, fissures, holes, and voids in the surface as well as the cross-sectional area of the recycled EVA polymer surface in addition to the collagen fibrils-bundles. This may be caused by fibers from leather shaving waste being removed after tensile testing, as shown in Figures 1 and S5a,b. It was simple to identify the cause or position of fiber rupture or deformation close to pull-out apertures, indicating poor adhesion between leather shaving fibers and recycled EVA polymer. However, increasing the number of leather shaving fibers in composites improved their permeability, bonding, and interfacial-adherence to recycled EVA polymer. In fact, as demonstrated in Figures 1 and S5a,b, the distribution of leather shaving fibers in composites made of recycled EVA polymer and leather shaving fibers was fairly intense. The leather shaving fibers appeared to be getting smaller, proving that the bundle of leather fibers had been broken up into individual single fibers. As a result, every fiber in the recycled EVA polymer matrix can effectively shift and reorient, resulting in a reasonable distribution of leather-shaving fibers. However, as leather fiber volume increased, the region of the stress-relaxation gradient curvature was dramatically boosted by a decrease in fiber-matrix interaction. Additionally, the strong bonding between the leather shaving fibers and recycled EVA polymer shows a remarkably large improvement in the physico-mechanical properties of the composite materials made from leather shavings and recycled EVA.

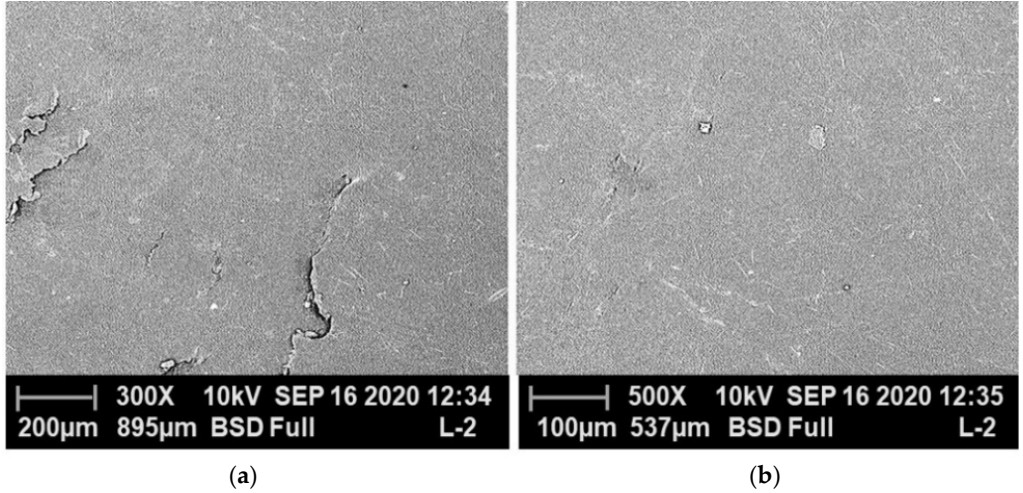

(a)          (b)

**Figure 3.** "SEM tensile-fractured micrographs of the surface of Leather Shavings/recycled EVA polymer composites" at a magnification of (**a**) 300× and; (**b**) 500×.

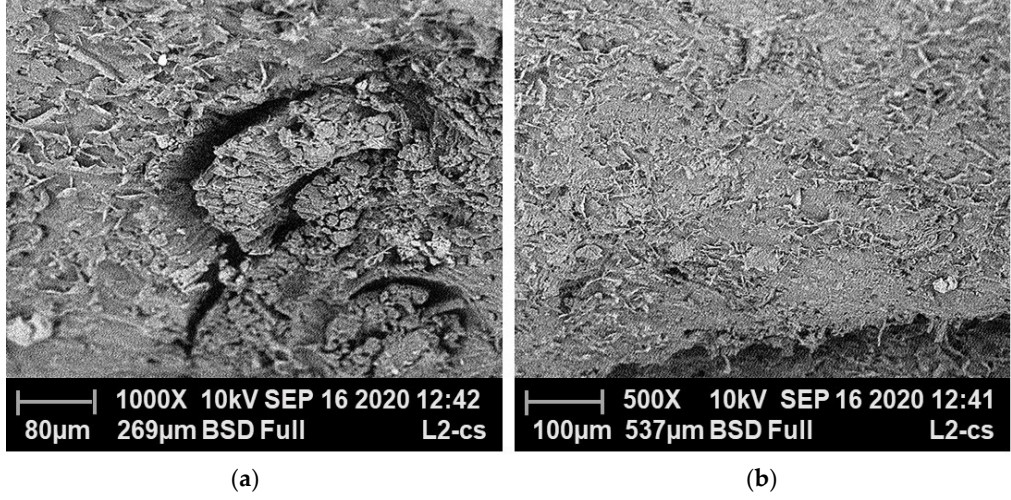

(a)          (b)

**Figure 4.** SEM tensile-fractured micrographs of the cross-section of "Leather Shavings/recycled EVA polymer composites at magnification of, (**a**) 1000× and; (**b**) 500×".

Leather shavings/recycled EVA polymer composite, a representative specimen, has a lower average compressive modulus, measuring 63.63 MPa. As shown in Figure 5, the average compressive modulus of "plain recycled EVA polymer" without fiber content was determined to be 66.695 MPa. Though the compressive modulus is a measure of "the material's stiffness, and higher is the compressive modulus which leads to stiffer in the recycled polymer composites" [16,21,26]. The comparable outcomes have been unveiled by the prior studies wherein the investigators have enumerated that brittleness is a characteristic of polymeric materials that have been simply cracked, fractured, distorted, ruptured, and/or shattered [16,21,26,46]. As brittleness is characteristic of the materials that fracture once stressed but seem to have minimal ability to degrade/deform prior to fracture or breakage. Brittle materials exhibit minimal deformation/distortion/cracking, a lower-capability to withstand/bear the shock as well as vibrating deformation, a relatively higher compressive-strength, and a poor tensile-strength [16,21,26,35,37,39,46,47]. At the same time, stiffness seems to be a property that indicates resistance to cracking and therefore is considerably easier than physico-mechanical strength, which has been associated with failure breakdown [16,21,26,35,37,39,46,47]. Also, stiffness is a polymeric characteristic represented by the Flexural-modulus as well as bending elastic-modulus. The three-point bending-test was utilized to measure a material's stiffness/resistance to bending whenever a loading is acting perpendicular towards the longest side of a specimen. The stiffer and denser the materials, the significantly larger the bending modulus of elasticity; the lesser the bending modulus, therefore the more flexibility it is indeed [16,21,26,35,37,39,46,47]. As reported from the literary sources wherein the difference between stiffness and brittleness has been reported in such a manner that the stiffness alludes toward the leather polymeric composites that are rigid/stiff, inflexible, unyielding, difficult to bend, as well as inflexible, whereas, brittle pertains to that of leather polymeric composites that would be inflexible and likely to fracture, crack, shatter, or rupture quickly underneath applied stress or pressure [35,37,39,46,47].

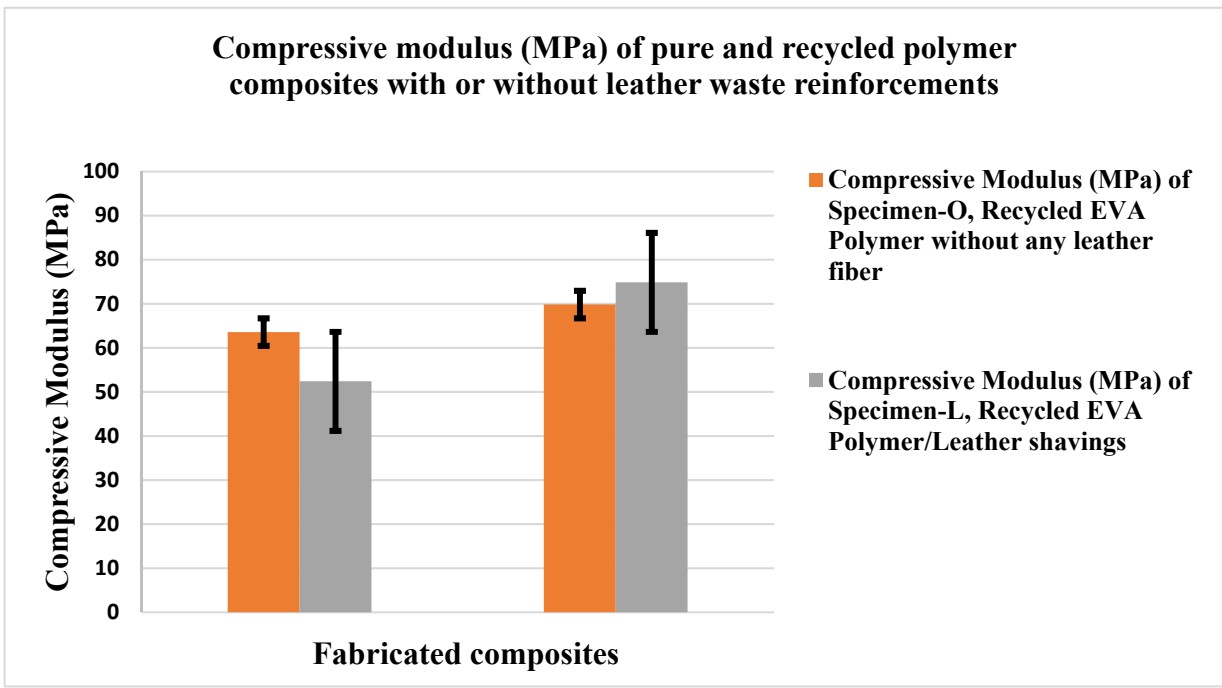

**Figure 5.** Comparison of "Compressive modulus (MPa)" for "neat-recycled EVA" and "recycled-EVA polymer composites".

As shown in Figure S6, the compressive strength for the leather shavings/recycled EVA polymer composite tends to be somewhat higher (9.88 MPa) than the plain recycled EVA polymer (9.87 MPa).

The "stress-strain profile of thermoplastic polymers" is consistent with the nonlinearity of the stress-strain curve. According to Musa et al. (2017) [66], the control (HDPE) had a higher strain than its composites because the filler had not been used to temper the ductility. The stiffness and brittleness of a filler have infused its stringency into some of the "most ductile matrix (HDPE)" that results in reducing strain at minimal waste content (within 10–20 wt.% of fiber). Results also demonstrate that the composite "HDPE90/Chrome-tanned waste at 10% has higher stress and strain than the control (HDPE)". Due to the presence of natural rubber, the strain was further impacted (serving as an extender), and the filling dispersion in the matrix was facilitated. As a result, "HDPE90/Chromium-tanned waste with a 10% loading had become more robust and ductile".

The "solid leather waste/recycled EVA composites" tear strength also shows how dependent this feature is on the leather waste, which has the highest fiber concentration in the composites. The average tear strength, however, was discovered to be around 17.5575 N/mm when leather shavings were used as reinforcement fibers. The average tear strength for "the "neat recycled-EVA" polymer was determined to be 9.48 N/mm", as shown in Figure 6. According to Madera-Santana et al. (2004) [16], the highest leather fiber percentage increases tear resistance. This is due to the collagenous (proteinous) fibrous shredded form of "leather shavings" in the "recycled EVA matrix" being assimilated ("uniformly dispersed/blended with matrix").

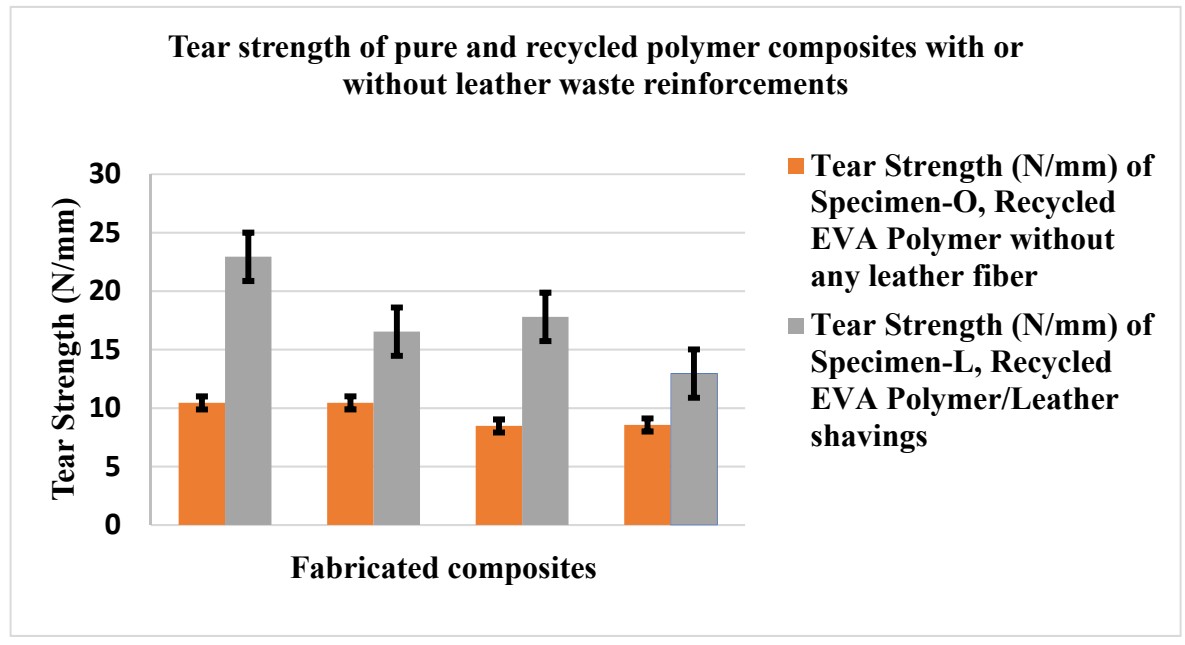

**Figure 6.** Comparison of "tear strength (N/mm)" for "neat-recycled EVA", and "recycled-EVA polymer composites".

The presence of "additives, according to Ambrósio et al. (2011) and Musa et al. (2017), has increased the fiber dispersion in HDPE-matrix, enabling superior compatibility and interacting between the waste fibers and matrix" [38,66]. The main function of the additives was "to act as a chemical group, creating an interface between the matrix and the fiber surface that allowed for effective stress transfer". A lack of a wetting surface could cause the deterioration of greater waste contents by drastically reducing stress transfer through the interface.

According to Ravichandran and Nmg (2005), the mechanical characteristics of "NBR/PVC composites filled with treated leather fibers" have significantly improved [67]. The elasticity modulus of the mixes is increased when leather fibers are added. Ammonia-treated leather shavings significantly increase tensile strength and tear strength. It is possible to ascribe the strong polymer-leather contact and effective vulcanization properties to the increased modu-

lus and decreased elongation at break values. It is also well known that a composite always has a larger effective surface fracture energy than an empty polymer. Dispersed leather fibers lengthen or delay the crack propagation process. Additionally, they take in some of the energy, which slows down the matrix's deformation. The enhanced "tear strength" of the composites with leather filling makes this process more obvious. This is generally in agreement with the observation that short fiber additions enhance the "tear strength" of elastomers. "Low elongation at break" values could be related to the development of crevasses, punctures, microscopic cracks surrounding the fillers, and the "emergence of void cavities" induced by a localized detachment of the matrix from the fibers [67].

The mean peel strength among the polymer to synthetic adhesive based-fabrics for leather shavings as fiber in "recycled EVA matrix" was higher and was found to be around 1.3125 N/mm, according to Figure 7 "(as the "Adhesion-peel strength" is the 'anti-stripping' property from 'polymers' to 'leather'/'Adhesive bond-strength' or 'peel-test' is performed to estimate the 'strength' of a 'bond' by ascertaining the force that the 'force' is necessary to "peel-apart" the 'fused materials')". This is because after milling, a significant amount of leather shavings has been obtained, enhancing the interaction between shaving waste and recycled -EVA. While the 'neat recycled-EVA' was found to have a reduced adhesive peel strength of about 0.675 N/mm.

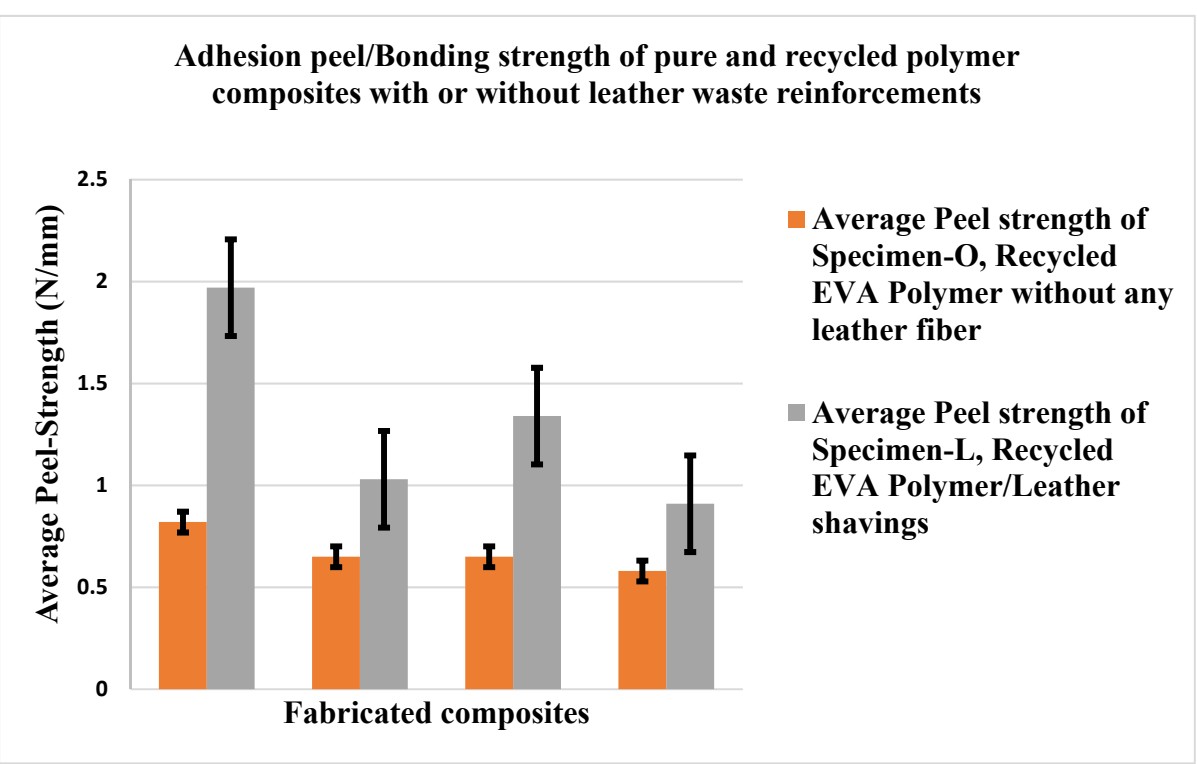

**Figure 7.** Comparison of "adhesion peel-strength (N/mm)" for "neat-recycled EVA" and "recycled-EVA polymer composites".

According to the test results for 'neat recycled-EVA' polymer, the combination of an upper and a sole has weak adherence. In this case, separation or disintegration of the 'adhesive-film' at the surface was noted in relation to textiles made of synthetic adhesive-based materials. This may also be due to 'insufficient roughing', 'inadequate surface-preparation', and 'insufficient drying-time'.

When evaluating the adhesion among "recycled EVA polymer to synthetic adhesive based-fabrics and solid leather waste reinforced recycled EVA polymer composites to synthetic adhesive based-fabrics", the results clearly show that the upper surface fabrics failure occurs in the case of the 'neat recycled-EVA' polymer composites, as shown in Figure 7. While throughout the experiment, the leather fibers in the 'recycled-EVA' polymer

composites with synthetic adhesive-based fabrics and leather shaving waste experience adhesion to top surface failure.

"Collagen-hydrolyzates" was produced by chrome-shaving during the early phase, claim Ali Shaikh et al. (2017) [66]. It was a viscid, frequently sticky, slightly cream- or pale-yellowish, odorless gel-like substance with a pH of 8.15. Due to its intense interfacial-bonding strength, this solvent-based substance was extensively used in the leather and garment industries. The drying process takes only around 10 to 15 min because the adhesive is obviously present, which is helpful for future improvement, efficacy, and functioning. As a result, there was a good chance of getting amazing results from the study that was done to create solvent-borne sticking agents. The four samples were used in the studies, which included different amounts of poly-vinyl alcohol (1 to 6 percent) and poly-vinyl acetate (1 to 16 percent). All four of the aforementioned samples, including the one stated above, showed average adhesiveness and bonding ability. After incorporating the adherent into a leather specimen, the "SATRA-TM416 test" was used to measure the bonding strength. The mean "peel strengths" of the specimens were "0.00312, 0.00325, 0.00295 and 0.0025 N/m", according to the results [68].

The thickness of the leather fabric has an impact on "compression resistance", and "hot-press/rolling mill processing", which is intended to improve the finish, is likely to improve the handle. The compressional resilience will decrease as the fiber becomes softer.

According to the findings in Figure 8, "the compression deformation property for leather shaving wastes as fiber in the recycled EVA matrix was greater by about 6.69 percent as compared to the neat recycled EVA matrix, which was found to be about 1".

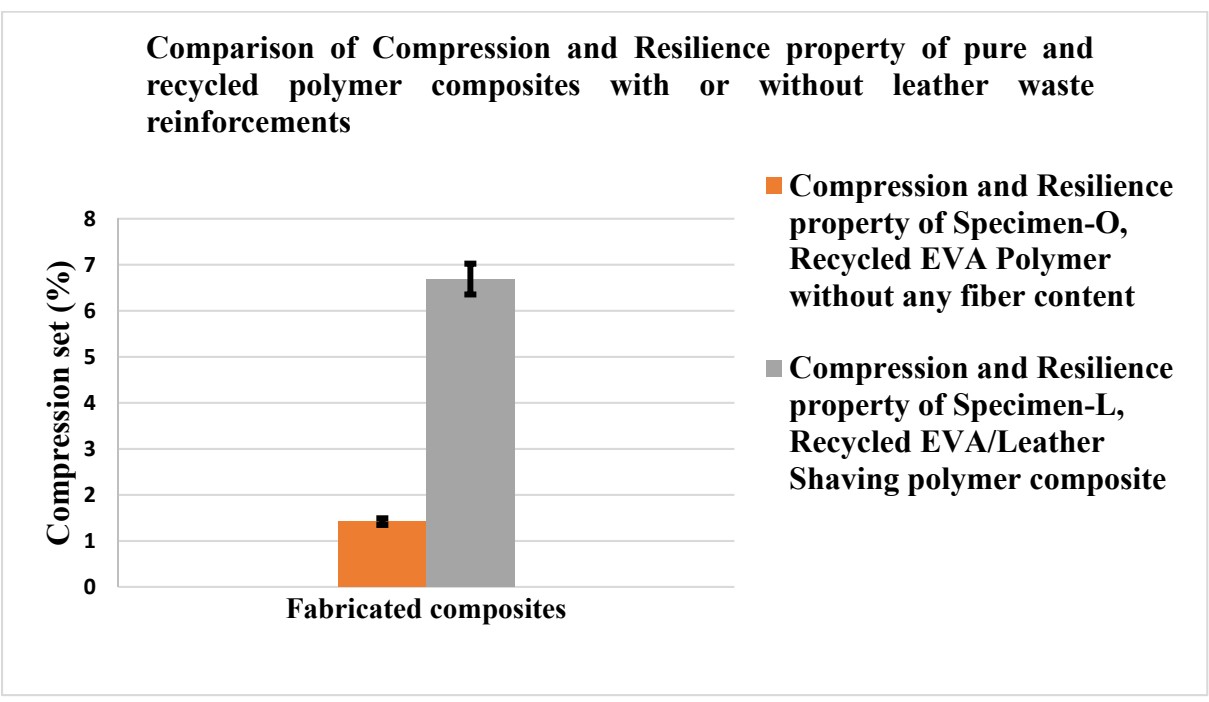

**Figure 8.** Comparison of "Compression and Resilience (%)" for "neat-recycled EVA" and "recycled-EVA polymer composites".

To determine "the compression deformation properties of "neat recycled-EVA", as well as leather waste/recycled EVA polymer composites, the Compression and Resilience Tester (CRT) of the recycled EVA polymeric composites" was investigated.

This 'CRT test' demonstrates that the 'resilience-energy' and 'load-compression' capabilities of the "leather shavings/recycled EVA composites" are noticeably better than those of the "neat-recycled EVA polymer composites".

Composite materials have superior elastic recovery when the compression set is lower. The fullness and compressibility of shaved leather fibers increase with higher compression

energy levels. The values of compression energy rise as the thickness values grow. One possible indicator of softness is the compression-decompression curve's linearity. A leather fiber with a softer grip has a lower linearity of compression. It follows logically that "the linearity of the compression and compression energy values" should be used to describe the softness of solid leather fiber wastes.

It has been discovered that as the fiber thickness of solid leather waste increases, both the "linearity of the compression" and "compression energy" values rise.

To increase compression and toughness, it is essential to prevent chain slides (CRT). To lessen chain sliding, both the chemically-bonded polymer and organically modified nanoclays can be employed (organo-clay). The reduced "molecular-mass tri-methoxy-silyl-modified poly-butadiene (Silicon Hydride)" has been further used to chemically bond polymer and organoclay, according to Park et al. (2008)'s analysis [69]. The "poly(ethylenevinyl-acetate) co-polymer and the ethylene, polymer with 1-butene" can both be silane-grafted using peroxide [70–76]. According to Park et al. (2008) [69], "polymer with 1-butene radicals" was created as a result of peroxide interactions with "poly (ethylenevinyl-acetate) and ethylene". These produced radicals may combine with "poly-butadiene of silicon hydride". This led to the development of ethylene polymer with "1-butene" and "silicon hydride-grafted poly (ethylenevinyl-acetate)" [69]. Organoclay hydroxyl groups and silicon hydride silanol groups can react. Although CRT is one of the most important qualities for foam applications, "poly (ethylene vinyl acetate)/Ethylene, polymer" with "1-butene foams with clay addition" should have better compression-set properties [69]. Park et al. (2008) employed "chemical-bonded polymers and organoclays by Silicon Hydride in his research to restrict chain-slip across the clay substrate material since it was stated that the polymeric backbone chains slipping across the clay surface was the reason of the poor elastic recovery" [69]. Because of this, "poly(ethylenevinylacetate)"/'Ethylene', 'polymer with 1-butene'/'methyl tallow bis(2-hydroxyethyl) quats-nanoclays'/'Silicon Hydride-foams' either with or without" 'cis-Butenedioic anhydride-grafting', and 'poly(ethylenevinylacetate)'/'Ethylene', polymer containing 1-butene polymeric foams" have significantly lower compression sets According to CRT results, "methyl tallow bis(2-hydroxyethyl) nanoclays" are superior to "di(hydrogenated tallow)di-methyl-ammonium-chloride nanoclays" for enhancing compression-set because they contain hydroxyl-groups within the organo-clay layer and "alkyl-ammonium ion hydroxyl" groups within the active layer [69]. Based on the absence of the large pinnacle at 3398 cm$^{-1}$ for "poly (ethylenevinyl-acetate)/Ethylene", polymer with "1-butene/methyl tallow bis(2-hydroxyethyl) quats-nanoclays/Silane" and "poly (ethylenevinyl-acetate)/Ethylene", polymer with "1-butene/Di(hydrogenated tallow)di-methyl-ammonium-chloride" qu (2008) [69].

Abrasion resistance is a crucial quality requirement in any situation where resilience is crucial, such as in "automobile" and "upholstery leathers". Abrasion is significantly influenced by the physical-mechanical properties and size of the leather particles. The main determining factors that affect abrasion are the type of leather, its "fineness or flowability, and the length of the fiber. Higher elongation, fatigue strength, strain rate, elastic recovery, and work-of-rupture or breaking characteristics in leather fibers" enable them to withstand frequent, repeating distortions and attain higher levels of abrasion resistance.

Abrasion resistance is significantly predisposed by the "leather fabric thickness" and mass per square meter, which are important structural features of leather materials. The enhanced values of these parameters determine the relatively greater abrasion resistance. It yields improved "abrasion resistance" due to the high 'surface-area', which maximizes 'contact-points' with the recycled-EVA matrix.

Utilizing a rotating drum-type abrasion tester made of leather soles, "solid leather waste/recycled EVA composites" was tested for abrasion resistance. Results showed that "when compared to neat recycled-EVA polymer composites, the volumetric wear loss for the leather shavings used as fibers in the recycled EVA matrix was higher at about 237.53 mm$^3$", as shown in Figure 9. "The thread-like collagenous fibrillary-like filaments throughout the composite" have been worn almost in proportion to their content as the

percent-wt. of leather has increased, revealing a confirmed "linear increase in wear-rate" as described by Ambrósio et al. (2011) [38].

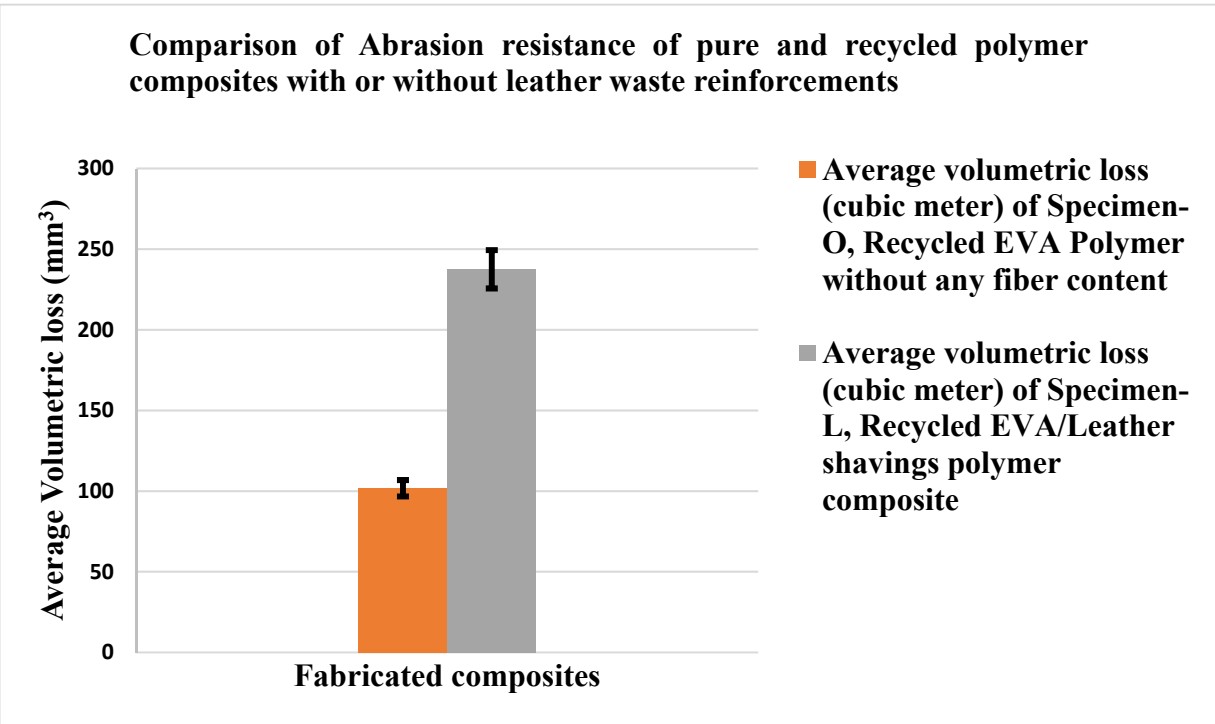

**Figure 9.** Comparison of "abrasion-resistance (mm$^3$)" for "neat-recycled EVA" and "recycled-EVA polymer composites".

The 'neat recycled-EVA' composites have exhibited a volumetric wear loss of about 101.75 mm$^3$ due to their "high elongation, work fracture, and elastic recovery".

"The shape, structure, and type of additive elements, such as zinc octadecanoate and Octadecanoic acid, and the resulting discontinuities in the matrices may serve as evidence of the reduction in wear resistance, tensile strength, percentage elongation, as well as flexing resistance with increasing waste filler inclusion". The void cavities can multiply in size, form, and relationship to one another due to abrasion, strain, and stretch, which results in materials debonding and fracture/rupture failure.

For testing composites' hardness, an indenter driven into the material is used. The modulus of elasticity and visco-elastic properties of materials has an impact on the hardness indent because it is inversely proportional to a penetrating value.

"The Shore A Hardness strength for leather shaving waste used as fiber in the recycled EVA matrix was determined to be approximately 90.5", as shown in Figure 10.

However, when compared to composites made of "recycled EVA and leather shavings, the 'hardness strength' for "neat recycled-EVA" was found to be much lower, at roughly 60. The Beach, The number of leather fibers added significantly increases the hardness of composites made of recycledEVA and solid leather waste. According to Ambrósio et al. (2011) [38], leather filaments have a harder surface than plasticized PVB matrices, which is where the growing trend in hardness originates from.

But the increase in modulus and hardness suggests some degree of interaction between the polymer and the fibers of the leather. These characteristics, together with "the high hardness ratings of these blends, may be due to the uniform distribution of the leather fibers".

"The composite's resistance to elastic deformation on its surface increased as the proportion of chrome-tanned wastes improved from 0 to 40% loading" [66].

This resilience, however, diminished as the proportion of wastes browned in chromium rose.

In general, the "density" and "voids content" of composites have a significant impact on the amount of water absorption. There is a significant increase in "water holding capacity" with increased leather-shavings 'fiber-length'. As a consequence, it is clear that the speed of water absorption increases with leather-fiber loading, and leather shavings have a high percentage of water absorption (about 14.07 percent) compared to "neat recycled EVA polymer composites" (1.56%), as shown in Figure 11.

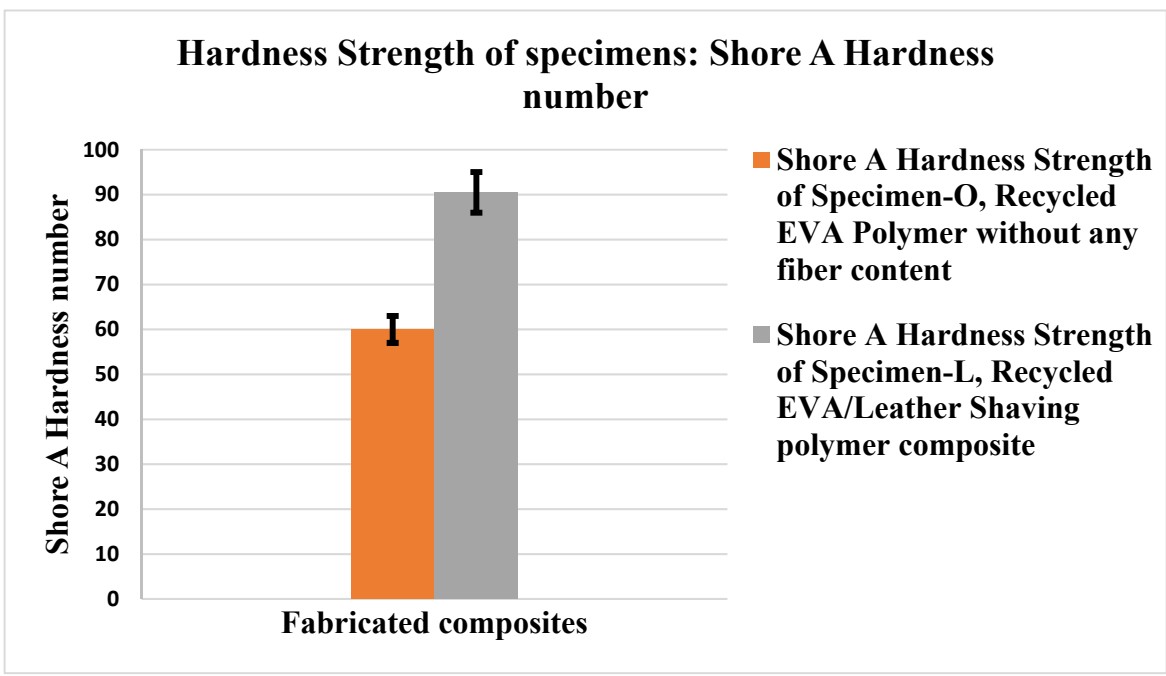

**Figure 10.** Comparison of "Hardness (Shore A)" for "neat-recycled EVA" and "recycled-EVA polymer composites".

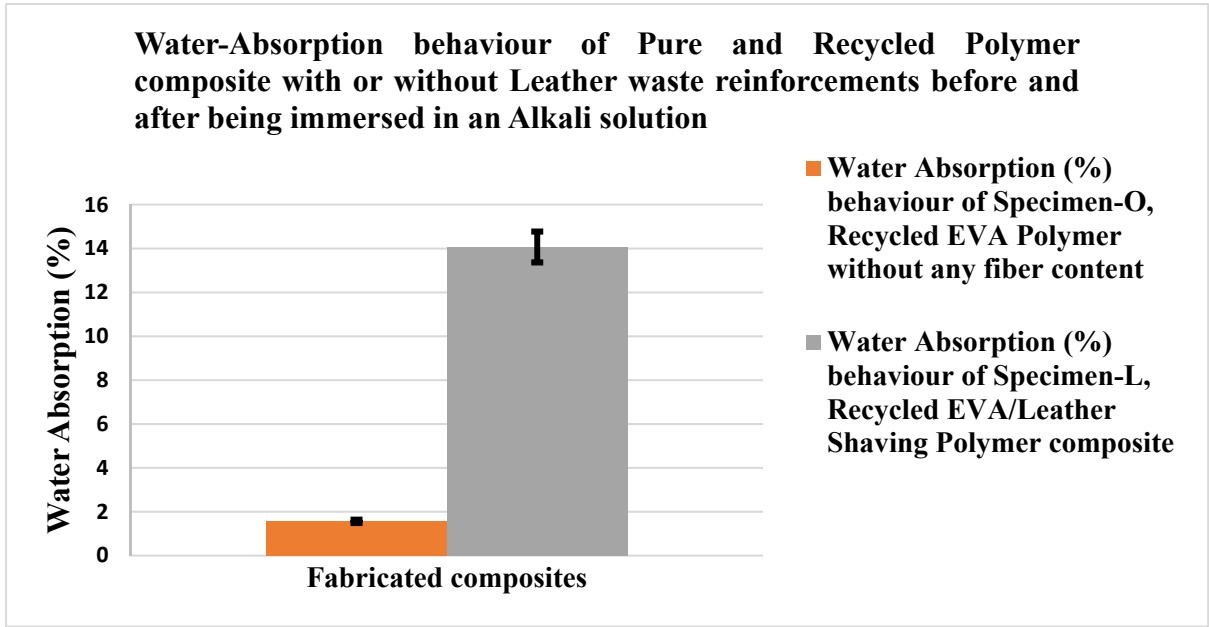

**Figure 11.** Comparison of "water-absorption (%)" for "neat-recycled EVA", and "recycled-EVA polymer composites".

Given that the leather fibers are hydrophilic by nature, the composite rises in water absorption as the percentage of leather shavings loading has been escalated to 1:1. In the 'footwear' and 'apparel' industries, this property has significant commercial significance.

Although a larger percentage of water uptake is observed in composites developed of chrome-tanned wastes and HDPE, the percentage of water uptake increases when the number of chrome-tanned wastes grows dramatically [66]. This suggested that certain additives, such as zinc octadecanoate and octadecanoic acid, were, in fact, responsible for the development of additional microgaps, abnormalities, irregularities, and other flaws in composite materials. This might also take into account the reduced tensile-modulus that Musa et al. (2017) [66] reported.

### 10.3. Thermal Studies

#### 10.3.1. Thermo-Gravimetric Analysis (TGA)

The primary elements of leather filaments are "collagenous protein, which is composed of long amino acid chains, chromic oxide, and other biological substances. To maintain chemical stability, the right surface texture, and resistance to deterioration/decomposition by fungus and micro-organisms/microbes during usage, these substances are added to leather". TGA and DSC were used, respectively, to evaluate the thermal properties of the composites. A minimal weight loss of the sample is defined as the temperature in a TGA thermogram that corresponds to a 5% weight loss.

"The lowest weight loss of all the samples was found to be in the 211–289 °C range, indicating that the generated composites are quite thermally stable, at least up to 211 °C".

Figure 12a,b illustrates the TG and DTG curves at distinct heating-rates of the leather-shavings in the $N_2$ environment, respectively. These graphs exhibit the characteristic form pattern of leather specimens: three-phases could be identified throughout the heating-phase (correlating to areas 1–3 as indicated in Figure 12a). As per the TG-FTIR findings, the very first phase (from 25–150 °C) is predominantly attributable to the emission of water as well as low–molecular–weight volatile chemical byproducts. The second phase ranges from 150–500 °C. This period is characterized by considerable weight-loss (around 50 percent of overall volatile-matter), which correlates to the primary breakdown-phase of collagen-fiber particles [16,21,26,35,37,39,46,47]. There seem to be two shoulders, one of these is noticeable while the other one is feeble, which might well be detected on 'DTG curvatures' in this area, which may have been induced by the collagenous with varying extent magnitudes of cross-linking [16,21,26,35,37,39,46,47]. During the last phase (500–600 °C), the weight-loss of leather-shavings steadily escalated with rising temperature, and the slow yet persistent degradation of the carbon-containing compounds in residuals could be attributed to this phase.

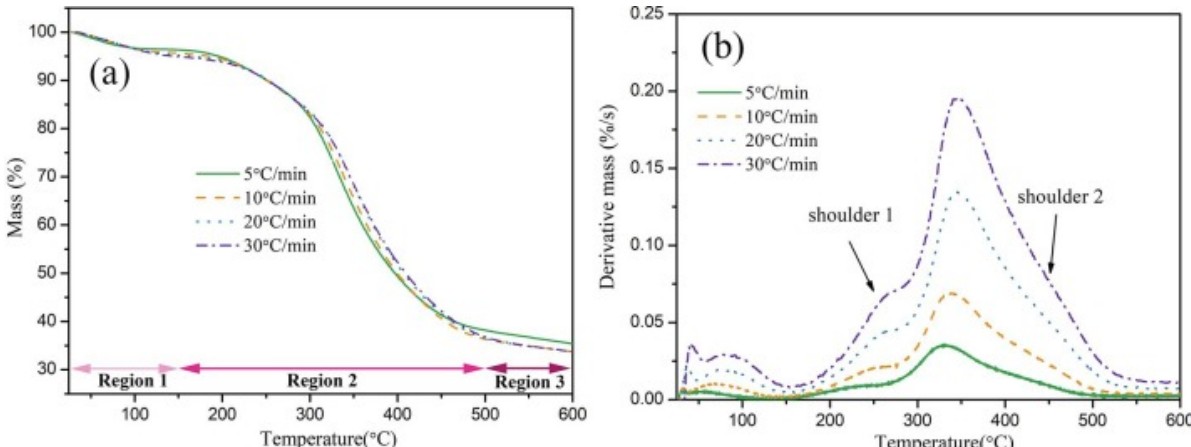

**Figure 12.** (**a**) TG of leather-shavings and (**b**) DTG curves of leather-shavings Adapted from the reference [77].

According to TGA research, the weight of the specimen remains constant up to 213.47 °C, after which there is a large weight loss (99.09318 percent) for the leather particles for "neat recycled-EVA" polymer composite specimens. In other words, 99.09 percent, which also shows that the specimen is 'thermally stable' up to 213.47 °C and begins to lose weight (95.651512 percent) after this temperature for the leather shavings/recycled EVA Polymer combination. 95.65 percent, which is also evidence that the specimen is 'thermally stable' up to 213.81 °C, as shown in Figure 13a,b.

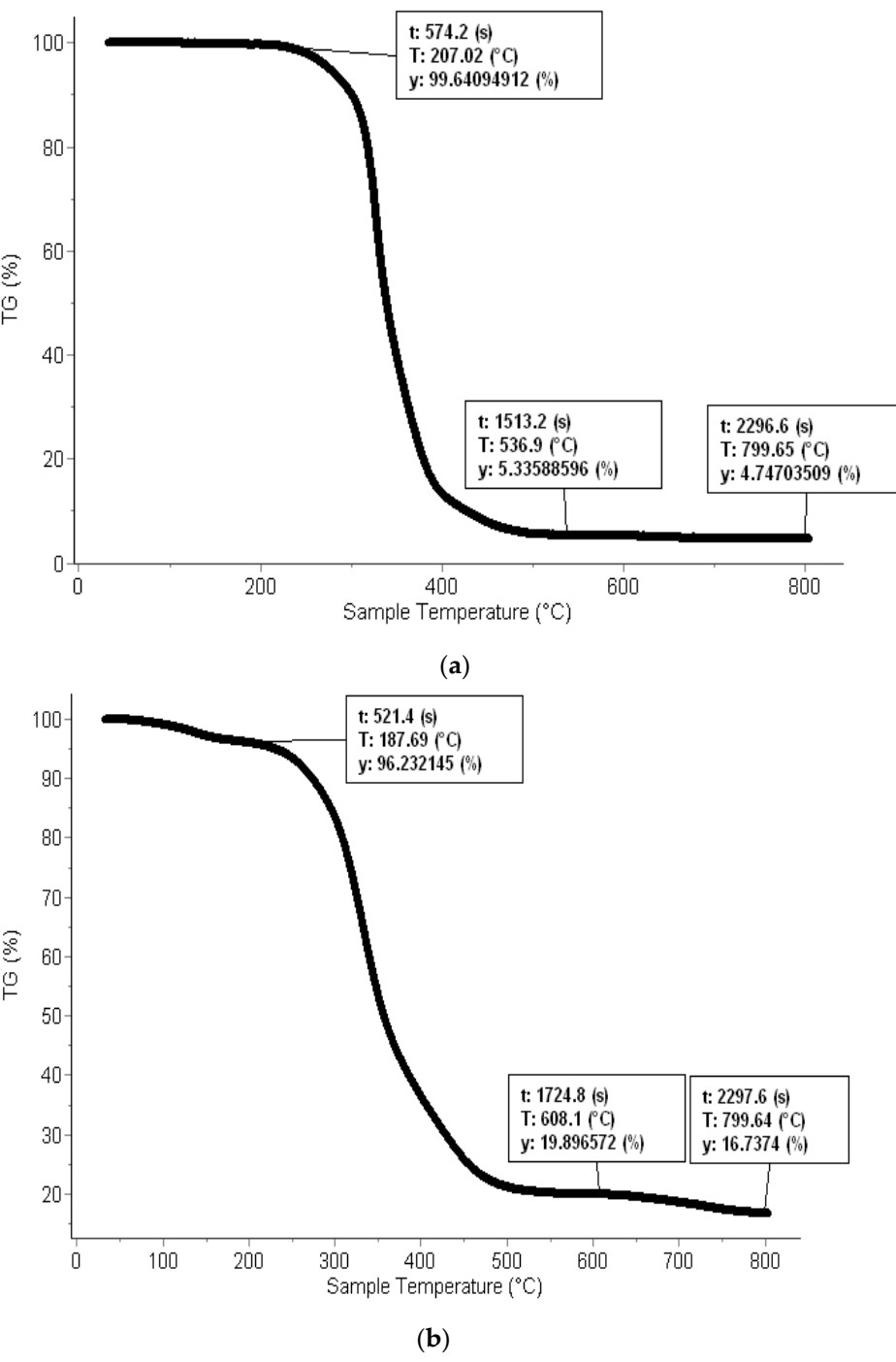

**Figure 13.** TGA curve of the (**a**) 'neat recycled-EVA polymer' without any 'leather-fibers' Adapted from the reference [45], and (**b**) 'Leather-shavings'/'recycled-EVA' polymer composites.

Acetic acid is released during the first stage of mass loss. "Acetic acid has already been completely discharged in the second mass-loss step. The first weight loss (213.47 °C to

300 °C for neat recycled-EVA and 213.81 °C to 300 °C for Leather shavings/recycled EVA polymer composites) is a result of the formation of C=C all throughout the core-polymeric framework structure and has been linked to similar findings" [16,21,26,35,37,39,46,47]. "The second mass loss occurred as a result of the oxidation and volatilization of hydrocarbon compounds as a result of the breakdown of the recycled EVA co-polymer network structure (300 °C to 460 °C for neat recycled EVA and 300 °C to 465 °C for leather shaving/recycled EVA polymer composites)".

The level one degradation relates to the deacylation of the vinyl acetate group with the removal of acetic acid, according to similar findings from in-depth literature. Additionally, level one degradation of vinyl polyethylene chains gives way to level two degradation. Rapid weight loss and a rise in temperature are also signs of the second stage of deterioration. The loss of water molecules is visible in the degradation in the region of 100 °C. According to Tegegn (2018) [48], there are three basic steps to weight loss caused by the heat degradation of collagen hydrolysate. Between 46.05 °C and 220.24 °C, the first stage was detected. There was a weight loss of 9.67%, which may have been caused by the gelatin's moisture evaporating. The degradation of proteins caused the second stage of weight loss, which started at a temperature between 220.2 and 350.98 °C, to lose the most weight (38.84 percent). This finding is marginally higher than that obtained by Camila de Campo et al. in 2017 [40], who reported that the beginning degradation temperature was seen at 200 °C. Collagen hydrolysate degradation began at 220.2 °C. The third stage of weight loss takes place between 350.98 and 431.35 °C in temperature. Using a temperature range of 431.4 to 587.06 °C and a weight reduction of 12.89 percent, the corresponding weight loss was (7.24 percent). At 587.06 °C, it has a residual mass of 30.45%. The greatest weight loss was recorded at 311.75 °C, or 0.766 percent/°C [48].

According to Ravichandran et al. (2005) [78], the structure of leather is bound to alter when leather particles are joined with scrap rubber and solidified at high temperatures under pressure. Additionally, the degradation of the recycled rubber matrix is expected to be accelerated by the leather's breakdown products, particularly trivalent chromium. There has been much discussion over the deterioration of leather-containing scrap rubber vulcanizates [78]. Additionally, the decomposition products were listed in their investigations along with the degradation studies employing thermal gravimetric analysis of the vulcanizates under inert and oxidizing atmospheric conditions.

Additionally, as reported by Ravichandran et al. (2005) [78], "TGA of leather samples were tested between 0 °C and 400 °C in nitrogen environment in order to understand the role of leather on the thermal stability of natural rubber vulcanizates. Two stages of leather degradation below 400 °C were visible in the thermograms of treated and untreated leather. The first stage of weight loss, which takes place below 100 °C, is attributed to released water, while the second stage, which happens between 200 and 400 °C for all samples, is assigned to leather degradation" [78]. Since the temperature at which "all leather samples decomposed was almost the same (between 200 and 300 °C), it was reasonable to assume that they would all have a similar impact on the thermal stability of the parent natural rubber-scrap rubber vulcanisates". The natural rubber vulcanizates' breakdown is influenced by the pieces created during this stage's decomposition [78]. "The TGA of natural rubber vulcanizates was carried out between 50 and 400 °C in a nitrogen environment to examine the effects of untreated, sodium-hydrogen-carbonate processed, azane processed, and carbamide processed leather parts on the 'thermal stability' of latex-reclaimed rubber-vulcanizates [78]". The results of the relevant thermograms better illustrated the thermal stability of latex-reclaimed rubber-vulcanizates up to 350 °C without leather. Although the disintegration patterns for the vulcanizates, including untreated and treated leather, are similar, as shown by the thermograms, a decrease in their 'thermally stable' can be seen when compared to the vulcanizate in the absence of leather. Therefore, it may be inferred that natural rubber vulcanizates without leather are more 'thermally stable' than those that do. The % weight was measured in steps of 50 °C for all the samples in order to compare the weight loss between 50 and 400 °C for the untreated and treated

natural leather rubber-scrap rubber vulcanizates. For all of the samples, the TGA results showed how much weight is retained for every 50 °C increase in temperature. The data comparison showed that the latex-reclaimed rubber-vulcanizates, which contain untreated leather up to 350 °C, retained more than 92 percent of their weight". However, the data for all the "latex-reclaimed rubber-vulcanizates" that made up the processed leather samples revealed that merely at 300 °C were around 92 percent of the weight retain. The 'thermally stability' of "treated leather natural rubber-scarp rubber vulcanizates" showed a 50 °C temperature decrease, amply proving the perspective [78].

"The weight loss at 400 °C under isothermal conditions was also observed for 46 min in increments of four minutes in order to compare the rates of deterioration for all the samples. The measured weights for all the samples under isothermal conditions at 400 °C, had revealed that there was a slight increase in the rate of degradation in the former compared to the latter, despite the fact that the rate of degradation for latex-reclaimed rubber-vulcanizates containing untreated leather appears to be similar to that for vulcanizates containing treated leather". Additionally, it was obvious that "latex-reclaimed rubber-vulcanizates" without leather demonstrate a rapid weight loss at a temperature that is significantly greater than those of "latex-reclaimed rubber-vulcanizates" that contain 'leather-particulates', as reported by Ravichandran et al. 2005 [78]. At 400 °C, under 'isothermal conditions', a weight loss of around 50% was seen for all the samples [78].

The results of reclaimed rubber-only (without leather) TGA analyses of natural-rubber (NR) vulcanizates were conducted between 0 and 600 °C in the air [78]. The investigation is being done to determine how untreated leather affects the thermostability of vulcanizates made from natural rubber and recycled rubber. There was no weight loss below 150 °C. Hence the material does not appear to contain any water in the matrix. The in-situ FTIR is applied to the TGA analysis effluents for vulcanizates that do not contain leather at various time intervals [78]. The four spectra resemble their corresponding untreated leather-loaded sample spectra almost perfectly. The thermogravimetric analysis (TGA) of latex-reclaimed rubber-vulcanizates, including untreated leather particles, which was conducted between 0 °C and 600 °C in air, produced the following results.

Previous research revealed that the thermogram was there even though the present case's pace of rubber matrix disintegration and the beginning of water release from the rubber matrix looked to be different [78]. "The rate of degradation is lower for the vulcanizate without any leather than for the loaded sample with untreated leather. Additionally, the thermogram shows the last residue of the leather-loaded sample below 500 °C. Its untreated leather may be to blame for this variance in behaviour. According to Ravichandran et al. (2005) [78] the structural water present in leather that takes time to get out of leather due to the likely structural changes in leather may be to blame for the time-dependent water loss that was shown to occur up to 200 °C".

Due to the loss of poorly adsorbed water, the TGA trace has shown that weight loss might start even below 100 °C [78]. But the material immediately starts to decompose. Between 220–350 °C, there is a very slight weight loss (around 18%), followed by two further stages between 350 and 500 °C. In this temperature range, the overall weight loss was equivalent to about 42%. Starting at 500 °C, the final residue begins to lose weight extremely quickly. Given that the TGA trace does not completely disappear, there must be a thermally stable residue, most likely inorganic chromium oxide, as chromium is a frequent element found in leather [78].

According to the TGA data for leather vulcanizates treated with sodium bicarbonate, "water is continuously released below 200 °C, and the vulcanizates begin to decompose at 240 °C. From this, it can be inferred that the effect of sodium bicarbonate-treated leather vulcanizates on the thermal properties of "latex-reclaimed rubber matrix" would be comparable to that of untreated leather vulcanisates" [78].

The TGA results of azane-processed leather vulcanizates were comparable to those of urea-treated leather vulcanizates [78]. The discharged water remained unchanged, and the rubber matrix's breakdown had already begun. At 500 °C, the residue underwent

two stages of disintegration. This finding demonstrated that the degradation of "latex-reclaimed rubber-vulcanizates" including leather that had been azane-processed, would behave almost identically to leather that had been urea-treated [78].

The TGA results of the composites made from urea-treated leather showed similarities between them and the composites made from raw leather, but there were also clear discrepancies [78]. The effect of leather treated with urea on the thermal durability of a natural rubber-scrap rubber matrix was amply proven by this material's two-stage disintegration of the residue at temperatures of 500 and 540 °C. "The release of water was evident below 200 °C, and the rubber matrix started to break down at 240 °C, as with composites consisting of raw leather. The second stage decomposition of the vulcanised containing carbamide processed leather at 540 °C may be caused by the strong mixing and interaction between the matrix and leather that leads to a co-operative deterioration that can only occur at a high temperature" [78].

### 10.3.2. Differential Scanning Calorimetry (DSC)

According to research, a glass transition is, in fact, a second-order endothermic transformation that manifests as a gradual changeover shift in consecutive DSC thermogram heating graphs. At this moment, changes in chain mobility have caused a polymeric transition's physical and mechanical properties to change from elastic to brittle. It was found that the $T_g$ values for the "neat recycled-EVA" were around –20 °C and were unaffected by the cross-link density or vinyl acetate concentration. As shown in Figure 14a,b, the $T_g$ of leather shavings/recycled EVA polymer composites was discovered to be approximately −16 °C.

Exothermic crystallization causes a significant exothermic peak to form during "the quick cooling process at around 215 °C (for neat recycled EVA polymer composites) and 218 °C (for leather shavings/recycled EVA polymer composites)".

An 'endothermic peak maximal' on 'DSC heating graphs' could be viewed as a melting-temperature (Tm), a "first-order transformation". The Tm of 'neat recycled-EVA' was determined to be approximately 88.5 °C. At the same time, the Tm of the blend of recycled EVA polymer and leather shaving composites was discovered to be about 118 °C.

The specimens with higher water content and more volatile chemicals in the gradually progressing heating than the control-board could be the cause of the lower endothermic transitioning result. Comparing "neat recycled-EVA" polymer composites to leather shavings, the neat composites had greater endothermic values (Tm1) than the latter. This might be a result of the fact that all the fibers were spread very equally throughout the leather matrix, which caused their melting phase to shift toward a higher temperature. Due to the structural heterogeneity of leather, which includes remnants of tanned components that may interact with fibrous filaments, the second and third 'endothermic peak values' of such a composite are also converted to exceptionally high-temperatures.

Regardless of the presence or absence of leather, the polymeric chain's arrangement loses its stability and homogeneity. The addition of a plasticizer inside of polymers increases the chain motility, which in turn affects the $T_g$ of the polymer.

The DSC analysis also revealed that "solid leather fibers lose water at an endothermic transition around 100 °C, are thermostable up to 211 °C, and start to break down collagen around 332.56 °C for 'neat recycled-EVA' samples and 318.47 °C for leather shavings/recycled EVA polymer composite samples, respectively. The glass transition temperature ($T_g$), which was measured for the developed composites and was found to be between −16 and 30 °C, is a valuable feature for understanding the processing parameters as indicated by DSC". Between 325 and 500 °C, during the leather preparation process, a significant mass loss developed as a result of protein loss and the calcination of a material. "This mass loss persisted between 130 and 150 °C and was attributed to unstable volatile compounds such as lubricants (oils) and low molecular-weight greases observed throughout the leather fibrils".

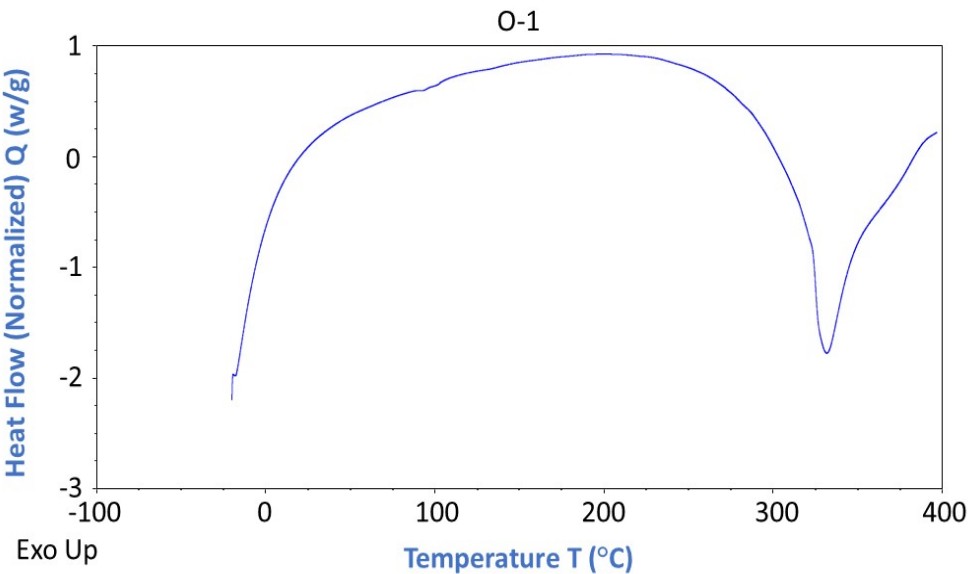

TA Instruments Trios V4.3.1.39215

(**a**)

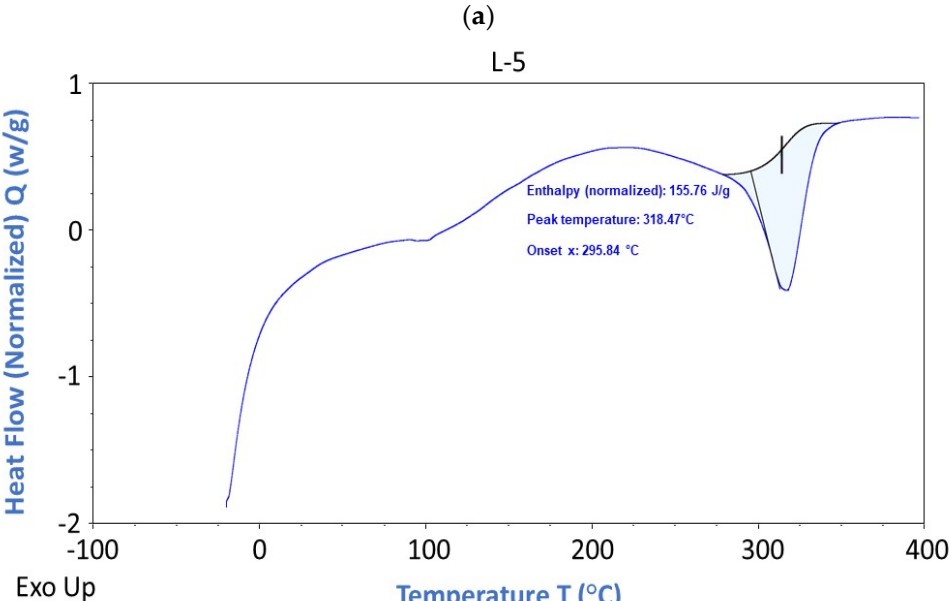

TA Instruments Trios V4.3.1.39215

(**b**)

**Figure 14.** "DSC thermograms for (**a**) Neat recycled EVA polymer without any leather fibers Adapted from the reference [45], and (**b**) Leather shavings/recycled EVA polymer composites".

According to Joseph et al. (2017) [41], the leather waste was fully amorphous in nature because "there was no melting peak for the pure leather sample according to comparable results for the DSC melting thermograms". A significant melting peak for the pure poly-caprolactone (PCL) sample was noted at 57.14 °C. Other compositions that contained PCL and leather only showed one peak, which is indicative of PCL melting, and there was

no change in the peak's position. This outcome also demonstrated that PCL and leather wastes did not interact. With an increase in leather content, the peak's intensity drops proportionally. Leather has a very small impact on the 'melting-temperature' of composites. The 'melting-point' is between 56.6 °C and 58.2 °C.

The influence of 'leather content' on the temperature of composite crystallization was demonstrated using crystallization thermograms. The crystallization temperature was raised to 34 °C by adding leather waste to the PCL matrix by 2 °C. Additionally, because it will shorten the chilling process and consequently the entire cycle time, this trait can be advantageous for processing this mixture by 'extrusion' or 'injection molding' in final products [41].

"DSC analysis of latex-reclaimed rubber-vulcanizates without leather, untreated leather, and urea-treated leather samples is carried out between 0–400 °C, at a heat-rate of 20 °C/min, according to Ravichandran et al. (2005) [78]. DSC trace of latex-reclaimed rubber-vulcanizates without leather, untreated leather, and urea-treated leather samples". For natural rubber-reclaimed-rubber and vulcanizates without leather as well as for all the samples treated with leather, the modest endothermic elevations up to 200 °C are due to the release of retained water. An endothermic hump has been observed between 200 and 250 °C for vulcanizates that contain treated leather such as urea, sodium hydrogen carbonate, and azane, as well as for vulcanizates devoid of leather [78]. The endothermic chain dis-entanglement of the polymer chains is what causes this hump. In practically all of the samples, the subsequent endothermic hump between 200 and 300 °C can be attributed to little decomposition. The thermograms for 'leather' and "latex-reclaimed rubber-vulcanizates" that contained leather could not adequately resolve this decomposition [78].

"The DSC traces of all the samples showed a significant endothermic elevation over 300 °C, which appeared to perfectly correlate with the 'decomposition-stage' of "latex-reclaimed rubber-vulcanizates" contained with and without 'leather samples'. A thorough examination of the principal endotherm's origin for each sample has revealed that "natural rubber-scrap rubber vulcanizates" exhibit more stability in the absence of leather than 'latex-reclaimed rubber vulcanizates' do, as the former's endotherm origin is located above the latter [78].

## 11. Fourier-Transform Infrared-Spectroscopy (FT-IR)

As it is apparently observed in this monograph that surface-layer changes in 'recycled-EVA polymer' may be correlated with absorption regions C-H (3100–2914.88 cm$^{-1}$), which, under humid conditions, increase in intensity. As a result of hydrogen bonding between vinyl acetate and water, this increment occurs. The 'stretching vibration' of the 'C=O' is caused by the terminal "trans-vinylene double bond" (1701.87–1736.58 cm$^{-1}$), the 'C=C' and the "methylene stretch" (1432.85–1466.6 cm$^{-1}$), 'CH$_3$' (1371.14 cm$^{-1}$), and the 'C-O' (1296.89 cm$^{-1}$). In comparison to the "deformation bands" of the 'CH$_{2'}$ group (1464 cm$^{-1}$), the band corresponds to a stretch mode in which 'C-O-C' is ascribed stretch mode. The EVA with VA content in it can be monitored using this band as an integral standard. As demonstrated in Figure 15a, there is a predominant absorption band at 1739 cm$^{-1}$ for the acetate function.

These peaks are attributed to the EVA groups of alcohols, phenols, and carboxylic acids, which are primarily in a change of the material's strength behavior. "The spectra of the 'neat recycled-EVA' co-polymers show absorption peaks at approximately 2917.77 cm$^{-1}$ and 2869.56 cm$^{-1}$, which are connected to the asymmetric and symmetric stretching of the co-polymers' methylene group (-CH$_2$). The VA groups' distinctive absorption peaks are as follows: The stretching vibration of the -C=O band is attributed to 1732.73 cm$^{-1}$; the asymmetric stretching vibration of the C-O band is attributed to 1130.08–1104.05 cm$^{-1}$; the symmetric stretching vibration of the C-O-C band is attributed to 996.053 cm$^{-1}$; and the inner rocking vibration of the methylene group is attributed to 810.92 cm$^{-1}$. The contributions from both VA and ethylene (CH$_2$) units are substantially responsible for the observed absorption maxima at 1379.82 cm$^{-1}$. As shown in Table S5, it is expected that the

intensities of the absorption peaks at 810.92, 996.053, 1130.08–1104.05, 1379.82, 2869.56, and 2917.77 cm$^{-1}$ seem to rise as the VA concentration rises".

"The solid leather wastes and recycled EVA composites treated using a combination two-roll mill and hot press yielded the ATR-FTIR results". The recycled EVA was created when polyethylene and vinyl acetate reacted. The curve compares the spectra of recycled EVA composites that were treated using both a hot press and a two-roll mill.

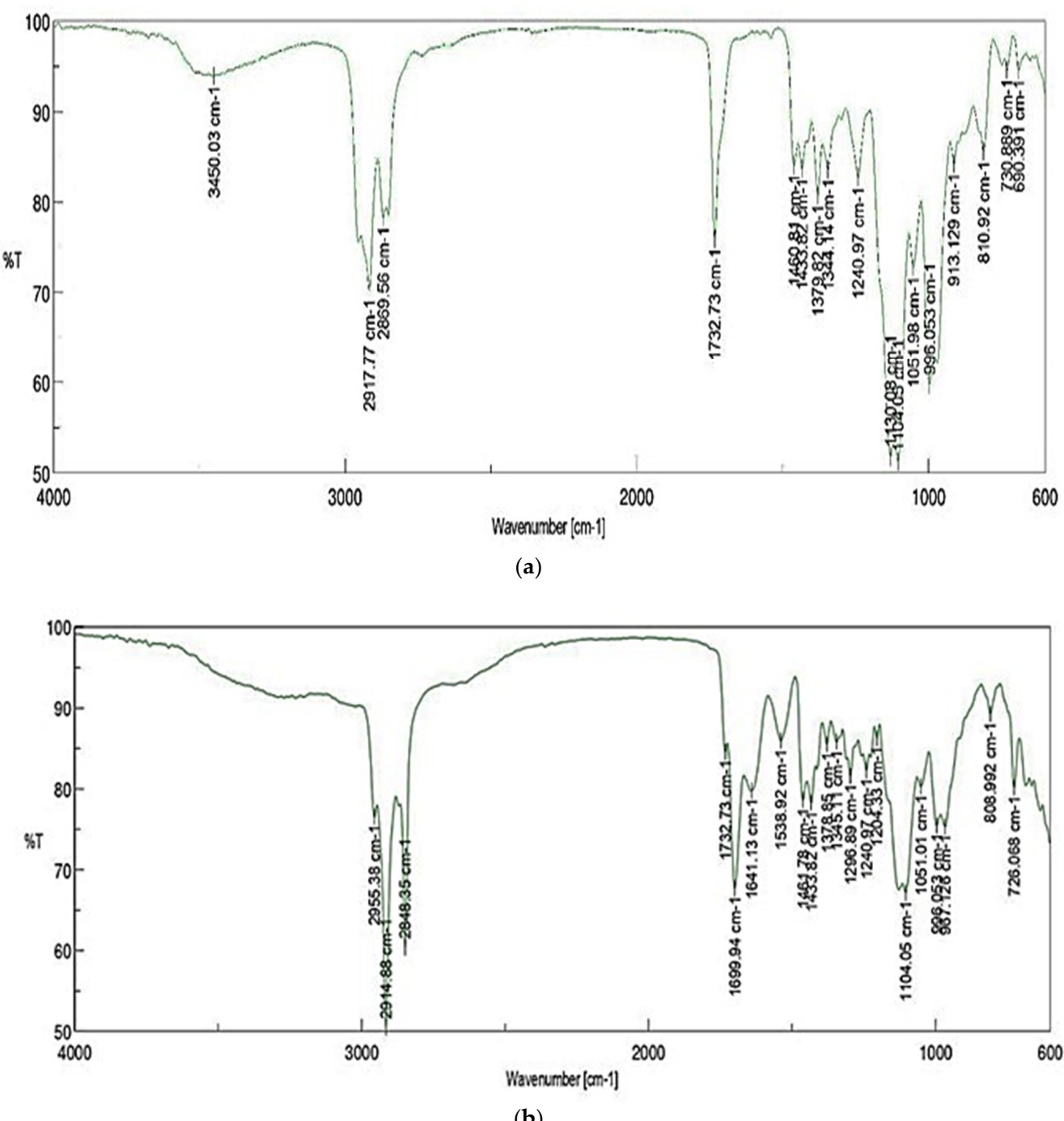

**Figure 15.** FT-IR spectra of (**a**) Neat recycled EVA polymer without any leather fibers Adapted from the reference [45], and (**b**) Leather shavings/recycled EVA polymer composites.

"While in ATR-FTIR monograph, molecular structure of Leather shavings/recycled EVA polymer composite are studied relative to the absorption regions O-H in side chains and terminal groups (3200.15–3500.40 cm$^{-1}$); (2914.88–2848.35 cm$^{-1}$) indicating the absence of fatty substances; A band of deformation vibration of first and second order amide (-NH) appears at 1641.13–1732.73 cm$^{-1}$; 1433.82–1461.78 cm$^{-1}$ is due to -COO$^{-}$ groups; 1345.11–1378.85 cm$^{-1}$ is due to -C-O- group; 1204.33–1296.89 cm$^{-1}$ is due to deformation vibration of -C=O group; 967.126–1051.01 cm$^{-1}$ is due to -C-O-C- ether group and below 726.068 to 600 cm$^{-1}$ is due to Cr-O bonds as demonstrated in the Figure 15b and Table S6".

As the comparable outcomes previously documented by Ambrósio et al. (2011) [38] have reported that "the co-polymer chains of the recycled EVA and of the plasticizers/additives do not lead to improvements in their molecular interaction geometries after being manufactured by the combined processing effect of two-roll mill and hot press". "The spectra of recycled-EVA and solid leather waste processed in a two-roll mill appear to be extremely similar".

According to previous research by 'Joseph et al. (2017)', the intensity of these peaks rises in leather-PCL composites as the wt.-composition of leather-fibers escalates. There are no other peaks in the leather-PCL composite other than the distinctive peaks for leather and PCL, confirming that there was no 'chemical interaction' between 'leather' and 'PCL', as had previously been validated by the DSC data examined by Joseph et al. (2017) [41].

Tegegn (2018) reported an analogue result for the FT-IR spectra of the "gelatin-starch cross-linked film" with 'KBr pellets' using the "Shimazu IR affinity-IS spectrometer" [75], which were used to analyze chemical properties. "FTIR analysis of a cross-linked composite film made of gelatin, starch, polyvinyl alcohol (PVA), and glucose from leather wastes revealed the following amide: A (3254.05 cm$^{-1}$) No discernible shift in the 'absorbance' of 'band amide A', it was (3252.12 cm$^{-1}$) associated with "NH stretching", and 'amide B' (2922.28 cm$^{-1}$) band inextricably correlated with "stretching vibration" of 'CH$_2$ bonds', it was (2920.35 cm$^{-1}$, 1645.35 cm$^{-1}$, 1543.12 cm$^{-1}$, and 1238.35 cm$^{-1}$ inextricably correlated with 'amide: I, II, and III', as well as 'amide I' associated with 'C=O stretching'. As a result of "CN stretching" and "NH bonding", a slight shift is seen in 'amide: I' (1649.21 cm$^{-1}$ to 1645.35 cm$^{-1}$) and in 'amide: II' and 'III' (1649.21 cm$^{-1}$ to 1645.35 cm$^{-1}$). A minor change occurred in 'amide III' (1238.35 cm$^{-1}$ to 1242.21 cm$^{-1}$), and a comparable alteration occurred as a result of the interaction between cross-linked leather-waste reinforced composite polymers" [79]. With 'CH$_2$ bending', and 'C-O-C stretching', respectively, there are bands at 1411.95 cm$^{-1}$ and 1097.54 cm$^{-1}$ [40].

According to Ravichandran et al. (2005) [78], "in-situ FTIR analysis of the breakdown products for 'vulcanizates' including "untreated leather" was performed between 500 and 4000 cm$^{-1}$ at varied time intervals in order to understand the "nature of decomposition". The IR spectra of the breakdown products recorded between 0 and 23 min were illustrated using the spectra acquired for various time intervals. The thermal-property -based investigation has revealed the emission of "hydrocarbon-waste fragments" by a very lesser-intense peak slightly below 3000 cm$^{-1}$.

Water was partially released at 3500 cm$^{-1}$ due to -OH stretch, and CO$_2$ was obviously released at 2354.5 cm$^{-1}$ due to its 'stretching vibration'. At 666.7 cm$^{-1}$, its 'bending mode' was also visible. These findings have unequivocally proven that the materials degraded through combustion throughout the specified time period. The IR spectra showed more intense peaks for the products released between 24 and 46 min, further demonstrating the release of more fragments and oxidized products, including CO$_2$ and water [78]. It could be explained by increasing activation as the temperature escalates.

The IR spectra of the released-products between 46 and 52 min have shown that the peaks caused by the aforementioned fragments were significantly more intense than those between 24 and 46 min [78]. Additionally, the maxima of the characteristic curves for water and CO$_2$ are right above 3500 cm$^{-1}$ and 2354. 3 cm$^{-1}$, respectively, were very clear indicators of the discharge of these gases. Once more, the high temperature may be to inflict such dramatic peaks since more oxidation may occur [78].

The IR spectra caused by the released products between 52 and 60 min revealed a decrease in the intensity of $CO_2$ vibration at 2354.3 $cm^{-1}$ due to the number of released hydrocarbon particles and water [78]. With this spectrum, it is important to note that less water was released, as evidenced by the decline in peak intensity above 3500 $cm^{-1}$. This observation has demonstrated that because the majority of the decomposition was already complete, the residue that has undergone decomposition at this specific stage may be of lower quantity.

According to Ravichandran et al. (2005) [78], "the in-situ FTIR spectra of the combustion products generated for vulcanizates containing sodium bicarbonate treated leather at different time intervals showed more release of $CO_2$ and water than organic particles". Between 46 and 52 min, there was an immense release of $CO_2$ and water; as a result, the temperature range associated with this decomposition may be more "combustion" and more rubber matrix. Between 52 and 60 min of combustion, there is also a decreased release of $CO_2$ and water, which is demonstrated by the lower level of cross-linked matrix residue [78].

The higher-volume of $CO_2$, hydrocarbons, and water was released when in-situ FTIR measurement of the combustion products for vulcanizates containing ammonia-treated leather was performed at various time intervals. The possible causes include a rise in temperature between periods of 30 and 50 min. The spectra also show that the residue produced when these combustion products were released in smaller quantities had greater thermal stability. However, the results have shown that more $CO_2$ and water have been released in some spectra and less volume of organic-fragments [78]. As a result, the residue needed additional thermal energy as the combustion process came to an end.

According to Ravichandran et al. (2005) [78], the FTIR analysis of the combustion products for vulcanizates comprising urea-treated leather over various time periods was demonstrated, showing the emission of $CO_2$, hydrocarbon fragments, and water. This was confirmed by inflections in the spectrum caused by "-OH stretch of water" just above 3500 $cm^{-1}$, "C-H stretch of organic-fragments" to just below 3000 $cm^{-1}$, and $CO_2$'s presence at 2354.3 $cm^{-1}$. The portion of the spectrum shows the well-dissolved sharper, more intense peaks for $CO_2$ at 2363.6 $cm^{-1}$, hydrocarbon at 2927.6 $cm^{-1}$, and water at 3600 $cm^{-1}$. The release of $CO_2$ and water is higher, while the release of organic-fragments is less, as seen by the other spectra. This could be a result of the cross-linked rubber matrix structure and leather residue coming together well [78].

## 12. Scanning Electron Microscopy with Energy-Dispersive Analysis of X-ray (SEM-EDAX)

The elemental composition and molecular dispersion of the entrapped solid leather fibers were revealed by EDX tests. "The SEM images have indicated that the particles are in the uniformly-dispersed and of spherical structure morphology, while E-DAX explores the stoichiometric-compounds and percent-chemical purity of the specimens to establish the occurrence/existence of chemical-elemental compositions".

As depicted in the micro-images, "recycled-EVA polymer" with leather-shaving fibers exhibits a structure that is characterized by the major elements ('Carbon', 'Sulphur', and 'Oxygen') as determined by EDAX analysis. According to Figure 16a, the surface of the 'neat recycled-EVA' matrix comprises C, S, Cl, O, and P. Saikia et al. (2017) [41] revealed similar results as well. "The incorporation of leather fibers and additives during the processing of combined-effect of rolling as well as hot-press compression molding has led to the removal of some chemical elements such as S, P, and Cl, according to the EDX of leather shavings/recycled EVA polymer composite surfaces" (Figure 16b) [35,37,39,41,46,47].

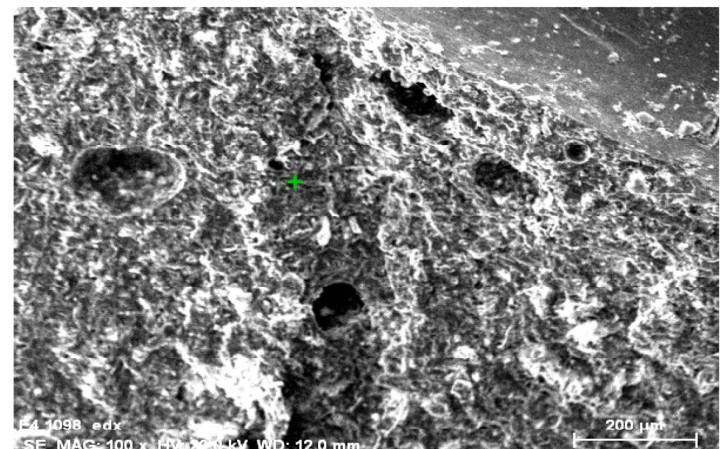

E4 1098Date:9/15/2020 3:55:00 PM Image size:600 x 450Mag:100xHV:20.0kV

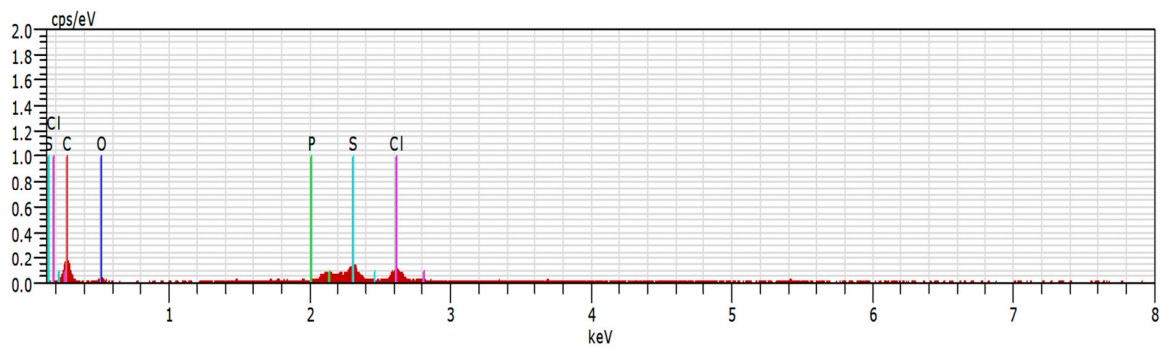

**Spectrum: O2 1235**     Date:9/15/2020 3:55:23 PM HV:20.0Kv     Puls     th.:0.23kcps

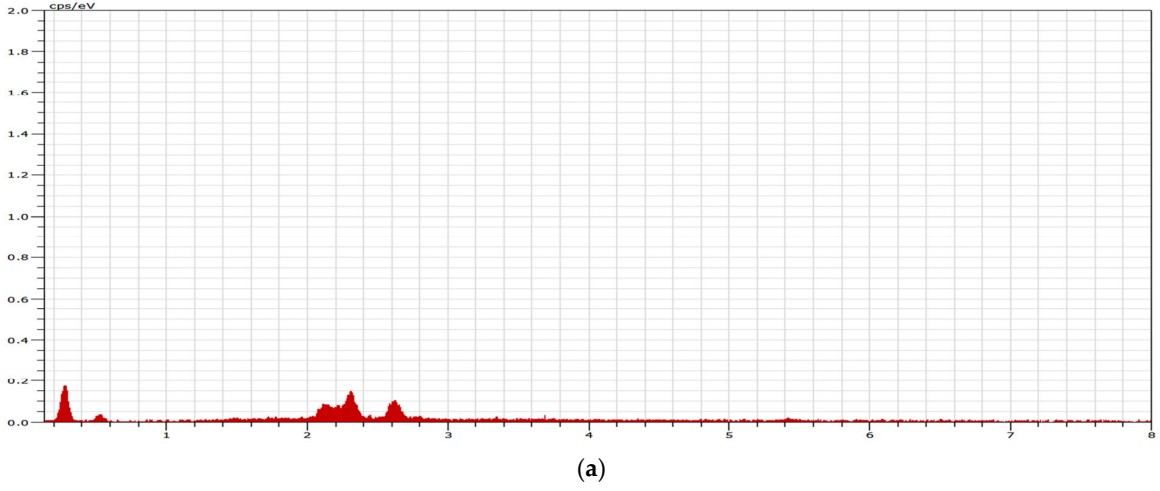

(**a**)

**Figure 16.** *Cont.*

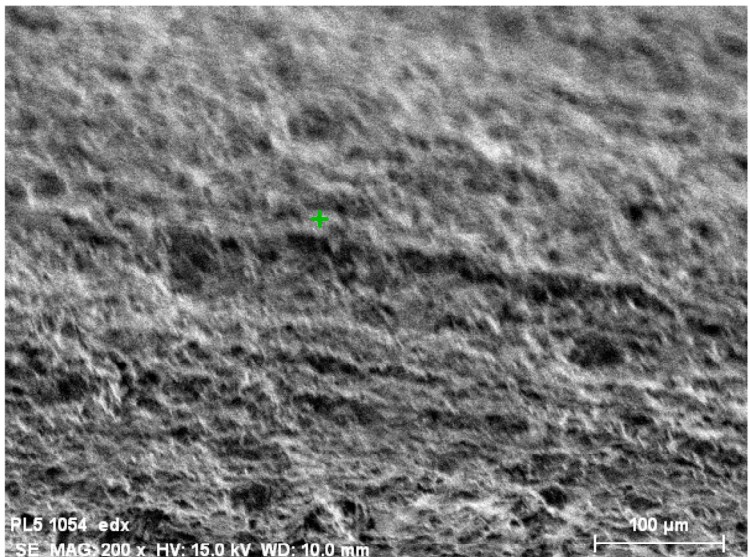

**PL5 1054Date:9/3/2020 4:38:39 AM Image size:600 x 450Mag:200xHV:15.0kV**

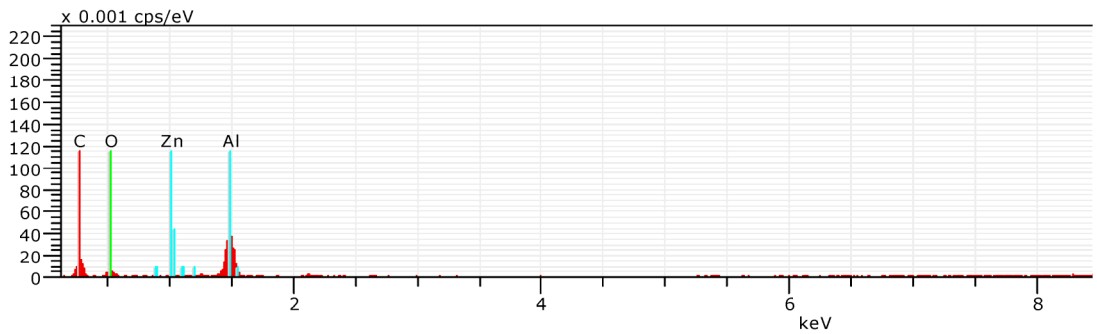

**Spectrum: L-3 1181 Date:9/3/2020 4:39:01 AM HV:15.0kV Puls th.:0.11kcps**

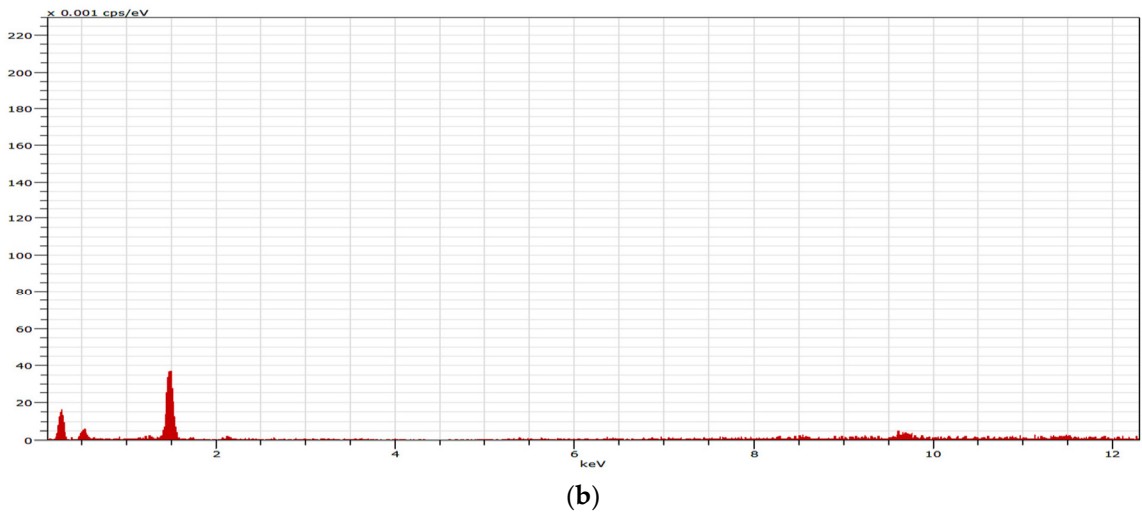

**(b)**

**Figure 16.** (**a**) SEM-EDAX analysis of the surface of neat recycled EVA polymer without any leather fibers Adapted from the reference [45]. (**b**) SEM-EDAX analysis of the surface of Leather shavings/recycled EVA polymer composites.

In the SEM-EDAX analysis, the occurrence of "trivalent chromium (0.1%)" in the leather shaving fibers, which have a "proteinaceous collagenous fibril-type threaded-form rugged structure", was identified. This is believed to result from the "tanning-process". The existence of "oxygen" can be attributed to the "collagenous in the leather-shaving fibers", which can serve as an "actively binding-site" for "trivalent chromium". These "binding-sites" typically include "carboxylate groups", which form "organic-salts" with "trivalent chromium". The appearance of "sulfur" and "chloride" is thought to be owing to the existence of "inorganic-salts" of "trivalent chromium". According to the EDX analysis of the "leather shavings/recycled EVA polymeric composites", the processing, which involved the inclusion of "leather fibers" and "additives", along with a combination of "rolling", and "hot-press compression molding", resulted in the eradication of certain "chemical elements" such as, "sulfur (S)", "phosphorus (P)", and "chloride (Cl)". SEM, as exhibited in Figure 16a, has illustrated that the recycled-EVA matrix typically reported the 'uniformity' or 'compatibility' of the "recycled-EVA matrix," revealing an excellent "interfacial bonding strength". The SEM, as exhibited in the Figure 16b, has illustrated the 'trace fragments' of recycled EVA on a layer of leather fabric aggregates, demonstrating that the recycled EVA is binding the agglomeration along with the fact that the recycled EVA and the leather fibers have a strong, reasonable interfacial bonding and adhesion strength. The potential for the interface to transmit matrices stress to the coalescence of leather fibers is shown because, in this case, the composites' cryogenically breaking or deformation/rupture would have occurred inside the coalescing amalgamates rather than through the interfacial-framework.

### 13. SEM Morphological Analysis

The micrographs of the 'neat recycled-EVA composites' are shown in Figure 17a,b, while the cross-sectional areas and microstructures of the leather shavings/recycled EVA composites are shown in Figures 18a,b and 19a,b, respectively. The recycled-EVA matrix typically contains an even distribution of the leather fiber 'agglomeration'. However, the 'uniformity' or 'compatibility' of the 'leather fibers' aggregates within the "recycled-EVA matrix" reveals an excellent "interfacial bonding strength". Some 'crevasses', 'openings', and 'voids' in the "surface morphology" are present and appear to be related to the 'de-fibrillation' and 'extraction' of the 'leather fiber aggregates'. Figure 17a,b indicates the locations of "interface-layer adhesion" or 'bonding' with 'interface consistency' or 'cohesion'. Due to the physicochemical differences between 'hydrophilic fibers' (made of "collagenous macro-molecules") and 'hydrophobic thermoplastic matrix', it is typically challenging to achieve strong 'compatibility', better 'bonding', excellent 'stability', and strong 'adherence binding' between "thermoplastic matrices" and "leather fibers". By adding stabilizers, additives, binding agents, or compatibilization agents to 'polypropylene blends' for finely ground wood, it is possible to increase the "interfacial bonding-strength", "biocompatibility", "miscibility", and produce "uniform", "contiguous surface areas" in 'thermoplastic composites' made of 'natural fabrics'. One such compatibilization agent has been utilized, named "cis-butenedioic anhydride" [38,78,80].

Figures 18a,b and 19a,b shows the micrographs of the recycled EVA/leather shavings composite with 50% leather fiber. In Figures 18a and 19a, an 'agglomeration' of leather fibers is 'encapsulated' within a recycled EVA skin. Such recycled EVA skin implies strong compatibility, interface adhesion, or close contact between the recycled EVA matrix and the leather fiber, including the possibility that the interphase may actually be firmer than the aggregation of leather fibers. The structural layout of leather fibers, including microfibrils of size less than 1 m, has been illustrated by the higher-magnification representation of a leather fabric thread-like, coalesces or 'agglomerate' in Figures 18b and 19b (symbolized via agglomerations throughout Figures 18a and 19a) [38].

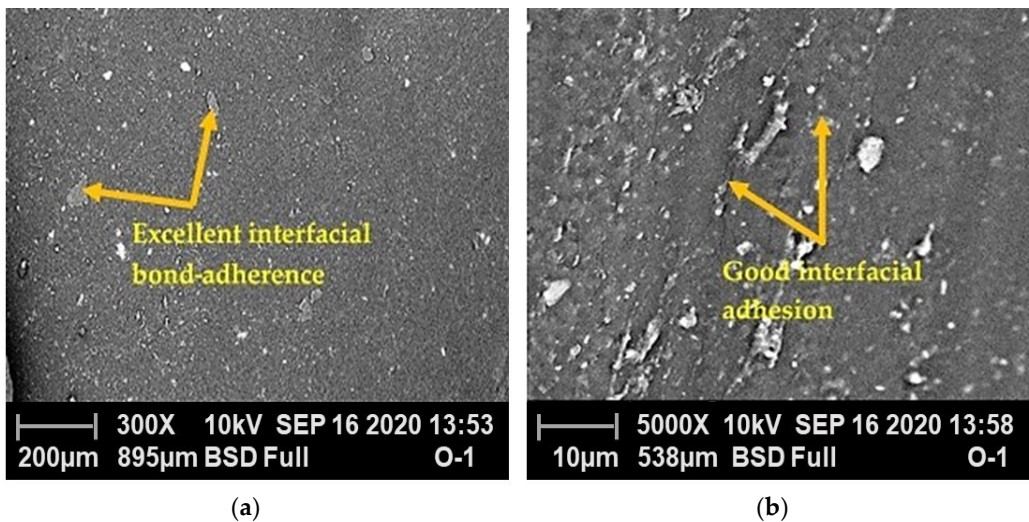

**Figure 17.** Micrographs of Neat recycled EVA polymer without leather fibers at a magnification of (**a**) 300× and; (**b**) 5000× Adapted from the reference [45].

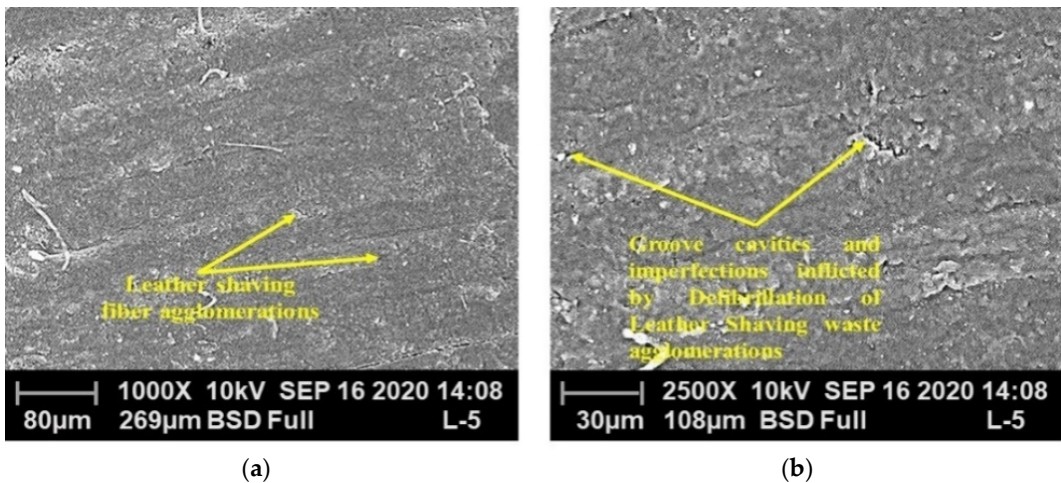

**Figure 18.** Micrographs of the surface of Leather shavings/recycled EVA polymer composites at a magnification of (**a**) 1000× and; (**b**) 2500×.

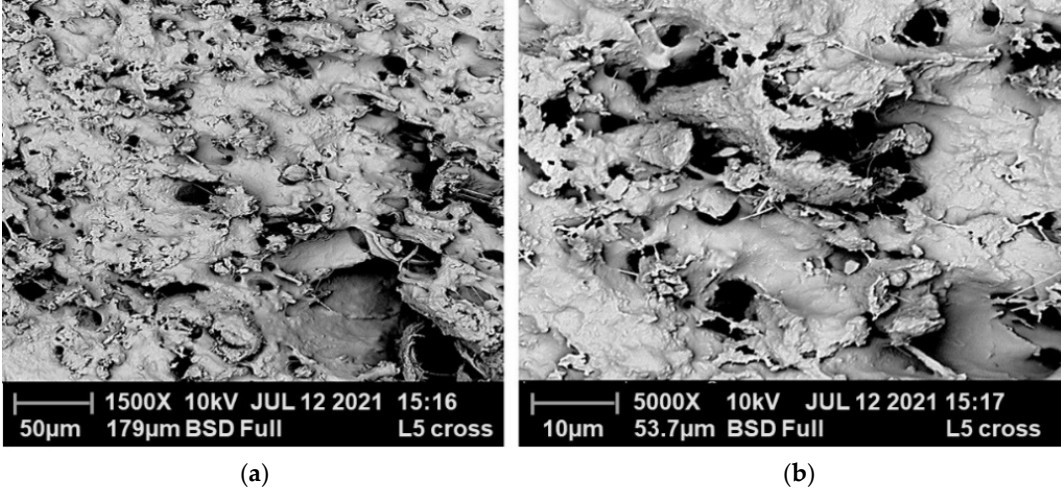

**Figure 19.** Micrographs of a cross-section of Leather shavings/recycled EVA polymer composites at a magnification of (**a**) 1500× and; (**b**) 5000×.

Figures 18b and 19b show 'trace fragments' of recycled EVA on a layer of leather fabric aggregates, demonstrating that the recycled EVA is binding the agglomeration along with the fact that the recycled EVA and the leather fibers have a strong, reasonable interfacial bonding and adhesion strength. The potential for the interface to transmit matrices stress to the coalescence of leather fibers is shown because, in this case, the composites' cryogenically breaking or deformation/rupture would have occurred inside the coalescing amalgamates rather than through the interfacial-framework.

As the proportion of leather-particles within recycled EVA matrix increases, as illustrated in Figures 18a and 19a, the leather fibrous aggregates seem to be reasonably close to and in contact with one another. If the leather particles were properly filled with recycled EVA matrix, the grooves, discontinuities, and voids in the microstructure seen in Figures 18a,b and 19a,b, which were reportedly generated by de-fibrillation and isolated/detached, would be eased. A twin-screw extruder can improve the blend formulation even though the blending employing the combined effects of two roll-milling and hot-press compression molding operations was successful. For dissipating and diffusing leather-fibrous agglomerations and then uniformly/homogeneously/spreading the leather shavings textiles throughout the recycled EVA matrix, a twin-screw extrusion or injector with conveying/kneading blocks are preferred.

The degree of fiber alignment, homogeneity of fiber dispersion, and degree of fiber elastomer adherence have all been studied using scanning electron microscopy [81,82]. Excellent fiber orientation in cellulose fiber rubber composites has been found through SEM analysis. Fibers are really close to being parallel [83]. The degree of fiber alignment and fiber-elastomer adhesion is found to be poor with 'nylon' and 'aramid fibers' among the 'short fibers' such as 'glass', 'carbon', 'aramid', and 'nylon', and 'cellulose' [84]. Both good dispersion and the good adherence of cellulose fibers to the elastomer matrix have been demonstrated. Specific interactions between the ingredients may be partially responsible for the increased adherence of cellulose fibers to elastomers. When used with polar polymers such as nitrile rubber and its blends with PVC, leather, a highly polar material, may likewise display comparable interactions. Additionally, it should be noted that the leather has a very high cohesive strength relative to its adhesive strength with the matrix.

The cut-section of the neat recycled EVA matrix is smooth and even on the surface, while the fibers of the leather shavings are evenly dispersed over the surface. However, in some areas, the binding, bonding, and compaction of the leather shavings to the recycled EVA matrix are insufficient. Contiguous split leather fibers are visible in smaller quantities; this could be the result of shear stresses being applied during rolling and hot-pressing of the composition, or it could be caused by twisting forces. More amalgamation, fragmentation, and trapping of leather fibers are observed as the volume of fibers rises. The penetration of recycled EVA matrix through entanglement, which is primarily attributed to the 'physical adherence' between matrix and fibers, is shown. As a result, the tensile strength has been enhanced with a maximum content of 1:1 in leather-fiber.

Incorporating leather shavings into recycled EVA composite has increased the amount of fibrillar of recycled EVA matrix that was collected, and it has also caused the volume of fibrillar-interaction structures to tend toward being higher in volume (1:1), as observed in micrographs taken at a magnification of 5000×. Additionally, the lower magnification at 300× demonstrates the uniform distribution and intricate structure of the leather shavings, which appear to be completely covered in fibrillar structures made from recycled EVA matrix. This shows that the fibers made from recycled EVA matrix and leather shavings have good binding and adherence.

Collagen strands are tightly woven together to form leather. Researchers have examined the degree of adhesion between leather shaving fibers and recycled EVA blends using scanning electron microscopy (SEM). The interface between the fibers and the matrix in recycled EVA polymer that has been blended with leather shaving fibers at high temperatures is greatly dispersed, according to the material's morphology. As the fibers are incorporated into the matrix at high temperatures, a substantial fusion of the fibers with the

matrix has been observed, revealing that the fibers are entirely covered by the matrix [85]. Even though leather shavings have been employed in this study in the form of particles, it is anticipated that they will form a significant portion of the recycled-EVA polymer.

The physical characteristics of leather shavings as they are formed in the fibrous form are shown in Figures 18a,b and 19a,b. The closely knitted fibrous leather is easily disseminated into an EVA matrix, especially when there is a significant amount of recycled-EVA polymer present.

According to Ravichandran et al. (2005) [67,78], the same result has been observed for the effect of untreated as well as neutralized leather particle morphology of NR vulcanizates included 500 parts of a scrap of the same particle size (200–300 μm). The leather appears to have existed as a co-continuous stage across the matrix with well-defined boundaries between the two phases in the morphology of scrap rubber vulcanizates, which contained untreated leather particles. This was a direct effect of the 'untreated leather particles' 'rigid' and 'closely-knitted structure'. Although leather is typically a firmly knit fibrous material, treated leather particles were discovered to have a loose-bounded structure [67,78]. Depending on the type of treatment, the treated leather particles appear to occur as a co-continuous stage in the rubber matrices with various degrees of interfacial adhesion [67,78].

According to Joseph et al. (2017) [41], the comparable same SEM micrograph findings showed that the leather fibers were evenly dispersed over the surface as well as a weak adhesion, bond strength, and permeability of shavings to the Poly-caprolactone-matrix. A confined, segregated, and isolated fiber may be seen at lower concentrations; this could be due to shear tension loads applied during the different composition formulations in the extrusion process. The above-mentioned 'leather-fibers' with enhanced composition have led to further fiber accumulation, amalgamation, and entrapment. The improvement in tensile strength was achieved by the PCL-matrix penetrating through entanglement, which was attributable to forming of a physical adhesion between the matrix and fibers [41,77,79–84].

Huang et al. have examined the comparable findings about relevance with existing materials' morphologies, thermostability, and multi-objective optimization of process/operating parameters. With the use of a melt mixing technique and a variety of HDI compounds, ref. [86] has developed PLA and PBS mixture-blends. Analysis of the impact of HDI composition on the morphology and distinctive properties of the mixture was the aim of the previous research. The investigators have noted from the SEM results that when the HDI contents have increased, the size of PLA and PBS has decreased. In addition, the PBS has contributed to accelerating the rate at which PLA crystallizes. "Additionally, because of improved interface bonding between PLA and PBS, the 'toughness' of newly produced composites' has greatly increased. A comprehensive overview on the use of various catalyst substances during the catalytic pyrolysis of waste materials including plastics (such as PP, PE, PS, and PVC) along with the waste from the petroleum industries" was published by Pan et al. in [87]. (petroleum mud). The authors investigated how to handle petroleum waste and 'plastic-wastes' by using molecular sieves, carbons, metal oxides, and M-series catalysts. In the review article, the mechanism, difficulties, and potential advancements of CP were also discussed. The authors came to the conclusion that more emphasis needs to be paid to systematic planning for the reuse of waste products from the petroleum and plastics industries. Metal microcapsules (MMCs) and polymer microcapsules (PMCs) used in epoxy-based composites were investigated by Sun et al. in terms of their tribological behavior. The investigators noted that under typical conditions, MMCs have more compressive strength, wear loss, and modulus than PMCs. In comparison to MMCs, PMCs had a lower friction coefficient. Additionally, the friction value was decreased in PMCs with an increase in microcapsule concentration. However, the MMCs exhibit the contrary findings. The impact of nano clay in the glass fiber-reinforced PMCs was assessed by Prabhu et al. [88]. Using the resin infusion technique under vacuum-conditions, the hybrid polyester-based PMCs were fabricated. The composition of the clay was changed from 1 to 55 by weight in increments of 1. Various common testing methods were used to investigate mechanical and machining behavior. The newly fabricated composites were

characterized using TEM, tensile testing, impact testing, SEM testing, and fracture testing. Drilling was used to carry-out the machining behavior. The investigators have found that as clay content was increased, tensile strength also escalated. A composite with a 3% of "clay-composition", in comparison with the others, possessed better mechanical properties. According to a microstructure study, this composite exhibits less delamination at a high feed rate. Additionally, hybrid composites had higher fracture toughness and strain energy than traditional composites. Using the hand lay-up technique, Gopinath et al. [89] have developed glass and jute fiber-reinforced vinyl ester polymer matrix composites. The authors have analyzed various 'deck-configurations', and the investigators have used ANSYS and FEM to validate the findings. The authors noticed that the "stiffness" performed better in the alteration of 'responses' when they were placed in the proper location and shape. The 'V' form stiffener deck layout and the 'V and U' shape combination produced better results than other configurations. Additionally, the web joint—where there was the least deflection and strain—benefitted from the addition of stiffener, producing more remarkable outcomes than other 'locations'. Guo et al. [90] fabricated HDPE-based polymer matric composites that had glass and wood fibers as reinforcements. When developing PMCs, the authors used the injection molding process. The "mechanical behavior", and "water-absorption capability" of newly formed composites' were tested. The same conditions were used to compare hybrid composites to wood-fiber reinforced composites, and the authors found that there has been an enhancement of 40% and 253% for the tensile strength and modulus, respectively. The limitations of greater water absorption by WPCs are fiber swelling and loss of interface bonding. Additionally, the mechanical characteristics of the PMCs, as compared to WPC, were greatly enhanced by the addition of glass fibers. The nHA-reinforced PEEK polymer matrix composites were developed by Ma et al. [91] using the 3D braiding-self retention-hot pressing method. The weight percentage of the nHA varied between 6.5 and 14.5 percent. The modulus, hardness, and strength attained in the PMC with 6.5 percent nHA were 8.3 GPa, 3.34 GPA, and 155.32 MPa, respectively. These values were significantly higher than those of the 'base-matrix', according to the researchers. A reduction of 23.6 percent was observed with the addition of 14.5 percent; however, the toughness was raised up to 54.9 percent with the addition of 6.5 percent HA. This decrease was owing to the poor interfacial bonding that the rise in concentrations of nHA has attained. Glass-fiber (GF), Kevlar-fiber (KF), and carbon-fiber (CF) reinforced nylon polymeric matrix composites were developed by Mei et al. [92], and their effects on the properties of the basic matrix were assessed. As a part of the fabrication process, the 'rings' and 'layers' of fiber were used. SEM and tensile testing were used to investigate the newly developed composite materials. The investigators unveiled that, in comparison with the others that the CF-reinforced composites had superior tensile strength (110 MPa) and modulus (3941 MPa). The concentration of fiber rings and layers was increased, which improved the mechanical behavior. Pan et al. [93] studied various methods for recycling 'plastic-wastes' in order to get rid of plastic waste. The researchers have also discussed the benefits and drawbacks of the techniques used to recycle and get rid of 'plastic-wastes'. The authors argued that in order to reduce air pollution and protect human life, it is necessary to develop innovative techniques for disposing of or recycling 'plastic-wastes'. The related outcomes have been revealed by the prior studies [59–61,65,94–128].

## 14. X-ray Diffraction Analysis (XRD)

An X-ray diffraction study was conducted to identify the chemical-constituents or phases involved throughout the crystal-structures of recycled EVA polymer composites.

Figure 20 exhibits the X-ray analysis findings for the peak position of the recycled polymer composite samples, respectively. The positional-angles, peaked-heights, peaked-widths at half-maximal (FWHM), atomic d-spacing, and relative-intensity of each specimen peaking have been presented. The results from Tables 1 and 2 have illustrated that the main noteworthy variation between the spectrum seems to be the crystallinity value of the neat-recycled EVA polymeric composite sample has been reported to be 7%, and the

amorphous value has been unveiled to 93%. In contrast the crystallinity value of the leather-shavings/recycled EVA Polymeric-composite sample has been reported to be 19.3%, and the amorphous value has been unveiled to 80.7%, which corresponds to orthorhombic crystalline-forms. This novel outcome furthermore shows that the crystallinity value of leather-loaded recycled EVA composites is maximum and the neat-recycled EVA composites are minimum, which eventually confirms that leather-filled recycled EVA composites are having more hardness value as compared to neat recycled EVA composite samples. Figure 20 displays the locations (2θ) of the recycled polymeric composite's spectra, with its inter-planar spacing and Miller-Indices. Among all samples, there was also a match for stearic-acid, zinc-stearate, and other additives.

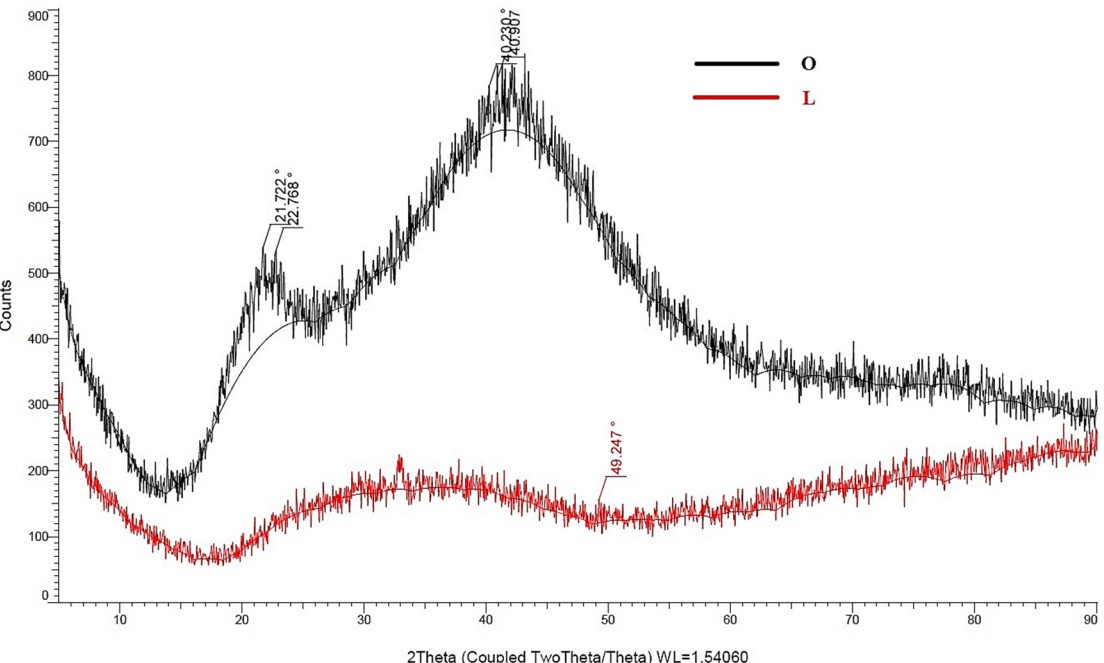

Where 'O' stands, "neat-recycled EVA polymeric composites", and 'L' refers "leather-shavings/recycled EVA Polymeric-composites".

**Figure 20.** XRD spectra-pattern of neat-recycled EVA polymeric composites and leather-shavings/recycled EVA Polymeric-composites.

The impact of leather wastes on the crystalline structure of recycled EVA polymer composites was explored utilizing XRD-Diffraction. The intense reflections observed in recycled EVA polymeric were seen to be in leather-wastes/recycled EVA Polymeric-composites. Chen and Wu reported comparable outcomes utilizing PCL-multiwalled carbon-nanotubes. Investigators discovered that incorporating MWCNT into PCL had no influence on the crystalline formation of PCL [83,84]. Jiang et al. found that the size of crystallites orthogonal to the (hkl) plane in PCL/silica composites rises progressively as the PCL wt.% rises. This phenomenon has been ascribed to insufficiently crystallized-macromolecules or tiny and metastable-crystalline that are firmly bound by rigid silica network-frameworks [83,84].

This novel outcome furthermore shows that the crystallinity value of leather-loaded recycled EVA composites is maximum and the neat-recycled EVA composites are minimum, which eventually confirms that leather-filled recycled EVA composites have more hardness value as compared to neat recycled EVA composite samples. The relative-intensity of a peak on another peak was determined and recorded versus the leather-waste-loading as depicted in Figure 20.

To recapitulate, the noisy lines observed from XRD Diffraction peaks are due to the presence of the extracellular fibrous protein collagenous in the leather fibers. The XRD

analysis of leather composite materials revealed distinct considerable diffraction noisy pattern-peaks caused by the prolonged-range ordered sequence of collagenous fibrillar molecules, with repetitive gap/overlap locations [118–124].

**Table 1.** XRD findings for chosen peak-regions with Net-intensity values, 2-theta values, relative-intensities, full-width half-maximum values (FWHM), and lattice-parameters (d).

| Neat-Recycled EVA Polymeric Composites | | | | | Leather-Shavings/Recycled EVA Polymeric-Composites | | | | |
|---|---|---|---|---|---|---|---|---|---|
| 2θ (°) | FWHM | Net Intensity | Relative Intensity | d Line-Spacing | 2θ (°) | FWHM | Net Intensity | Relative Intensity | d Line-Spacing |
| 21.722 | 1.032 | 145 | 100% | 4.08798 Å | | | | | |
| 22.768 | 0.903 | 122 | 84.2% | 3.90256 Å | 49.247 | 0.509 | 34.6 | 100% | 1.84877 Å |
| 40.230 | 0.774 | 71.3 | 49.2% | 2.23984 Å | | | | | |
| 40.907 | 1.011 | 77.5 | 53.5% | 2.20434 Å | | | | | |

**Table 2.** %age Crystallinity test for the fabricated polymeric composites.

| Samples | Crystallinity % Age | Amorphous % Age |
|---|---|---|
| Neat-recycled EVA polymeric composites | 7.0 | 93.0 |
| Leather-shavings/recycled EVA Polymeric-composites | 19.3 | 80.7 |

　　　Comparable findings have been unveiled in numerous studies. For instance, the XRD patterns of composites containing leather shavings and hybrid leather powder reveal that the crystalline structure collapsed due to the presence of $SiO_2$ particles in hybrid leather power-reinforced composites, which block the role between leather fibers in the ordered region. However, both composites possessed the triple helical intrinsic fibrous structure, and it is an indication of the use of ultrafine leather powder as a functional reinforcement [125]. The leather fibrous powder can also be effectively used as an adsorbent to remove dyes from the wastewater, and the X-ray diffraction analysis reveals that cow leather powders retain their protein structure [126]. A thin film X-ray diffraction analysis is also performed by researchers to examine the collagen fiber spacing and to identify the phases of isolated and commercial collagens [127]. The results of this study suggested the elimination of acetic acid residues from the isolated collagen for the purpose of a stable environment for the cells and their applications. X-ray diffraction patterns of Bacterial Cellulose/collagen hydrogel confirm the introduction of collagen in it and suggest a more amorphous structure than the Bacterial Cellulose alone [128]. When the XRD spectra of leather fibers with chromium-tanned, the intensity of Bragg peaks continuously increases. There seem to be two probable reasons for this: an enhancement in the overall long-order structural-arrangement of the collagenous fibrils and/or the inclusion of Cr(III) ions in the collagenous framework, which further strengthens the electron-density contrasts [119,121–124]. Thus accordingly, Maxwell et al., if the improvement in absorption diffraction peak-intensities in the XRD-spectrum of chromium-tanned leather composites was primarily exacerbated by the increased electronic-density contrasts induced by Cr(III), therefore the adhesion-binding of Cr(III) all along the length of the collagenous-network molecule must be considered to even affect the same absorption spectra intensity dispersal of the peaks similar to that of the untanned leather fibers. If collagen binding is restricted to specific amino acids, then these amino-acids. If collagen-binding interaction is restricted to specific amino-acid residues, therefore these organic-molecules should always be dispersed reasonably uniformly throughout the collagenous network molecule [119]. The amplitude of the maxima peak attributable to intermolecular lateral packing was seen to reduce as chromium content was increased. After the inclusion of chrome-content, the inter-molecular laterally pack distances dropped somewhat, and the full-width at half-maximum (FWHM) of such a peak-point increased, implying a significantly wider dispersion. The drop in diffraction

absorption peak seemed to be associated with the decline in inter-molecular pack-distance ordered arrangement [119].

Earlier studies have revealed the outcomes from a comparative assessment of previous XRD-outcomes. Similarly, Ammasi et al. [129] proposed a new method to reuse solid waste materials obtained from leather industries. The XRD patterns of untreated and polypeptide-treated leather samples illustrated the high intensity of the diffraction peaks. This type of pattern was obtained due to their highly e crystallinity. XRD patterns also indicated that the crystallinity of treated samples was better than the untreated ones. The noisiness in the patterns is also attributed to the alteration in the inter/intra molecular packing of the leather molecules. Cardona et al. [43] utilized leather waste materials in the natural rubber and characterized. The authors observed similar spectra of leather waste material and urea-treated leather waste materials. This type of formation is due to the addition of sodium carbide and sodium sulfate during leather tanning, which makes sure the interaction of chrome into collagen networks. Wang and Jin [130] synthesized waterborne polyurethane in the form of fillers, which were used in the development of synthetic leather. XRD patterns revealed that the micro-phase was separated (increasing intensity in XRD peaks). This is due to the enhancement in steric hindrance and hydrogen bonding of the ingredient molecules. Ma et al. [131] prepared chromium-tanned leather porous carbon samples using different mass ratios of alkali and carbon. The authors observed that as-prepared carbon exhibits broad diffraction peaks from 15° to 30° of theta angle due to amorphous crystalline structures. However, the pattern of BWLPC-2 indicated the presence of CrO at 36° and 64° theta angles. Also, with the introduction of KOH, diffraction peaks of activated carbon weakened continuously until they disappeared. Kanagaraj et al. [132] prepared protein-based material using trimming wastes; this material was then used in the chrome tanning process to increase the exhaustion level of the Cr. The increasing intensity in XRD patterns revealed enhanced exhaustion of the Cr contents. This improvement in exhaustion was due to the increasing interaction among the collagen sites and Cr contents. Ribeiro et al. [133] explored the utilization of tannery waste materials to resolve environmental issues. Authors treated leather saving on immobilizing chromium ions from the Portland cement matrix. XRD patterns revealed the loss of a large amount of organic material during ignition, and $Cr_2O_3$ is the inorganic phase that existed in the material, which is formed due to the oxidation of chromium. Dey et al. [134] introduced ZnO nano particulates in buffing dust to develop anti-bacterial characteristic footwear sols. The broad diffraction peak at $2\theta = 26°$ is observed in the XRD pattern of the ZnO-reinforced composite. This broadening nature was due to the elongation of cross-linked chains and the interaction of collagen with amorphous crystals that ensures the growth of ZnO particulates on the buffing dust. Li et al. [44] developed polyvinyl alcohol-based composite films, which were reinforced with leather fibers. Polyvinyl alcohol indicates strong peaks at $2\theta = 19.5°$ due to better crystalline performance and low-intensity peaks at 11.8° because of closer packing arrangement. Saikia et al. [135] used industrial leather waste to prepare composite products and evaluated their properties. Dye-trimmed waste materials, together with natural fibers (jute and cotton), were used during fabrication. The authors observed a composite containing a 1:1 blend ratio of leather waste and natural fiber waste possessed the best characteristic properties among others. XRD pattern of fabricated composites indicated weaker and broader peak intensities than base reinforcing materials due to structural-modifications in it and also due to interactions among the collagen and natural fibers during formulation.

## 15. Atomic Force Microscopy (AFM) Analysis

AFM has been utilized to determine the size (thickness, width, etc.) of the interphase and its stiffness relative to the bulk phase of neat and leather-shavings/recycled EVA polymer composites. AFM has been utilized to examine the AFM Roughness Analysis and AFM Grain Analysis of the fiber-reinforced with the leather-shavings/recycled EVA polymer composites. Phase imaging was focused on individual fibers to map the leather fibers/recycled EVA polymer composites interface and to see the fracture behavior (cracks etc.).

AFM has been utilized to explore the interfacial adhesion properties of Mechanical tested specimens of neat and leather-shavings/recycled EVA polymer composite surface, their surface topography mapping, phase-imaging analysis, and lateral forces of polymer composites that have leather-shavings.

AFM has been utilized to analyze the surface topography mapping, phase-image analysis, and lateral forces of neat and recycled EVA polymer composites that have leather-shavings as fibers.

AFM has been utilized to analyze the worn surface of the leather-shavings/recycled EVA polymer composites and to determine the elastic modulus characterization of neat polymer and leather-shavings/recycled EVA polymer composites. AFM phasing image analyzes variability in compositions/constituents, adherence, frictional, rheological characteristics, and certain other characteristics in addition to fundamental topographical map imaging. AFM has been employed to determine the surface structure, roughness of neat, and leather-shavings/recycled EVA polymer composites, leather fiber geometry, leather fiber diameter, filler distribution, leather fiber-dimensions and leather fiber-recycled EVA matrix adhesion in leather-shavings/recycled EVA polymer composites.

AFM analysis reveals a surface-layer of a neat recycled EVA and leather-shavings loaded recycled EVA polymeric-composites with several void-spaces, cavities, and significantly larger-depths, as well as peak-position ranges, whereas the inclusion of additives and lubricants, such as zinc octadecanoate and Octadecanoic acid into the recycled EVA polymeric matrix reveals a relatively uniformly smooth-surface with fewer void-spaces in material, but much more prominent peak was observed as showed in Figure 21.

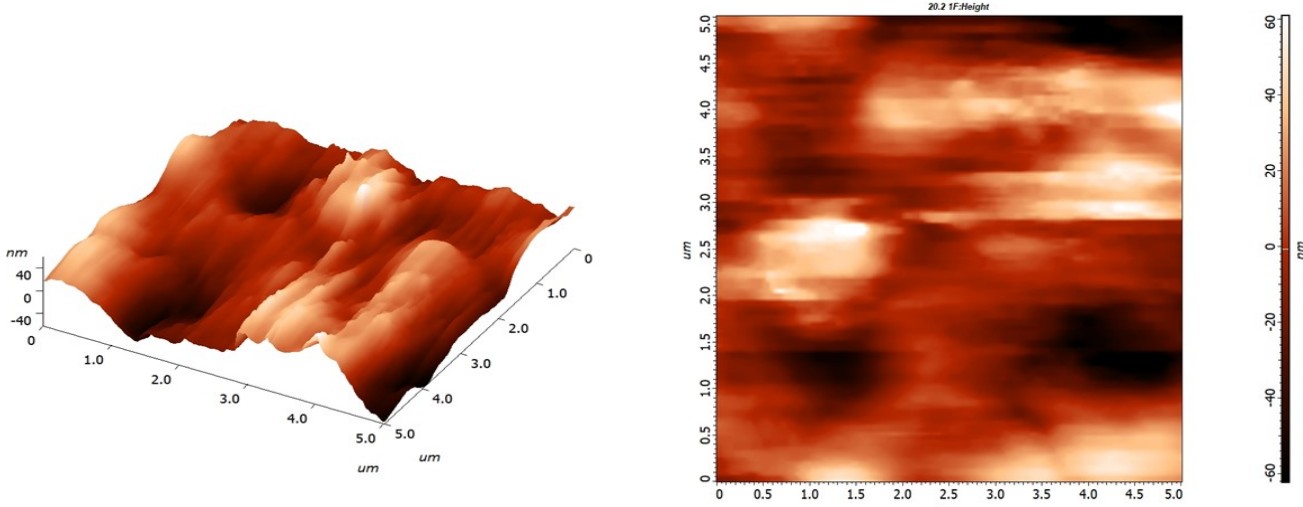

**Figure 21.** AFM topographic pictograph illustrating the surface and micro-hole profilometry for neat recycled EVA polymer composites.

Figure 22 shows that the surface-roughness values rise with the incorporation of Leather-shavings into the blend, indicating that the interfacial-contact among the leather-wastes, as well as the recycled EVA polymeric-matrix, is weaker, resulting from inadequate distribution and compliance. This revealed that leather-shavings, as well as the recycled EVA polymeric-matrix were compatible. It is claimed that the inclusion of additives, such as plasticizers and fillers, has certainly enhanced the interaction of leather-shaving wastes with recycled EVA polymeric-matrix and high interfacial surface-smoothness.

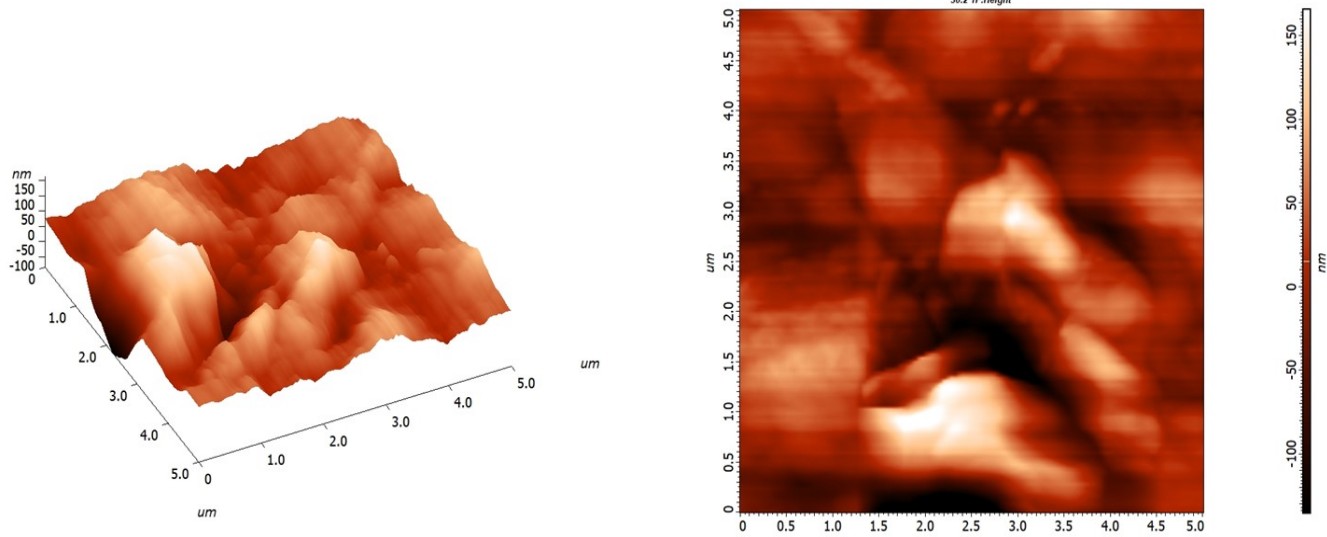

**Figure 22.** AFM topographic pictograph illustrating the surface and micro-hole profilometry for Leather shavings wastes/recycled EVA polymer composites.

## 16. Applications of the Developed Leather Waste/Recycled EVA Polymer Composites

As a result of developing sustainable functional or viable products from waste material, there will be a synergistic influence on the surrounding environment, in addition to producing 'value-added products' derived from "waste materials" accumulated in industries such as the 'Leather' and 'Polymer' sectors. The utilization of 'leather wastes', and 'recycled polymers' are likewise explored, which implies the widespread deployment of "leather wastes" technology for the manufacture of composites with enhanced characteristics for a diverse spectrum of application sectors, particularly the 'footwear industry', 'apparel accessories', 'automobile', 'construction', and 'industrial usages'.

Numerous multipurpose applications for the unique class of manufactured composites can be found in the 'footwear' and 'apparel sectors', 'composite sheets', and 'Leather-like personal-goods'. The 'floorings', 'personal safety products', applications for 'leather in decorative furniture', 'leather boards', and 'inexpensive adsorbents', 'Activated Carbon', 'energy generation' and 'energy recovery', 'adhesives in the woodworking industry', 'ceramic glaze pigments', 'production of biogas', 'Agriculture uses', 'biodegradable materials', and 'polymer films', 'Poultry feeds', 'cosmetics', 'biomedical uses', 'biocomposites', and the ability to 'microencapsulate drugs', as well as 'construction' and 'building materials', 'automotive interior trim molding components', 'thermal' and 'acoustical insulation-panels', 'cushioning boards', 'shoe soles', 'floor covering', and 'moldings' with remarkably exceptional 'physical properties', 'air permeance', and 'desirable aesthetics'.

## 17. Concluding Remarks

In this study, recycled EVA polymer matrices and the high-weight content of leather shavings with 1:1 were mixed with fabricating the flexible composite sheets through the combined effects of two-roll milling and hot-press compression molding with uniform blending. Observations based on thorough, detailed characterization investigations include the following:

i.      The physico-mechanical properties of "leather shavings/recycled EVA" composites exhibited to be significantly influenced by the leather-fibrous loading with a composition of 1:1. The tensile strength tends to rise slightly when the proportion of leather shavings in composites has been increased to 1:1. As the volume of leather fibers within the composites has increased, the modulus of elasticity of the composites has significantly improved. Leather shavings were discovered to have a stronger compressional deformation property in the recycled EVA matrix than the

      'neat-recycled EVA' matrix, which was reported to be higher by about 7.7%. The average peel strength between the polymer and leather (for leather shavings as fiber) in the recycled EVA matrix was determined to be approximately 0.9575 N/mm, according to the results obtained.

  ii.  The TGA investigation showed that 'recycled-EVA' polymer was thermostable up to 213.47 °C, whereas leather fibers showed no discernible major weight loss nearly comparable to 211 °C. According to the DSC results, the 'release of moisture' from the 'leather shavings' through an endothermic transition that occurs at about 100 °C is thermostable up to 211 °C and begins to decompose collagen at 332.56 °C for 'neat-recycled EVA' samples and 327.23 °C for "leather shavings/recycled EVA" polymer composite samples, respectively.

  iii.  In reference to the absorbance bands O-H (3100–2914.88 cm$^{-1}$) that increased in intensity under humid conditions due to hydrogen bonding interactions between the carbonyl group in vinyl acetate and water, the ATR-FTIR spectra have explored surface-interface-layer alterations and abnormalities in recycled EVA co-polymer. While the molecular structure of "leather shavings/recycled EVA" composites can be seen in the ATR-FTIR monograph as the band at 3314.07 cm$^{-1}$ corresponding to the vinyl alcohol -OH group; 2849.31–2916.81 cm$^{-1}$ is due to the -CH$_2$, -CH$_3$ groups inside chains and terminal groups; and below 723.175 to 600 cm$^{-1}$ is due to Cr-O bonds.

  iv.  Leather fibril particles are widely present as conglomerate aggregate clusters and are effectively distributed or interspersed throughout the recycled EVA matrices, according to the SEM surface morphology examination of the "leather shavings/recycled EVA" composites. The micrograph results show numerous interfaces with remarkable bonding strength and interfacial contact between the recycled EVA matrix and the leather shavings' residual particles.

  v.  According to "XRD-analysis", the "crystallinity value" of the 'neat-recycled EVA' polymeric composite sample is 7%, and the amorphous content is revealed to be 93%. While the crystallinity of the leather-shavings/recycled EVA polymeric composite sample was revealed to be 19.3% and the amorphous content to be 80.7 percent. This unexpected result has also demonstrated that the crystallinity value of the leather-filled recycled EVA composites is higher than that of the neat recycled EVA composite samples, which eventually confirms that the leather-filled recycled EVA composites have higher values for strength, modulus, and hardness.

  vi.  The inclusion of additives and lubricants into the recycled EVA polymeric matrix has revealed a relatively uniformly smooth-surface with fewer void-spaces in the material, but a much more prominent peak has been observed.

      To recapitulate, it is necessary to conclude that these leather shavings waste/recycled EVA polymer composites with lower cost can be employed for multipurpose applications as well as in the reduction of environmental pollution.

## 18. Suggestions for Future Work

      The subsequent suggestions were proposed premised on aforesaid experimental outcomes.

      The leather shavings are merely employed in the current study to reuse, reprocess, and fabricate flexible composites. However, this study also strongly recommends that future research can be done to assess the various other leather residues which have been produced during the reutilization and reprocessing techniques.

      By using various cross-linkers, modifications can be made to the "leather fibers reinforced recycled EVA" polymer composites in order to broaden their application areas and improve their mechanical properties.

      The precise degradation mechanisms would be made clearer by a combined TGA-FTIR investigation of such model composition (1:1). In order to explore the energy changes associated with the degradation features of these model compounds, advanced calorimetric techniques may also be used. It is equally important to combine recycled EVA polymer with leather wastes in suitable reactors on a large scale and, furthermore, to study the various

processing parameters for their pyrolysis in controlled environments, even though such microscopic analysis could increase understanding of the fundamental degradation process.

While an increase in contact points between the constituents could significantly degrade the recycled EVA matrix, a reduction in particle size and the resulting increase in surface area would improve reinforcing. Therefore, it may be possible to study the influence of surface area on the potential degradation of the recycled EVA matrix. Additionally, the processes that would eliminate chromium from the leather shavings might theoretically avert recycled EVA matrix deterioration. It is worthwhile to look at the characteristics of leather composites with recycled EVA polymer if the leather shavings are to be utilized directly without any processing. Thus, it might be conceivable to incorporate the recycling of 'recycled-EVA polymer' and the 'tannery wastes' in this process.

**Supplementary Materials:** The following supporting information can be downloaded at: https://www.mdpi.com/article/10.3390/su15054333/s1, Figure S1. (a) Leather-shavings employed as reinforcing-particulates, and (b) Recycled EVA granules Adapted from the reference [45]. Figure S2. Process flow diagram for recycling thermoplastic elastomer and polymer composites from solid leather wastes. Figure S3. (a) "Neat recycled-EVA polymeric-composites Adapted from the reference [45]", and (b) "Leather-shavings/recycled EVA-polymeric composites". Figure S4. "Dumb-bell shaped' Tensile Test Sample Adapted from the reference [45]. Figure S5. (a,b) "Variation of tensile load (N) against extension (mm) of recycled EVA polymer composites with or without leather shavings". Figure S6. Comparison of "Compressive strength (MPa)" for "neat-recycled EVA", and "recycled-EVA polymer composites". Table S1. Compositions of polymer composite samples in wt.-ratios. Table S2. Standard test methods, equipment details with model/machine numbers, and standards were followed in this investigation. Table S3. Absorbance values were obtained with the phosphate buffer testing procedure. Table S4. Absorbance values were obtained with a water testing procedure. Table S5. Presence of functional group in neat recycled EVA without incorporating leather fiber reinforcements in FTIR analysis. Table S6. Presence of functional group in Leather shavings/recycled EVA polymer composites in FTIR analysis.

**Author Contributions:** Conceptualization, S.S. (Shubham Sharma); methodology, S.S. (Shubham Sharma); formal analysis, S.S. (Shubham Sharma), P.S., J.S., S.M.R., S.S. (S. Siengchin); investigation, S.S. (Shubham Sharma); resources, S.S. (Shubham Sharma), P.S., J.S., S.M.R., S.S. (S. Siengchin); writing—original draft preparation, S.S. (Shubham Sharma); writing—review and editing, S.S. (Shubham Sharma), P.S., J.S., S.M.R., S.S. (S. Siengchin); supervision, S.S. (Shubham Sharma), P.S., J.S., S.M.R., S.S. (S. Siengchin); project administration, S.S. (Shubham Sharma), P.S., J.S. All authors have read and agreed to the published version of the manuscript.

**Funding:** This article received no external fundings from any of the sources.

**Institutional Review Board Statement:** Not applicable.

**Informed Consent Statement:** Not applicable.

**Data Availability Statement:** No data were used to support this study.

**Acknowledgments:** The author Shubham Sharma wishes to acknowledge the Department of RIC, IK Gujral Punjab Technical University, Kapurthala, Punjab, India, for providing an opportunity to conduct this research task. The author P. Sudhakara gratefully acknowledges the support from Science and Engineering Research Board (SERB, YSS/2015/001294), New Delhi, India and CLRI-MLP-27. Furthermore, the authors are thankful to the department of CATERS, Chennai, for providing the characterization facilities (CSIR-CLRI Communication no. 1671).

**Conflicts of Interest:** The authors declare no conflict of interest.

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
