# Peer review of "Fabrication of Novel Polymer Composites from Leather Waste Fibers and Recycled Poly(Ethylene-Vinyl-Acetate) for Value-Added Products"

_sustainability, doi:10.3390/su15054333_

Round 1

Reviewer 1 Report

The author has made a good attempt to understand the Developing novel polymer composites produced from leather waste fibers and recycled Poly (Ethylene-vinyl-acetate) polymer wastes: Recycling of footwear and plastic wastes for application in value-added products. However few comments are mentioned below:

1. Author should clarify whether it is a review or an experimental article.

2. Why the author has used SEM-EDX reference pictures in the article? If it is an original experimental article then it can be generated.

3. Sample location and procedure of sampling are missing.

4. Author has not defined the scope and objective effectively. 

5. Author is discussing application as an objective how it is possible?

6.  Author didn't discuss any mechanism process and interpretation of SEM_EDX images. 

 7. Research component is missing in this article. 

Author Response

09.02.2023

Dear Prof. (Dr.) Editor-in-chief,

Thank you for considering the manuscript entitled, “Fabrication of novel polymer composites from leather waste fibers and recycled Poly (Ethylene-vinyl-acetate) polymer: Re-cycling of footwear and plastic wastes for application in value-added products”, for the publication in SUSTAINABILITY (MDPI). I am grateful to you and the reviewers for the valuable suggestions provided. I like to resubmit our revised version of the manuscript by adding response to all your comments. Below please find the answers and actions taken to address these comments. All the suggestions are incorporated and highlighted with the RED COLOR in manuscript.

NOTE: All the necessary changes/added sentence has been shown in the RED COLOR.

The locations of these changes have been mentioned, where possible, in the action points that respond to each reviewers’ comments. Here are the responses to the reviewer comments:

AUTHOR RESPONSE TO REVIEWER AND EDITOR COMMENTS

Manuscript Number.: sustainability-2135688

Title: Fabrication of novel polymer composites from leather waste fibers and recycled Poly (Ethylene-vinyl-acetate) polymer: Re-cycling of footwear and plastic wastes for application in value-added products

Journal: Sustainability (MDPI)

The manuscript has been thoroughly modified and improved the quality of the content to meet the standards of the Journal. All the suggestions made by the learned referees are included in the revised manuscript. We are extremely thankful to the referees & editor(s) for their constructive comments and appreciation.

Response to Reviewer’s Comments

The authors are grateful to the reviewers for their suggestions that have all contributed to improving the manuscript. Once again, the authors are extremely thankful for the observations and the comments of the reviewers. All the comments are appropriately addressed and now the quality of the article has been appreciably enhanced before the consideration for publications. The rebuttal file is enclosed indicating the revisions incorporated in the article as suggested. The revisions are carried out in RED COLOR in the text of the manuscript for better visibility to the reviewers and as well as to the editor. We have made the modifications as per their suggestions in the revised manuscript and changes are also marked up using the RED COLOR function.

All in all, the authors should thank the reviewers for their meticulous observations in reviewing the article. All the issues raised by the authors are appropriately addressed as stated in the following table,

Response to Reviewers comments

S. No.

COMMENTS

Action taken

REVIEWER -1 COMMENTS

The author has made a good attempt to understand the Developing novel polymer composites produced from leather waste fibers and recycled Poly (Ethylene-vinyl-acetate) polymer wastes: Recycling of footwear and plastic wastes for application in value-added products. However, few comments are mentioned below:

1

Author should clarify whether it is a review or an experimental article.

Thank you for your insightful observations. With due respect to the comment of learnt referee, the authors would like to mention that the work is an original/experimental research article.

2

Why the author has used SEM-EDX reference pictures in the article? If it is an original experimental article then it can be generated.

Thank you for your learnt observations. The outcomes of SEM-EDX analysis are original produced from the experimental article. The SEM/EDX analysis of the fabricated composites are generated by using a “Scanning Electron Microscopy” (SEM) equipped with “Energy-dispersive analysis of X-Ray” (EDAX), a “Phenom World PhenomPro machine model at an accelerating voltage of 10 kV". Gold ions were used to sputter-coat samples (1 cm2), acting as a "conducting medium" when they were scanned with a "scanning microscope" made by Phenom World, model Phe-nomPro.

As Scanning electron microscopy coupled with energy-dispersive X-ray spectroscopy (SEM-EDX) is an important analytical technique that can provide valuable information about the microstructure and chemical composition of neat recycled EVA polymer composites and leather shaving fibers reinforced recycled polymer composites.

SEM-EDX provides detailed images of the microstructure of the fabricated leather loaded recycled EVA polymer composites, which can be used to evaluate the dispersion and distribution of the leather shaving fibers within the recycled EVA polymer matrix. The microstructure information can also be used to assess the compatibility and adhesion between the two components, which are critical factors in determining the mechanical properties of the fabricated composites.

In addition, SEM-EDX can be used to identify the chemical composition of the leather shaving fibers and the recycled polymer matrix. This information is important for understanding the behavior of the fabricated leather loaded recycled EVA polymer composites under different conditions and for determining the effect of the leather shaving fibers on the mechanical, morphological, chemical, microstructural, and thermal properties of the recycled EVA matrix.

Furthermore, SEM-EDX can be used to detect the presence of contaminants, such as salts, oils, or other impurities, in the leather shaving fibers and the recycled EVA polymer matrix. Contaminants can have a significant impact on the performance of the composites, and their detection and quantification using SEM-EDX can help to improve the quality of the final composites.

Moreover, SEM-EDX can be used to investigate the surface morphology of the leather shaving fibers and the recycled EVA polymer matrix. This information is critical for understanding the interactions between the two components and the effect of these interactions on the overall performance of the composites.

In conclusion, SEM-EDX analysis is an important tool for characterizing leather shaving fibers reinforced recycled EVA polymer composites. It provides valuable information about the microstructure, chemical composition, and surface morphology of the fabricated leather loaded recycled EVA polymer composites, which is essential for understanding their behavior and improving their performance.

3

Sample location and procedure of sampling are missing.

Thank you for your insightful observations. The details of the Sample location (materials procurement location with complete details), and procedure of sampling (Fabrication procedure of polymer composites) have been mentioned in the Materials section as highlighted with a red color.

4

Author has not defined the scope and objective effectively.

The authors are enormously thankful to the learnt referee for their knowledgeable insights on the same. The scope and objective in context with the novelty statement and scientific contributions have extensively been enumerated at the end of Introduction section as exhibited in the revised manuscript.

Suggestions have been incorporated in the revised version of the article as per the learnt comment by including the novelty/significance part immediately at the end of the Introduction section.

At the end of Introduction section, the literature review has been emphasized and further accompanied with research gap, problem formulation, research objectives, and methodology for the purpose to show the criticality of the current research.

For your kind consideration and perusal, I am enumerating the response to the valuable comment as mentioned below,

NOVELTY STATEMENT:

None of the above has carried out experiments regarding the efficient utilization of leather waste (Leather shavings) with maximum leather fiber-loading of 1:1 in combination with recycled EVA polymer matrix for footwear, transportation applications, etc. [17, 19, 22, 47, 49, 51, 55, 68, 70, 127, 133]. As a consequence, the recommended suggestions of this research might well be valuable for the leather as well as polymeric industries in order to significantly alleviate the load of disposal of solid-waste to the greatest extent possible. Till date, no work has been reported on the utilization of the solid leather waste as a reinforcing particulate in the recycled EVA thermoplastic matrix for the fabrication of leather waste reinforced polymeric composites. Thus, the suggested concept of such research could be substantially effective and compelling for the leather and polymeric sectors to eradicate the strain of solid-waste management to the greatest extent practicable.

No literature is available in which researchers have tried to modify the processing method of solid leather waste/recycled EVA thermoplastics matrix-based polymeric composites by adding suitable additives such as lubricants/oils, paraffins and naphthalene's to improve the homogeneity/ proper blending of polymer and leather thus to produce composites of superior properties for desired applications.

Limited work is available on the study of mechanical, thermal, structural, morphological, CRT, cushioning properties of leather waste reinforced recycled/scrap thermoplastic polymer composites. Therefore, opportunities exist is to investigate the properties of these leather waste reinforcement polymer composites so as to explore the uses for multifunctional applications.

It is also intended to see the effect of additives, and plasticizers on the physicochemical, mechanical, thermal, volumetric wear-loss, damping, cushioning/shock absorption, compression and resilience, and Morphology analysis on the fabricated polymer composites

Finally, the key goal of the present study is to develop Leather solid waste/Polymer composites for wide range of applications such as footwear, leather ancillaries, automotive, transportation and packaging at relatively low cost, thus to create value addition to the waste as well as to minimize the effect on environment from the leather and polymer waste disposal.

Referred articles:

[17]. T.J. Madera-Santana, M. Aguilar-Vega, A. Marquez, F. Moreno, M. Richardson, J. Machin, Production of leather-like composites using short leather fibers, II. Mechanical characterization, Polymer Composites 23 (2004).

[19]. G. Garcia, E. A. P. dos Reis, E. R. Budemberg, D. L. da Silva Agostini, L. O. Salmazo, F.C.  Cabrera, A. E. Job, Natural rubber/leather waste composite foam: A new eco-friendly material and recycling approach, Journal of Applied Polymer Science 132 (2014) n/a–n/a. doi:10.1002/app.41636. URL https://dx.doi.org/10.1002/app.41636

[22]. M. Parisi, A. Nanni, M. Colonna, Recycling of Chrome-Tanned Leather and Its Utilization as Polymeric Materials and in Polymer-Based Composites: A Review, Polymers 13 (3) (2021) 429–429. doi:10.3390/polym13030429.

[47]. R. D. Kale, N. C. Jadhav, Utilization of waste leather for the fabrication of composites and to study its mechanical and thermal properties, SN Applied Sciences 1 (10) (2019) 1–9. doi:10.1007/s42452-019- 1230-9.

[49]. J. D. Ambrósio, A. A. Lucas, H. Otaguro, L. C. Costa, Preparation and characterization of poly (vinyl butyral)-leather fiber composites, Polymer Composites 32 (5) (2011) 776–785. doi:10.1002/pc.21099.

[51]. Nanni, A.; Parisi, M.; Colonna, M.; Messori, M. Thermo-Mechanical and Morphological Properties of Polymer Composites Reinforced by Natural Fibers Derived from Wet Blue Leather Wastes: A Comparative Study. Polymers 2021, 13, 1837. https://doi.org/10.3390/polym13111837.

[55]. K. Ravichandran, N. Natchimuthu, Vulcanization characteristics and mechanical properties of natural rubber-scrap rubber compositions filled with leather particles, Polymer International 54 (3) (2005) 553– 559. doi:10.1002/pi.1725.

[68]. S. Joseph, T. S. Ambone, A. V. Salvekar, S. N. Jaisankar, P.  Saravanan, E. Deenadayalan, Processing and characterization of waste leather based polycaprolactone biocomposites, Polymer Composites 38 (12) (2017) 2889–2897. doi:10.1002/pc.23891.

[70]. P. Saikia, T. Goswami, D. Dutta, N. K. Dutta, P. Sengupta, D. Neog, Development of a flexible composite from leather industry waste and evaluation of their physico-chemical properties, Clean Technologies and Environmental Policy 19 (8) (2017) 2171–2178. doi:10.1007/s10098-017-1396-z.

[127]. Cardona, N., Velásquez, S. and Giraldo, D., 2017. Characterization of leather wastes from chrome tanning and its effect as filler on the rheometric properties of natural rubber compounds. Journal of Polymers and the Environment, 25(4), pp.1190-1197.

[133]. Li, C., Feng, X. and Ding, E., 2015. Preparation, properties, and characterization of novel fine leather fibers/polyvinyl alcohol composites. Polymer Composites, 36(7), pp.1186-1194.

5

Author is discussing application as an objective how it is possible?

In appreciation of the knowledgeable comments of the learned referee, the authors extend their gratitude. the authors have now modified the objective at the end of Introduction section by removing the vague content in reference to the objective of the current study.

Furthermore, in context with the valuable query, the present novel research study is limited to meticulously modifying the processing method by incorporating the highest leather fibre loading of 1:1 composition with 100% leather fibre absorption with respect to the recycled EVA polymer matrix and excellent remarkable properties for multifunctional value-addition applications.

The applications of the leather filled polymer composites have been decided based upon the extensive characterizations performed which includes physicomechanical (Tensile, Compressive strength, Density, Abrasion-resistance, Adhesion-strength, Hardness, Tear-resistance, Compression, and resilience, damping, and water-absorption), thermal (TGA and DSC studies), morphological (SEM, EDAX), chemical properties (FTIR-ATR), structural crystallization (XRD), elemental mapping (EDAX), and topographic image-mapping studies/characterizations (AFM) in order to suggest the fabricated composite materials for broad range of applications.

For your kind consideration and perusal, I am enumerating the response to the valuable comment as mentioned below,

NOVELTY STATEMENT:

None of the above has carried out experiments regarding the efficient utilization of leather waste (Leather shavings) with maximum leather fiber-loading of 1:1 in combination with recycled EVA polymer matrix for footwear, transportation applications, etc. [17, 19, 22, 47, 49, 51, 55, 68, 70, 127, 133]. As a consequence, the recommended suggestions of this research might well be valuable for the leather as well as polymeric industries in order to significantly alleviate the load of disposal of solid-waste to the greatest extent possible. Till date, no work has been reported on the utilization of the solid leather waste as a reinforcing particulate in the recycled EVA thermoplastic matrix for the fabrication of leather waste reinforced polymeric composites. Thus, the suggested concept of such research could be substantially effective and compelling for the leather and polymeric sectors to eradicate the strain of solid-waste management to the greatest extent practicable.

No literature is available in which researchers have tried to modify the processing method of solid leather waste/recycled EVA thermoplastics matrix-based polymeric composites by adding suitable additives such as lubricants/oils, paraffins and naphthalene's to improve the homogeneity/ proper blending of polymer and leather thus to produce composites of superior properties for desired applications.

Limited work is available on the study of mechanical, thermal, structural, morphological, CRT, cushioning properties of leather waste reinforced recycled/scrap thermoplastic polymer composites. Therefore, opportunities exist is to investigate the properties of these leather waste reinforcement polymer composites so as to explore the uses for multifunctional applications.

It is also intended to see the effect of additives, and plasticizers on the physicochemical, mechanical, thermal, volumetric wear-loss, damping, cushioning/shock absorption, compression and resilience, and Morphology analysis on the fabricated polymer composites

Finally, the key goal of the present study is to develop Leather solid waste/Polymer composites for wide range of applications such as footwear, leather ancillaries, automotive, transportation and packaging at relatively low cost, thus to create value addition to the waste as well as to minimize the effect on environment from the leather and polymer waste disposal.

REALISTIC APPLICATIONS OF THE DEVELOPED LEATHER WASTE/RECYCLED EVA POLYMER COMPOSITES

The applications of the leather filled polymer composites have been decided based upon the extensive characterizations performed which includes physicomechanical (Tensile, Compressive strength, Density, Abrasion-resistance, Adhesion-strength, Hardness, Tear-resistance, Compression, and resilience, damping, and water-absorption), thermal (TGA and DSC studies), morphological (SEM, EDAX), chemical properties (FTIR-ATR), structural crystallization (XRD), elemental mapping (EDAX), and topographic image-mapping studies/characterizations (AFM) in order to suggest the fabricated composite materials for broad range of applications.

For your kind consideration and perusal, I am enumerating the response to the valuable comment in detailed manner as mentioned below,

Table. Detailed experimental testing studies performed with the desired applications of the developed leather waste/recycled EVA polymer composites used in this research study

S.No.

Experiments performed

Prospective applications of the developed leather waste/recycled EVA polymer composites

1

Tensile Strength, Elongation (%), and Modulus of Polymer composites

Composite-sheets, footwear and clothing industries, Leather-like personal goods, Floorings, personal safety goods, construction and buildings materials, automotive interior-trim molding parts, Shoe soles, floor-covering and moldings with quite-outstanding physical characteristics, footwear, leather ancillaries, automotive, transportation, and packaging.

2

Compressive Strength and Compressive Modulus of Polymer composites

Composite-sheets, automotive interior-trim molding parts, Shoe soles, floor-covering and moldings with quite-outstanding physical characteristics, footwear, leather ancillaries, automotive, transportation

3

Tear Strength of Polymer composites

Footwear, furniture leathers, garments, and automotive upholstery

4

Adhesion Strength of Polymer composites

Footwear industry, bonding upper/sole, sports shoes, safety shoes, Decorative furniture applications, Leather boards, Adhesives in wood working industries

5

Compression and resilience test (CRT) of Polymer composites

Transportation, structural load-bearing, Shock absorbing applications, machinery mountings, vibration dampers, and seals

6

Abrasion resistance of Polymer composites

Durability, structural, transportation, footwear, floor coverings and moldings, Construction and buildings materials

7

Hardness of Polymer composites

Composite sheets, automotive interior-trim molding parts, Shoe soles, floor-covering and moldings with quite-outstanding physical characteristics, footwear, leather ancillaries, automotive, transportation, and structural

8

Thermogravimetric Analysis (TGA) of Polymer composites

Composite sheets, thermal stability of materials, footwear, automotive, transportation, structural, and thermal insulation panels.

9

Differential Scanning Calorimetry (DSC) of Polymer composites

Composite sheets, Glass-transitions, phase-transitions of materials, footwear, automotive, transportation, and structural.

10

Fourier-transform infrared spectroscopy- Attenuated total reflection (FTIR-ATR) of Polymer composites

Functional group analysis, contaminants identification, and microanalysis of leather composites for automotive applications

11

Scanning Electron Microscopy (SEM Analysis) of Polymer composites

Construction materials, automobile interior moldings, heat and sound insulating boards, shoe soles, flooring materials and moldings with good anti-static properties

12

Elemental Composition analysis of Polymer composites

Construction materials, automobile interior moldings, shoe soles, flooring materials and moldings with good viscoelastic properties.

13

Phase identification and Structural analysis of Polymer composites

Construction materials, automobile interior moldings, shoe soles, flooring materials and moldings, air permeability, and good appearances

14

Atomic Force Microscopy (AFM analysis) of Polymer composites

Topographic image mapping properties for Construction materials, automobile interior moldings, heat and sound insulating boards, shoe soles, flooring materials and moldings with good anti-static properties, air permeability, and good appearances

6

Author didn't discuss any mechanism process and interpretation of SEM_EDX images.

Thank you for your constructive comments. The response in context with your valuable query has thoroughly been mentioned in text which has been highlighted with red colour. The detailed discussion on the SEM-EDAX analysis of the fabricated composites has now been incorporated in the revised article with proper justification and physics behind the mechanisms.

The authors have detailed described and extensively mentioned the general trend variation in the properties of the composites with the deep-discussion on the materials behaviour with proper justification or physics behind the mechanisms, trend variation, novelty in the properties, applications, processing fabrication techniques, by appropriately citing the relevant literature studies.

Furthermore, the findings/outcomes have been thoroughly addressed by detailed comparative analysis among the related composite materials.

The authors hope that this will give an impetus to the overall content, and novelty in the article and thus, enable readers to identify the cutting-edge technology by thoroughly go-through the article.

As a consequence, the recommended suggestions of this research might well be valuable for the leather as well as polymeric industries in order to significantly alleviate the load of disposal of solid-waste to the greatest extent possible. Till date, no work has been reported on the utilization of the solid leather waste as a reinforcing particulate in the recycled EVA thermoplastic matrix for the fabrication of leather waste reinforced polymeric composites. Thus, the suggested concept of such research could be substantially effective and compelling for the leather and polymeric sectors to eradicate the strain of solid-waste management to the greatest extent practicable.

No literature is available in which researchers have tried to modify the processing method of solid leather waste/recycled EVA thermoplastics matrix based polymeric composites by adding suitable additives such as lubricants/oils, paraffins and naphthalene's to improve the homogeneity/ proper blending of polymer and leather thus to produce composites of superior properties for desired applications.

Limited work is available on the study of mechanical, thermal, structural, morphological, CRT, cushioning properties of leather waste reinforced recycled/scrap thermoplastic polymer composites. Therefore, opportunities exist is to investigate the properties of these leather waste reinforcement polymer composites so as to explore the uses for multifunctional applications.

It is also intended to see the effect of additives, and plasticizers on the physicochemical, mechanical, thermal, volumetric wear-loss, damping, cushioning/shock absorption, compression and resilience, and Morphology analysis on the fabricated polymer composites

Finally, the key goal of the present study is to develop Leather solid waste/Polymer composites for wide range of applications such as footwear, leather ancillaries, automotive, transportation and packaging at relatively low cost, thus to create value addition to the waste as well as to minimize the effect on environment from the leather and polymer waste disposal.

As, the matrix material used in the above-mentioned previous published literature source is a pure/virgin polymer, however, in our case, we have used recycled EVA polymer matrix, secondly, in the aforementioned literature article, the reinforcing constituents have been used as leather buffing dust (Proteinaceous collagen fibril-type fine powder form which has been generated when finished leather has subjected to the abrasion process), however, in this current study, we have used leather shaving fibers (Proteinaceous collagen fibril-type threaded form rugged structure) as reinforcing constituents.

Additionally, the impregnation of Leather shaving fibers into the polymer in higher quantities is a complex process. However, in this present study, suitable additives such as lubricants/oils, paraffin, and naphthalene's have been utilised to improve the homogeneity/ proper blending of polymer and Leather shaving fibers, thus producing composites of superior properties for desired multifunctional value-addition applications. From in-depth literary studies, it has been unveiled that no work has been made accessible on carrying-out experiments regarding the efficient utilization of leather waste (Leather shavings) with maximum leather fiber-loading of 1:1 in combination with recycled EVA polymer matrix for footwear, transportation applications, etc [17, 19, 22, 47, 49, 51, 55, 68, 70, 127, 133].

In addition, no literature has been made available in which researchers have tried to modify the processing method of Leather shaving fibers/recycled EVA thermoplastics matrix-based polymeric composites by adding suitable additives such as lubricants/oils, paraffin and naphthalene are to improve the homogeneity/ proper blending of recycled EVA polymer and Leather shaving fibers thus to produce composites of superior properties for desired applications.

Although, the authors have tried to mention in this paragraph is that this study is first timely reported as no existing literature studies has used the recycled EVA polymer as a matrix material and leather shaving fibers as reinforcing constituents to fabricate the flexible polymer composite sheets with maximum leather fiber loading content of 1:1.

Although in this regard, the authors would like to mention that the till-date, the prior studies have merely reported the usage of the virgin/pure thermoplastic polymers or elastomers as matrix materials and no research work has claimed to use both the recycled polymer materials and leather fiber wastes as particulate fillers to develop the composite sheets.

From the in-depth prior literature studies, no one has carried out experimentation on leather wastes reinforced recycled EVA polymer composite with maximum fiber loading content of 1:1.

Researchers have conducted their experiments with lower concentration of utilizing leather wastes of maximum up to 40% content by percent weight, however, with different virgin polymer matrix materials (not recycled EVA material), as they have reported difficulties in homogenous/uniform blending or mixing of leather fiber wastes and polymer matrix with high amount of leather content.

In this context of the aforementioned former concrete statement, our main focus has centered to fabricate the composites with the higher amount or quantity or volume of leather fiber content of 1:1 by thoroughly refining the manufacturing process using novel additives/fillers/plasticizers/lubricant oils during processing in two-rolling mill method. In-addition to this, the authors have also done the comparative analysis of the 1:1 content with the neat recycled EVA polymer.

Moreover, this study also covers an extensive characterization for future applications prospective.

The authors would like to respectfully mention that the related comparison evaluation has already been enumerated in detailed manner for all the respective physicomechanical, thermal, morphological, chemical properties, structural crystallization, elemental mapping, and topographic image-mapping studies/characterizations for the leather polymer composites of leather loading content of 1:1 with the neat recycled EVA matrix (without leather-fiber reinforcements).

Therefore, the present novel research study is limited to meticulously modifying the processing method by incorporating the highest leather fibre loading of 1:1 composition with 100% leather fibre absorption with respect to the recycled EVA polymer matrix and excellent remarkable properties for multifunctional value-addition applications.

Referred articles:

[17]. T.J. Madera-Santana, M. Aguilar-Vega, A. Marquez, F. Moreno, M. Richardson, J. Machin, Production of leather-like composites using short leather fibers, II. Mechanical characterization, Polymer Composites 23 (2004).

[19]. G. Garcia, E. A. P. dos Reis, E. R. Budemberg, D. L. da Silva Agostini, L. O. Salmazo, F.C.  Cabrera, A. E. Job, Natural rubber/leather waste composite foam: A new eco-friendly material and recycling approach, Journal of Applied Polymer Science 132 (2014) n/a–n/a. doi:10.1002/app.41636. URL https://dx.doi.org/10.1002/app.41636

[22]. M. Parisi, A. Nanni, M. Colonna, Recycling of Chrome-Tanned Leather and Its Utilization as Polymeric Materials and in Polymer-Based Composites: A Review, Polymers 13 (3) (2021) 429–429. doi:10.3390/polym13030429.

[47]. R. D. Kale, N. C. Jadhav, Utilization of waste leather for the fabrication of composites and to study its mechanical and thermal properties, SN Applied Sciences 1 (10) (2019) 1–9. doi:10.1007/s42452-019- 1230-9.

[49]. J. D. Ambrósio, A. A. Lucas, H. Otaguro, L. C. Costa, Preparation and characterization of poly (vinyl butyral)-leather fiber composites, Polymer Composites 32 (5) (2011) 776–785. doi:10.1002/pc.21099.

[51]. Nanni, A.; Parisi, M.; Colonna, M.; Messori, M. Thermo-Mechanical and Morphological Properties of Polymer Composites Reinforced by Natural Fibers Derived from Wet Blue Leather Wastes: A Comparative Study. Polymers 2021, 13, 1837. https://doi.org/10.3390/polym13111837.

[55]. K. Ravichandran, N. Natchimuthu, Vulcanization characteristics and mechanical properties of natural rubber-scrap rubber compositions filled with leather particles, Polymer International 54 (3) (2005) 553– 559. doi:10.1002/pi.1725.

[68]. S. Joseph, T. S. Ambone, A. V. Salvekar, S. N. Jaisankar, P.  Saravanan, E. Deenadayalan, Processing and characterization of waste leather based polycaprolactone biocomposites, Polymer Composites 38 (12) (2017) 2889–2897. doi:10.1002/pc.23891.

[70]. P. Saikia, T. Goswami, D. Dutta, N. K. Dutta, P. Sengupta, D. Neog, Development of a flexible composite from leather industry waste and evaluation of their physico-chemical properties, Clean Technologies and Environmental Policy 19 (8) (2017) 2171–2178. doi:10.1007/s10098-017-1396-z.

[127]. Cardona, N., Velásquez, S. and Giraldo, D., 2017. Characterization of leather wastes from chrome tanning and its effect as filler on the rheometric properties of natural rubber compounds. Journal of Polymers and the Environment, 25(4), pp.1190-1197.

[133]. Li, C., Feng, X. and Ding, E., 2015. Preparation, properties, and characterization of novel fine leather fibers/polyvinyl alcohol composites. Polymer Composites, 36(7), pp.1186-1194.

7

Research component is missing in this article.

The authors gratefully convey their admiration to the knowledgeable referee for their constructive insights. With due respect to the comment of learnt expert, the authors would like to mention that this is an ingenious/novel original research article.

Based upon the fruitful recommendations of knowledgeable expert, the authors have tried to mention in this paragraph is that this study is first timely reported as no existing literature studies has used the recycled EVA polymer as a matrix material and leather shaving fibers as reinforcing constituents to fabricate the flexible polymer composite sheets with maximum leather fiber loading content of 1:1.

Although in this regard, the authors would like to mention that the till-date, the prior studies have merely reported the usage of the virgin/pure thermoplastic polymers or elastomers as matrix materials and no research work has claimed to use both the recycled polymer materials and leather fiber wastes as particulate fillers to develop the composite sheets.

Moreover, the authors have added the table showing the formulations or compositions of polymer composite samples in wt.-ratios to make it more descriptive, concise and precise to illustrate the different compositions used in the current study.

From the in-depth prior literature studies, no one has carried out experimentation on leather wastes reinforced recycled EVA polymer composite with maximum fiber loading content of 1:1.

Researchers have conducted their experiments with lower concentration of utilizing leather wastes of maximum up to 40% content by percent weight, however, with different virgin polymer matrix materials (not recycled EVA material), as they have reported difficulties in homogenous/uniform blending or mixing of leather fiber wastes and polymer matrix with high amount of leather content.

In this context of the aforementioned former concrete statement, our main focus has centered to fabricate the composites with the higher amount or quantity or volume of leather fiber content of 1:1 by thoroughly refining the manufacturing process using novel additives/fillers/plasticizers/lubricant oils during processing in two-rolling mill method. In-addition to this, the authors have also done the comparative analysis of the 1:1 content with the neat recycled EVA polymer.

Additionally, this study also covers an extensive characterization for future applications prospective.

The authors would like to respectfully mention that the related comparison evaluation has already been enumerated in detailed manner for all the respective physicomechanical, thermal, morphological, chemical properties, structural crystallization, elemental mapping, and topographic image-mapping studies/characterizations for the leather polymer composites of leather loading content of 1:1 with the neat recycled EVA matrix (without leather-fiber reinforcements).

Therefore, the present novel research study is limited to meticulously modifying the processing method by incorporating the highest leather fibre loading of 1:1 composition with 100% leather fibre absorption with respect to the recycled EVA polymer matrix and excellent remarkable properties for multifunctional value-addition applications.

The applications of the leather filled polymer composites have been decided based upon the extensive characterizations performed which includes physicomechanical (Tensile, Compressive strength, Density, Abrasion-resistance, Adhesion-strength, Hardness, Tear-resistance, Compression, and resilience, damping, and water-absorption), thermal (TGA and DSC studies), morphological (SEM, EDAX), chemical properties (FTIR-ATR), structural crystallization (XRD), elemental mapping (EDAX), and topographic image-mapping studies/characterizations (AFM) in order to suggest the fabricated composite materials for broad range of applications.

For your kind consideration and perusal, I am enumerating the response to the valuable comment as mentioned below,

NOVELTY STATEMENT:

None of the above has carried out experiments regarding the efficient utilization of leather waste (Leather shavings) with maximum leather fiber-loading of 1:1 in combination with recycled EVA polymer matrix for footwear, transportation applications, etc. [17, 19, 22, 47, 49, 51, 55, 68, 70, 127, 133]. As a consequence, the recommended suggestions of this research might well be valuable for the leather as well as polymeric industries in order to significantly alleviate the load of disposal of solid-waste to the greatest extent possible. Till date, no work has been reported on the utilization of the solid leather waste as a reinforcing particulate in the recycled EVA thermoplastic matrix for the fabrication of leather waste reinforced polymeric composites. Thus, the suggested concept of such research could be substantially effective and compelling for the leather and polymeric sectors to eradicate the strain of solid-waste management to the greatest extent practicable.

No literature is available in which researchers have tried to modify the processing method of solid leather waste/recycled EVA thermoplastics matrix-based polymeric composites by adding suitable additives such as lubricants/oils, paraffins and naphthalene's to improve the homogeneity/ proper blending of polymer and leather thus to produce composites of superior properties for desired applications.

Limited work is available on the study of mechanical, thermal, structural, morphological, CRT, cushioning properties of leather waste reinforced recycled/scrap thermoplastic polymer composites. Therefore, opportunities exist is to investigate the properties of these leather waste reinforcement polymer composites so as to explore the uses for multifunctional applications.

It is also intended to see the effect of additives, and plasticizers on the physicochemical, mechanical, thermal, volumetric wear-loss, damping, cushioning/shock absorption, compression and resilience, and Morphology analysis on the fabricated polymer composites

Finally, the key goal of the present study is to develop Leather solid waste/Polymer composites for wide range of applications such as footwear, leather ancillaries, automotive, transportation and packaging at relatively low cost, thus to create value addition to the waste as well as to minimize the effect on environment from the leather and polymer waste disposal.

REALISTIC APPLICATIONS OF THE DEVELOPED LEATHER WASTE/RECYCLED EVA POLYMER COMPOSITES

The applications of the leather filled polymer composites have been decided based upon the extensive characterizations performed which includes physicomechanical (Tensile, Compressive strength, Density, Abrasion-resistance, Adhesion-strength, Hardness, Tear-resistance, Compression, and resilience, damping, and water-absorption), thermal (TGA and DSC studies), morphological (SEM, EDAX), chemical properties (FTIR-ATR), structural crystallization (XRD), elemental mapping (EDAX), and topographic image-mapping studies/characterizations (AFM) in order to suggest the fabricated composite materials for broad range of applications.

For your kind consideration and perusal, I am enumerating the response to the valuable comment in detailed manner as mentioned below,

Table. Detailed experimental testing studies performed with the desired applications of the developed leather waste/recycled EVA polymer composites used in this research study

S.No.

Experiments performed

Prospective applications of the developed leather waste/recycled EVA polymer composites

1

Tensile Strength, Elongation (%), and Modulus of Polymer composites

Composite-sheets, footwear and clothing industries, Leather-like personal goods, Floorings, personal safety goods, construction and buildings materials, automotive interior-trim molding parts, Shoe soles, floor-covering and moldings with quite-outstanding physical characteristics, footwear, leather ancillaries, automotive, transportation, and packaging.

2

Compressive Strength and Compressive Modulus of Polymer composites

Composite-sheets, automotive interior-trim molding parts, Shoe soles, floor-covering and moldings with quite-outstanding physical characteristics, footwear, leather ancillaries, automotive, transportation

3

Tear Strength of Polymer composites

Footwear, furniture leathers, garments, and automotive upholstery

4

Adhesion Strength of Polymer composites

Footwear industry, bonding upper/sole, sports shoes, safety shoes, Decorative furniture applications, Leather boards, Adhesives in wood working industries

5

Compression and resilience test (CRT) of Polymer composites

Transportation, structural load-bearing, Shock absorbing applications, machinery mountings, vibration dampers, and seals

6

Abrasion resistance of Polymer composites

Durability, structural, transportation, footwear, floor coverings and moldings, Construction and buildings materials

7

Hardness of Polymer composites

Composite sheets, automotive interior-trim molding parts, Shoe soles, floor-covering and moldings with quite-outstanding physical characteristics, footwear, leather ancillaries, automotive, transportation, and structural

8

Thermogravimetric Analysis (TGA) of Polymer composites

Composite sheets, thermal stability of materials, footwear, automotive, transportation, structural, and thermal insulation panels.

9

Differential Scanning Calorimetry (DSC) of Polymer composites

Composite sheets, Glass-transitions, phase-transitions of materials, footwear, automotive, transportation, and structural.

10

Fourier-transform infrared spectroscopy- Attenuated total reflection (FTIR-ATR) of Polymer composites

Functional group analysis, contaminants identification, and microanalysis of leather composites for automotive applications

11

Scanning Electron Microscopy (SEM Analysis) of Polymer composites

Construction materials, automobile interior moldings, heat and sound insulating boards, shoe soles, flooring materials and moldings with good anti-static properties

12

Elemental Composition analysis of Polymer composites

Construction materials, automobile interior moldings, shoe soles, flooring materials and moldings with good viscoelastic properties.

13

Phase identification and Structural analysis of Polymer composites

Construction materials, automobile interior moldings, shoe soles, flooring materials and moldings, air permeability, and good appearances

14

Atomic Force Microscopy (AFM analysis) of Polymer composites

Topographic image mapping properties for Construction materials, automobile interior moldings, heat and sound insulating boards, shoe soles, flooring materials and moldings with good anti-static properties, air permeability, and good appearances

Referred articles:

[17]. T.J. Madera-Santana, M. Aguilar-Vega, A. Marquez, F. Moreno, M. Richardson, J. Machin, Production of leather-like composites using short leather fibers, II. Mechanical characterization, Polymer Composites 23 (2004).

[19]. G. Garcia, E. A. P. dos Reis, E. R. Budemberg, D. L. da Silva Agostini, L. O. Salmazo, F.C.  Cabrera, A. E. Job, Natural rubber/leather waste composite foam: A new eco-friendly material and recycling approach, Journal of Applied Polymer Science 132 (2014) n/a–n/a. doi:10.1002/app.41636. URL https://dx.doi.org/10.1002/app.41636

[22]. M. Parisi, A. Nanni, M. Colonna, Recycling of Chrome-Tanned Leather and Its Utilization as Polymeric Materials and in Polymer-Based Composites: A Review, Polymers 13 (3) (2021) 429–429. doi:10.3390/polym13030429.

[47]. R. D. Kale, N. C. Jadhav, Utilization of waste leather for the fabrication of composites and to study its mechanical and thermal properties, SN Applied Sciences 1 (10) (2019) 1–9. doi:10.1007/s42452-019- 1230-9.

[49]. J. D. Ambrósio, A. A. Lucas, H. Otaguro, L. C. Costa, Preparation and characterization of poly (vinyl butyral)-leather fiber composites, Polymer Composites 32 (5) (2011) 776–785. doi:10.1002/pc.21099.

[51]. Nanni, A.; Parisi, M.; Colonna, M.; Messori, M. Thermo-Mechanical and Morphological Properties of Polymer Composites Reinforced by Natural Fibers Derived from Wet Blue Leather Wastes: A Comparative Study. Polymers 2021, 13, 1837. https://doi.org/10.3390/polym13111837.

[55]. K. Ravichandran, N. Natchimuthu, Vulcanization characteristics and mechanical properties of natural rubber-scrap rubber compositions filled with leather particles, Polymer International 54 (3) (2005) 553– 559. doi:10.1002/pi.1725.

[68]. S. Joseph, T. S. Ambone, A. V. Salvekar, S. N. Jaisankar, P.  Saravanan, E. Deenadayalan, Processing and characterization of waste leather based polycaprolactone biocomposites, Polymer Composites 38 (12) (2017) 2889–2897. doi:10.1002/pc.23891.

[70]. P. Saikia, T. Goswami, D. Dutta, N. K. Dutta, P. Sengupta, D. Neog, Development of a flexible composite from leather industry waste and evaluation of their physico-chemical properties, Clean Technologies and Environmental Policy 19 (8) (2017) 2171–2178. doi:10.1007/s10098-017-1396-z.

[127]. Cardona, N., Velásquez, S. and Giraldo, D., 2017. Characterization of leather wastes from chrome tanning and its effect as filler on the rheometric properties of natural rubber compounds. Journal of Polymers and the Environment, 25(4), pp.1190-1197.

[133]. Li, C., Feng, X. and Ding, E., 2015. Preparation, properties, and characterization of novel fine leather fibers/polyvinyl alcohol composites. Polymer Composites, 36(7), pp.1186-1194.

Hence, a scientific explanation of the obtained results has been refined and ameliorated up to a fervent extent. Results are enumerated, test methods are utterly described, interpretation have been corelated with results and previous literature findings. The overall summary should indicate the progress of the research and the limitations. 

Note: All the necessary changes/added sentence has been shown in the RED COLOR.

Thank you very much in advance for taking your time in reviewing this manuscript.

Sincerely, we hope you will find our revision satisfactory.

Reviewer 2 Report

General comment

The text is too long chaotic, and the quality of the English style must be improved. A lot of details can be eliminated (also on account of repetitions).

Abstract and Introduction should be revised and possibly some graphs could be either eliminated or included as supplementary material.

Just a few more specific comments of the first part of the results but the article should be re-examined after the first correction.

The EVA materials is not characterized. Some details must be added as for example the Molecular weight or the MFI.

Line 691 Something is missing?

Line 707 Unclear nanoparticles?

Figure 5 Repetitions and unclear X axis?

Line 748 Figure 5? Wrong number? Moreover, problems with the x axis.

SEM (Figure 6 that should have another number) Indeed It is hard to see what is described in the text surface is lined and filled with "bulged, twisted, and elongated materials"

Figure 10 Y axis Newtons or MPa? What is reported on the x-axis?

But the text ( as well as the Figures) must be changed in all the paragraphs.

Author Response

aa09.02.2023

Dear Prof. (Dr.) Editor-in-chief,

Thank you for considering the manuscript entitled, “Fabrication of novel polymer composites from leather waste fibers and recycled Poly (Ethylene-vinyl-acetate) polymer: Re-cycling of footwear and plastic wastes for application in value-added products”, for the publication in SUSTAINABILITY (MDPI). I am grateful to you and the reviewers for the valuable suggestions provided. I like to resubmit our revised version of the manuscript by adding response to all your comments. Below please find the answers and actions taken to address these comments. All the suggestions are incorporated and highlighted with the RED COLOR in manuscript.

NOTE: All the necessary changes/added sentence has been shown in the RED COLOR.

The locations of these changes have been mentioned, where possible, in the action points that respond to each reviewers’ comments. Here are the responses to the reviewer comments:

AUTHOR RESPONSE TO REVIEWER AND EDITOR COMMENTS

Manuscript Number.: sustainability-2135688

Title: Fabrication of novel polymer composites from leather waste fibers and recycled Poly (Ethylene-vinyl-acetate) polymer: Re-cycling of footwear and plastic wastes for application in value-added products

Journal: Sustainability (MDPI)

The manuscript has been thoroughly modified and improved the quality of the content to meet the standards of the Journal. All the suggestions made by the learned referees are included in the revised manuscript. We are extremely thankful to the referees & editor(s) for their constructive comments and appreciation.

Response to Reviewer’s Comments

The authors are grateful to the reviewers for their suggestions that have all contributed to improving the manuscript. Once again, the authors are extremely thankful for the observations and the comments of the reviewers. All the comments are appropriately addressed and now the quality of the article has been appreciably enhanced before the consideration for publications. The rebuttal file is enclosed indicating the revisions incorporated in the article as suggested. The revisions are carried out in RED COLOR in the text of the manuscript for better visibility to the reviewers and as well as to the editor. We have made the modifications as per their suggestions in the revised manuscript and changes are also marked up using the RED COLOR function.

All in all, the authors should thank the reviewers for their meticulous observations in reviewing the article. All the issues raised by the authors are appropriately addressed as stated in the following table,

Response to Reviewers comments

S. No.

COMMENTS

Action taken

REVIEWER -2 COMMENTS

1

General comment

The text is too long chaotic, and the quality of the English style must be improved. A lot of details can be eliminated (also on account of repetitions).

Thank you for the observations. Suggestions has been thoroughly incorporated in revised version of the article as per the learnt comment.

As per suggestions, all the mistakes in accordance with the writing skills and usage of the English language have now been polished up to a fervent extent with the assistance of the native English contributor/co-author of this article.

As per valuable suggestions received, the grammatical errors have now been removed and many sentences have been reframed so as to deliver clear interpretation in the revised manuscript.

The verbose explanation in the respective sections, and sub-sections, introduction, literature-review, experimentation, concluding remarks and suggestions for future work have been removed.

The authors have avoided the general discussion and furthermore has reported only the comparison of the significant findings unveiled different sections of the revised manuscript as highlighted in the red colour.

The authors would like to mention that the few undesirable sentences, figures, tables, and supplementary figures or tables from the revised manuscript have been omitted to a fervent extent. Moreover, the junk or unwanted content/text has been avoided by utterly eradicating the repeated phrases/words/sentences throughout the manuscript.

The authors have tried to refine the overall content, structure, formatting, and relevancy of the revised article as per the learnt recommendations.

The entire content in the revised article has been ameliorated up to fervent extent strictly as per the valuable recommendations. The abstract has been written in chronological sequential order as, it covers Introduction, Objective, Methodology, Results, and overall conclusion.

Additionally, the significant outcomes which are evident from the current investigation work has been enumerated in the abstract with percent improvement in the material properties with promising applications. The extensive revisions have been exhibited in the revised version of the manuscript.

Although, the authors have supported their valuable breakthrough findings with the previous existing studies as shown in the revised version of the manuscript with red color throughout the article.

The authors have been excavated or removed the unnecessary matter or data from the article and thus, the manuscript has been refined, prepare and contemplate it in more precise or accurate, removing the unwanted or junk content from the article, and thus, organized in the systematic orderly manner as till-date citing the literature studies, research gaps, problem formulation, objectives with research methodology as illustrated in the revised version of the manuscript with red color.

The point of discussion has been detailed enumerated in more comprehensive and extensive way by thoroughly supporting the present results and discussions with the previous literature studies.

The prominent noteworthy points or findings/outcomes has been thoroughly addressed in separate paragraphs with primarily emphasized on the percentage enhancement in the properties with detailed comparative analysis among the related composite materials by properly citing the past literary sources.

In-addition, the authors have detailed described and mentioned the general trend variation in the properties of the composites with the deep-discussion on the materials behaviour, trend variation, novelty in the properties, applications, processing fabrication techniques, by appropriately citing the relevant literature studies.

The authors hope that this will give an impetus to the overall content, and novelty in the article and thus, enable readers to identify the cutting-edge technology by thoroughly go-through the article.

2

Abstract and Introduction should be revised and possibly some graphs could be either eliminated or included as supplementary material.

Thank you for your valuable insinuations. The table 2 has thoroughly been deleted as per the suggestions.

Suggestions have been incorporated in the revised version of the article as per the learnt comment by including the novelty/significance part immediately at the end of the Introduction section.

At the end of Introduction section, the literature review has been emphasized and further accompanied with research gap, problem formulation, research objectives, and methodology for the purpose to show the criticality of the current research.

In addition, the authors have suitably modified the content in the Abstract section by primarily focusing on showing the substantial novel findings in the form of empirical/numerical values which have been reported in this current work.

Additionally, based upon the fruitful recommendations of knowledgeable expert, few figures have been moved to the supplementary list of figures (supporting materials) as recommended for better comprehensibility.

The word, “S” refers to the Supplementary information, fig.S. implies Supplementary figures, whereas, the table S. implies Supplementary tables.

3

Just a few more specific comments of the first part of the results but the article should be re-examined after the first correction.

Thank you for your constructive insights. The authors have thoroughly refined the article from the scratch level up to a fierce extent as per the fruitful recommendation of the expert referee.

The verbose explanation in the respective sections, and sub-sections, introduction, literature-review, experimentation, concluding remarks and suggestions for future work have been removed.

The authors have avoided the general discussion and furthermore has reported only the comparison of the significant findings unveiled different sections of the revised manuscript as highlighted in the red colour.

Moreover, the junk or unwanted content/text has been avoided by utterly eradicating the repeated phrases/words/sentences throughout the manuscript.

The authors have tried to refine the overall content, structure, formatting, and relevancy of the revised article as per the learnt recommendations.

The entire content in the revised article has been ameliorated up to fervent extent strictly as per the valuable recommendations. The abstract has been written in chronological sequential order as, it covers Introduction, Objective, Methodology, Results, and overall conclusion.

Additionally, the significant outcomes which are evident from the current investigation work has been enumerated in the abstract with percent improvement in the material properties with promising applications. The extensive revisions have been exhibited in the revised version of the manuscript.

Although, the authors have supported their valuable breakthrough findings with the previous existing studies as shown in the revised version of the manuscript with red color throughout the article.

The authors have been excavated or removed the unnecessary matter or data from the article and thus, the manuscript has been refined, prepare and contemplate it in more precise or accurate, removing the unwanted or junk content from the article, and thus, organized in the systematic orderly manner as till-date citing the literature studies, research gaps, problem formulation, objectives with research methodology as illustrated in the revised version of the manuscript with red color.

The point of discussion has been detailed enumerated in more comprehensive and extensive way by thoroughly supporting the present results and discussions with the previous literature studies.

The prominent noteworthy points or findings/outcomes has been thoroughly addressed in separate paragraphs with primarily emphasized on the percentage enhancement in the properties with detailed comparative analysis among the related composite materials by properly citing the past literary sources.

In-addition, the authors have detailed described and mentioned the general trend variation in the properties of the composites with the deep-discussion on the materials behaviour, trend variation, novelty in the properties, applications, processing fabrication techniques, by appropriately citing the relevant literature studies.

The authors hope that this will give an impetus to the overall content, and novelty in the article and thus, enable readers to identify the cutting-edge technology by thoroughly go-through the article.

4

The EVA materials is not characterized. Some details must be added as for example the Molecular weight or the MFI.

The authors are enormously thankful to the learnt referee for their knowledgeable insights. The related issue has now been rectified as per suggestions by incorporating the same in a precise manner in the Materials section as exhibited in the red colour in revised manuscript.

The molecular weight of recycled polyethylene-vinyl acetate (RPEVA) recycled polymer can vary widely depending on the composition of the recycled material and the processing conditions used.

RPEVA is a copolymer that is made up of both polyethylene and vinyl acetate monomers. The molecular weight of the RPEVA polymer chains is influenced by the ratio of these monomers and the polymerization conditions used to synthesize the material.

In the case of RPEVA, the molecular weight can be further impacted by the presence of contaminants, such as other polymers or additives, and by the extent of degradation that the polymer chains have undergone during processing and use. Degradation can cause chain scission, which reduces the molecular weight of the PEVA chains. The presence of contaminants can also influence the molecular weight by changing the monomer ratio in the polymer.

However, a molecular weight for recycled PEVA will depend on the source and processing history of the recycled material.

Thus, as evident from the Gel Permeation Chromatography (GPC) analysis which the supplier has confirmed from where we have purchased the raw material (Recycled EVA polymer) in context with the same, the average molecular weight of RPEVA is 170,000 g/mol.

As far as the Melt flow index (MFI) is concerned, as MFI of polyethylene-vinyl acetate (PEVA) recycled polymer can vary widely depending on the composition of the recycled material and the processing conditions used.

The MFI of a polymer is a measure of its flow properties when it is melted and subjected to a specified weight of force. A higher MFI value indicates that the polymer is more fluid and has lower viscosity, while a lower MFI value indicates that the polymer is more viscous and has higher viscosity.

In the case of recycled PEVA, the MFI can be influenced by a variety of factors, including the presence of contaminants, such as other polymers or additives, and by the extent of degradation that the polymer chains have undergone during processing and use. Degradation can cause chain scission, which increases the MFI by reducing the viscosity of the melted polymer. The presence of contaminants can also impact the MFI by changing the viscosity of the melt.

Since the MFI of recycled PEVA can vary widely depending on the source and processing history of the recycled material. Hence, after contemplating the MFI in a detailed manner for RPEVA with the supplier from where we have purchased the raw material (Recycled EVA polymer), the MFI for RPEVA with a standard composition and processing, the MFI can range from 0.1 to 45 g/10 min (As MFI has been expressed in units of mass per 10 minutes and is a useful indicator of the viscosity of the melted polymer).

5

Line 691 Something is missing?

Thank you for your valuable insinuations. The related valuable query has now been rectified as per the learnt suggestions.

6

Line 707 Unclear nanoparticles?

Thank you for your constructive insights. The related content has now been modified as per the learnt suggestions.

Since the authors have incorporated the nanofillers/nano-additives (Zinc Octadecanoate (Zinc Stearate) and Octadecanoic acid (Stearic acid)) during the fabrication of polymer composites for the purpose to ameliorate the physicomechanical properties, and meanwhile, to impart flexibility.

Thus, the authors have replaced the word nanoparticles with nano-additives/nanofillers.

7

Figure 5 Repetitions and unclear X axis?

Thank you for your constructive insights. The related content has now been modified as per the learnt suggestions. The authors gratefully convey their admiration to the knowledgeable referee for their constructive insights.

As, the figure 5 has illustrated the comparative analysis of the tensile strength of “neat recycled EVA polymer composites", and "Leather shavings/recycled EVA polymer composites”, and the x-axis (Abscissa) has been modified by rectifying the mistake and now, the axis title has been superseded with the
“Fabricated composites”.

As the tensile load (N) against extension (mm) is an important measurement in determining the mechanical properties of leather shaving fiber reinforced polymer composites (LSFRPCs). This measurement gives an insight into the material's ability to withstand stretching forces, which is crucial for applications that involve high stress and strain.

When a tensile load is applied to the LSFRPC, the fibers in the composite material begin to stretch and elongate. As the load continues to increase, the fibers will eventually reach their breaking point, causing the composite material to fail. The amount of load required to cause this failure, as well as the amount of extension that occurs before failure, provides important information about the material's strength, toughness, and other mechanical properties.

The scientific reason for this importance lies in the understanding of the behavior of materials under stress. When a material is subjected to a tensile load, it experiences a state of stress, which is a measure of the force per unit area. This stress causes the material to undergo deformation, which is a change in shape or size. The amount of deformation that occurs is proportional to the amount of stress applied, and this relationship is described by Hooke's Law.

By measuring the tensile load against extension for LSFRPCs, engineers and materials scientists can determine the material's stress-strain behavior, which can then be used to optimize the design and production of the composite material. This information can also be used to determine the suitability of the LSFRPCs for various applications and to ensure that they meet required mechanical performance standards.

In conclusion, the tensile load against extension measurement is critical in evaluating the mechanical properties of LSFRPCs and provides valuable information for optimizing the design and production of these composite materials.

Furthermore, the tensile and yield strength tests are analysed for testing the performance of the developed polymer composites. The behaviour of the materials is computed through the analysis of different mechanical properties of the material once related to various loading circumstances. The capability of a material to substance load of axial without break is computed with the use of a tensile test, which is also defined as tensile strength. The forces of applied continuously to the equal and opposites under each end of the material which is pulled and elongated in addition deduction of diameter occur. To analyse the properties of the materials such as percentage reduction of area, elongation, lower yield strength, upper yield strength, proof strength, Tensile strength, and tensile properties through the utilization of the tensile test.

We can demonstrate the stress strain curve based on the limits from deformation limit through the observation of tensile test. Figure 5 has showed the comparative analysis of tensile strength of neat recycled EVA polymer composites, and Leather shavings/recycled EVA polymer composites; whereas, the figure 6a and figure 6b has displayed load versus displacement curves for the developed polymer composites to thoroughly understand the nature of the fabricated polymer composite samples in terms of stress – strain curve.  Thus, there is no repetition in these figures.

The title of x-axis (Abscissa) has been modified by rectifying the mistake and now, the axis title has been superseded with the
“Fabricated composites”.

8

Line 748 Figure 5? Wrong number? Moreover, problems with the x axis.

Thank you for your learnt observations. Suggestions have been incorporated in the revised version of the article as per the learnt comment. The figure number has been modified as per the chronological sequence order.

As, the figure 5 has illustrated the comparative analysis of the tensile strength of “neat recycled EVA polymer composites", and "Leather shavings/recycled EVA polymer composites”, and the title of x-axis (Abscissa) has been modified by rectifying the mistake and now, the axis title has been superseded with the
“Fabricated composites”.

9

SEM (Figure 6 that should have another number) Indeed It is hard to see what is described in the text surface is lined and filled with "bulged, twisted, and elongated materials"

The authors express their thankfulness to the competent reviewer for their constructive insights.

Based upon the fruitful recommendations of knowledgeable expert, the labelling has been done on the SEM micrographs (Figure 6) to properly exhibit the bulging of leather shaving fibers and twisting of leather shaving fibers.

10

Figure 10 Y axis Newtons or MPa? What is reported on the x-axis?

The authors gratefully convey their admiration to the knowledgeable referee for their constructive insights.

The units of load in the y-axis title in figure 10 is “Newton”, while, the title of x-axis in the same figure 10 is extension or elongation is “mm”. All in all, figure 10 has displayed the compressive load versus displacement curves for the developed polymer composites to thoroughly understand the nature of the fabricated polymer composite samples in terms of stress – strain curve. 

The compressive load and compressive extension properties of a material play a crucial role in determining the material's behavior under stress and strain. These properties are particularly important for leather shaving fiber reinforced recycled polymer composites, which are used in various applications where mechanical performance is a critical factor.

Compressive load refers to the amount of force applied to a material in a compressive manner, perpendicular to its surface. A material's compressive strength is defined as the maximum stress that a material can withstand under compressive load without undergoing significant permanent deformation or failure.

Compressive extension refers to the stretching or elongation of a material under compressive load. The compressive extension behavior of a material is significant because it determines the material's ability to withstand deformations and recover from them.

For leather shaving fiber reinforced recycled polymer composites, both compressive load and compressive extension are important because these materials are often subjected to compressive loads in real-world applications. For example, the materials may be used in automotive or construction applications where they are subjected to compressive loads during use.

The leather shaving fibers present in the composites reinforce the matrix polymer, improving its mechanical properties such as compressive strength and compressive extension behavior. The fibers distribute the compressive loads over a larger area, reducing the stress concentration and improving the composite's overall mechanical performance.

In conclusion, the compressive load and compressive extension properties of leather shaving fiber reinforced recycled polymer composites are important because they determine the material's behavior under stress and strain. The presence of the reinforcing fibers improves these properties, making the composites suitable for various applications where mechanical performance is a critical factor.

11

But the text (as well as the Figures) must be changed in all the paragraphs.

The authors gratefully convey their admiration to the knowledgeable referee for their constructive insights.

The axis titles (both x-axis and y-axis) in each and every respective figures have thoroughly been modified as per the suggestions of the learnt referees.

Additionally, the information in accordance with citations of the figures in the content/text has extensively been incorporated with proper physics behind the mechanism as well as by making comparative analysis on the present results with the previous existing outcomes to make the content throughout the overall manuscript a more appealing, worthwhile and attractive.

Hence, a scientific explanation of the obtained results has been refined and ameliorated up to a fervent extent. Results are enumerated, test methods are utterly described, interpretation have been corelated with results and previous literature findings. The overall summary should indicate the progress of the research and the limitations. 

Note: All the necessary changes/added sentence has been shown in the RED COLOR.

Thank you very much in advance for taking your time in reviewing this manuscript.

Sincerely, we hope you will find our revision satisfactory.

Thanks, in anticipation.

Reviewer 3 Report

Dear responsible author

Thank you for your efforts

The article is well written and well researched, but some points and uncertainties remain:

- The volume of the text and contents of the article is large, it is necessary to reduce the number of pages by a quarter and summarize.

-Why are standards from different countries used in table 2? Should you use a country standard or a special international standard?

-Why in Fig. 5(b). Is the graph drawn as a broken line? And it is not drawn as a curve? be corrected.

-What is the horizontal axis of Figures 9 and 5? To be modified.

-The conclusion section is very extensive writing and repetitive explanations of the text. To be summarized, maybe 500 to 700 words will be enough.

-In the diagram Fig.21(b). SEM-EDX analysis in line 1525, the amount of zinc element is repeated twice, to check the accuracy of the graph.

-The title of the article is too long. be shorter and more specialized.

-If the waste has lost its original properties over time (aging phenomenon), how did you consider this in this research?

Very Thanks.

Author Response

09.02.2023

Dear Prof. (Dr.) Editor-in-chief,

Thank you for considering the manuscript entitled, “Fabrication of novel polymer composites from leather waste fibers and recycled Poly (Ethylene-vinyl-acetate) polymer: Re-cycling of footwear and plastic wastes for application in value-added products”, for the publication in SUSTAINABILITY (MDPI). I am grateful to you and the reviewers for the valuable suggestions provided. I like to resubmit our revised version of the manuscript by adding response to all your comments. Below please find the answers and actions taken to address these comments. All the suggestions are incorporated and highlighted with the RED COLOR in manuscript.

NOTE: All the necessary changes/added sentence has been shown in the RED COLOR.

The locations of these changes have been mentioned, where possible, in the action points that respond to each reviewers’ comments. Here are the responses to the reviewer comments:

AUTHOR RESPONSE TO REVIEWER AND EDITOR COMMENTS

Manuscript Number.: sustainability-2135688

Title: Fabrication of novel polymer composites from leather waste fibers and recycled Poly (Ethylene-vinyl-acetate) polymer: Re-cycling of footwear and plastic wastes for application in value-added products

Journal: Sustainability (MDPI)

The manuscript has been thoroughly modified and improved the quality of the content to meet the standards of the Journal. All the suggestions made by the learned referees are included in the revised manuscript. We are extremely thankful to the referees & editor(s) for their constructive comments and appreciation.

Response to Reviewer’s Comments

The authors are grateful to the reviewers for their suggestions that have all contributed to improving the manuscript. Once again, the authors are extremely thankful for the observations and the comments of the reviewers. All the comments are appropriately addressed and now the quality of the article has been appreciably enhanced before the consideration for publications. The rebuttal file is enclosed indicating the revisions incorporated in the article as suggested. The revisions are carried out in RED COLOR in the text of the manuscript for better visibility to the reviewers and as well as to the editor. We have made the modifications as per their suggestions in the revised manuscript and changes are also marked up using the RED COLOR function.

All in all, the authors should thank the reviewers for their meticulous observations in reviewing the article. All the issues raised by the authors are appropriately addressed as stated in the following table,

Response to Reviewers comments

S. No.

COMMENTS

Action taken

REVIEWER -3 COMMENTS

Dear responsible author

Thank you for your efforts

The article is well written and well researched, but some points and uncertainties remain:

1

- The volume of the text and contents of the article is large; it is necessary to reduce the number of pages by a quarter and summarize.

The authors are enormously thankful to the learnt referee for their knowledgeable insights.

Suggestions has been thoroughly incorporated in revised version of the article as per the learnt comment.

The verbose explanation in the respective sections, and sub-sections, introduction, literature-review, experimentation, concluding remarks and suggestions for future work have been removed.

The authors have avoided the general discussion and furthermore has reported only the comparison of the significant findings unveiled different sections of the revised manuscript as highlighted in the red colour.

The authors would like to mention that the few undesirable sentences, figures, tables, and supplementary figures or tables from the revised manuscript have been omitted to a fervent extent. Moreover, the junk or unwanted content/text has been avoided by utterly eradicating the repeated phrases/words/sentences throughout the manuscript.

The authors have tried to refine the overall content, structure, formatting, and relevancy of the revised article as per the learnt recommendations.

The entire content in the revised article has been ameliorated up to fervent extent strictly as per the valuable recommendations. The abstract has been written in chronological sequential order as, it covers Introduction, Objective, Methodology, Results, and overall conclusion.

Additionally, the significant outcomes which are evident from the current investigation work has been enumerated in the abstract with percent improvement in the material properties with promising applications. The extensive revisions have been exhibited in the revised version of the manuscript.

Although, the authors have supported their valuable breakthrough findings with the previous existing studies as shown in the revised version of the manuscript with red color throughout the article.

The authors have been excavated or removed the unnecessary matter or data from the article and thus, the manuscript has been refined, prepare and contemplate it in more precise or accurate, removing the unwanted or junk content from the article, and thus, organized in the systematic orderly manner as till-date citing the literature studies, research gaps, problem formulation, objectives with research methodology as illustrated in the revised version of the manuscript with red color.

The point of discussion has been detailed enumerated in more comprehensive and extensive way by thoroughly supporting the present results and discussions with the previous literature studies.

The prominent noteworthy points or findings/outcomes has been thoroughly addressed in separate paragraphs with primarily emphasized on the percentage enhancement in the properties with detailed comparative analysis among the related composite materials by properly citing the past literary sources.

In-addition, the authors have detailed described and mentioned the general trend variation in the properties of the composites with the deep-discussion on the materials behaviour, trend variation, novelty in the properties, applications, processing fabrication techniques, by appropriately citing the relevant literature studies.

The authors hope that this will give an impetus to the overall content, and novelty in the article and thus, enable readers to identify the cutting-edge technology by thoroughly go-through the article.

2

-Why are standards from different countries used in table 2? Should you use a country standard or a special international standard?

Thank you for your valuable insinuations. The related issue has now been rectified as per suggestions.

As this article has been focused on the utilization of leather fibers (leather shavings) reinforced recycled polymer composites for multifaceted applications, thus, in context with the testing standards for the performance of leather-fiber composite materials are provided by SATRA testing standards, and some of the characterizations or testing has been performed as per the ASTM/ISO testing methods.

As the SATRA (The Shoe and Allied Trade Research Association) is an internationally recognized testing and research organization that provides a range of services to support the footwear, leather and textile industries. SATRA's focus on leather-fiber composite materials testing standards is due to the importance of these materials in the footwear and leather industries.

Leather-fiber composite materials are used in a variety of applications, including footwear, clothing, and upholstery, and their performance is critical to the overall quality of the end product. For example, the durability and strength of the composite material can impact the wear and tear of the product, while its “breathability” can affect the comfort of the wearer.

SATRA uses a range of scientific methods and protocols to test the performance of leather-fiber composite materials. These tests are designed to assess the materials' “physical”, and “mechanical properties”, such as “tensile strength”, “tear resistance”, “abrasion resistance”, and “flexural properties”. The tests also evaluate the materials' “thermal”, and “dimensional stability”, and “resistance to moisture”, “chemicals”, and “UV light”.

By using standardized testing methods, SATRA can provide a level of consistency and reliability in the evaluation of leather-fiber composite materials. This helps to ensure that the materials meet specific performance requirements and that products made from these materials are of high quality.

In conclusion, SATRA's standards for the performance of leather-fiber composite materials are based on a combination of scientific research and practical experience. The tests provide a reliable and consistent way to assess the materials' performance, ensuring that products made from these materials are of high quality and meet the needs of the footwear and leather industries.

With all due respect to the reviewer suggestions, I have shared with you the following referred articles with the similar related testing standards for your kind reference and consideration,

Referred articles:

[1].      T.J. Madera-Santana, M. Aguilar-Vega, A. Marquez, F. Moreno, M. Richardson, J. Machin, Production of leather-like composites using short leather fibers, II. Mechanical characterization, Polymer Composites 23 (2004).

[2].      M. Parisi, A. Nanni, M. Colonna, Recycling of Chrome-Tanned Leather and Its Utilization as Polymeric Materials and in Polymer-Based Composites: A Review, Polymers 13 (3) (2021) 429–429. doi:10.3390/polym13030429.

[3].      Pan S, Zhao M, Andrawes B, Zhao H, Li L. Compressive behavior of cylindrical rubber buffer confined with fiber reinforced polymer. Journal of Low Frequency Noise, Vibration and Active Control. 2020;39(3):470-484. doi:10.1177/1461348418783570.

[4].      N. Natchimuthu, G. Radhakrishnan, K. Palanivel, K. Ramamurthy, J. S. Anand, Vulcanization characteristics and mechanical properties of nitrile rubber filled with short leather fibres, Polymer International 33 (3) (1994) 329–333. doi:10.1002/pi.1994.210330313.

[5].      R. D. Kale, N. C. Jadhav, Utilization of waste leather for the fabrication of composites and to study its mechanical and thermal properties, SN Applied Sciences 1 (10) (2019) 1–9. doi:10.1007/s42452-019- 1230-9.

[6].      J. D. Ambrósio, A. A. Lucas, H. Otaguro, L. C. Costa, Preparation and characterization of poly (vinyl butyral)-leather fiber composites, Polymer Composites 32 (5) (2011) 776–785. doi:10.1002/pc.21099.

[7].      E.T. Musa, A. Hamza, A.S. Ahmed, Investigation of the Mechanical and Morphological Properties of High-Density Polyethylene (HDPE)/Leather Waste Composites, IOSR Journal of Applied Chemistry 10 (01) (2017) 48–58. doi:10.9790/5736-1001014858.

[8].      K. Ravichandran, N. Natchimuthu, Vulcanization characteristics and mechanical properties of natural rubber-scrap rubber compositions filled with leather particles, Polymer International 54 (3) (2005) 553– 559. doi:10.1002/pi.1725.

[9].      K. Ravichandran, N. Natchimuthu, Natural rubber: leather composites, Polímeros 15 (2) (2005) 102– 108. doi:10.1590/s0104-14282005000200008.

[10].    S. Joseph, T. S. Ambone, A. V. Salvekar, S. N. Jaisankar, P.  Saravanan, E. Deenadayalan, Processing and characterization of waste leather based polycaprolactone biocomposites, Polymer Composites 38 (12) (2017) 2889–2897. doi:10.1002/pc.23891.

[11].    K. Ravichandran, N. Natchimuthu, Natural rubber: leather composites, Polímeros 15 (2) (2005) 102– 108. doi:10.1590/s0104-14282005000200008.

[12].    S. Joseph, T. S. Ambone, A. V. Salvekar, S. N. Jaisankar, P.  Saravanan, E. Deenadayalan, Processing and characterization of waste leather based polycaprolactone biocomposites, Polymer Composites 38 (12) (2017) 2889–2897. doi:10.1002/pc.23891.

[13].    Hang, L.T., Viet, D.Q., Linh, N.P.D., Doan, V.A., Dang, H.L.T., Dao, V.D. and Tuan, P.A., 2021. Utilization of leather waste fibers in polymer matrix composites based on Acrylonitrile-Butadiene rubber. Polymers, 13(1), p.117.

[14].    Ramaraj, B. 2006, Mechanical and thermal properties of ABS and leather waste composites. J. Appl. Polym. Sci., 101: 3062-3066. https://doi.org/10.1002/app.24113.

3

-Why in Fig. 5(b). Is the graph drawn as a broken line? And it is not drawn as a curve? be corrected.

Thank you for your constructive insights. Based upon the fruitful recommendations of knowledgeable expert, the visibility, scale, annotations, coordinate text/formatting, and ruler number of all the figures have been enhanced to fervent extent as recommended for the better comprehensibility.

The graph has not been drawn as a broken line.

As per the learnt suggestions, the quality of the figures have been ameliorated up to fervent extent. The resolutions and pixels of all the figures have been enhanced stringently as per the prestigious, SUSTAINABILITY MDPI Journal standard in a proper tiff. format. The formatting and quality of the figures have been strengthened as per the standard Journal template.

4

-What is the horizontal axis of Figures 9 and 5? To be modified.

In appreciation of the knowledgeable comments of the learned referee, the authors extend their gratitude. The horizontal axis for the respective figures 5 and 9 has now been modified.

5

-The conclusion section is very extensive writing and repetitive explanations of the text. To be summarized, maybe 500 to 700 words will be enough.

Thank you for your valuable insinuations. The concluding remarks section has now been reduced up to a fervent extent, while, the vague or junk or unwanted/verbose content has been eradicated as per valuable suggestions.

The authors would like to mention that the few undesirable sentences from the Conclusion section have been omitted to a fervent extent. Moreover, the junk or unwanted content/text has been avoided by utterly eradicating the repeated phrases/words/sentences throughout the manuscript.

The revisions have been shown on revised manuscript in the red color.

For your kind consideration and perusal, I am enumerating the response to the valuable comment as mentioned below,

In this study, recycled EVA polymer matrices and the high weight content of leather shavings with 1:1 were mixed to fabricate the flexible composite sheets through the combined effects of two-roll milling and hot-press compression moulding with uniform blending. Observations based on thorough, detailed characterisation investigations includes the following:

i.          The physico-mechanical properties of “leather shavings/ recycled EVA” composites exhibited to be significantly influenced by the leather-fibrous loading with a composition of 1:1. The tensile strength tends to slightly rise when the proportion of leather shavings in composites has been in-creased to 1:1. As the volume of leather fibers within the composites has increased, the modulus of elasticity of the composites has significantly improved. Leather shavings were discovered to have a stronger compressional deformation property in the recycled EVA matrix than the ‘neat-recycled EVA’ matrix, which was reported to be higher by about 7.7%. The average peel strength between the polymer and leather (for leather shavings as fiber) in the recycled EVA matrix was determined to be approximately 0.9575 N/mm, according to the results obtained.

ii.          The TGA investigation showed that ‘recycled-EVA’ polymer was thermo-stable up to 213.47°C whereas leather fibers showed no discernible major weight loss nearly comparable to 211°C. According to the DSC results, the ‘release of moisture’ from the ‘leather shavings’ through an endothermic transition that occurs at about 100°C, are thermostable up to 211°C, and begin to decompose collagen at 332.56°C for ‘neat-recycled EVA’ samples and 327.23°C for “leather shavings/recycled EVA” polymer composite samples, respectively.

iii.          In reference to the absorbance bands O-H (3100-2914.88 cm-1) that in-creased in intensity under humid conditions due to hydrogen bonding interactions between the carbonyl group in vinyl acetate and water, the ATR-FTIR spectra have explored surface-interface-layer alterations and abnormalities in recycled EVA co-polymer. While the molecular structure of “leather shavings/recycled EVA” composites can be seen in the ATR-FTIR monograph as the band at 3314.07 cm-1 corresponding to the vinyl alcohol -OH group; 2849.31-2916.81 cm-1 is due to the -CH2, -CH3 groups inside chains and terminal groups; and below 723.175 to 600 cm-1 is due to Cr-O bonds.

iv.          Leather fibrils particles are widely present as conglomerate aggregate clusters and are effectively distributed or interspersed throughout the re-cycled EVA matrices, according to the SEM surface morphology examination of the “leather shavings/recycled EVA” composites. The micrograph results show numerous interfaces with remarkable bonding strength and interfacial contact between the recycled EVA matrix and the leather shavings residual particles.

v.          According to “XRD-analysis”, the “crystallinity value” of the ‘neat-recycled EVA’ polymeric composite sample is 7%, and the amorphous content is revealed to be 93%. While the crystallinity of the leather-shavings/recycled EVA polymeric composite sample was revealed to be 19.3% and the amorphous content to be 80.7 percent. This unexpected result has also demonstrated that the crystallinity value of the leather-filled recycled EVA composites is higher than that of the neat recycled EVA composite samples, which eventually confirms that the leather-filled re-cycled EVA composites have higher values for strength, modulus, and hardness.

vi.          The inclusion of additives and lubricants into the recycled EVA polymeric matrix has revealed a relatively uniformly smooth-surface with fewer void-spaces in the material but a much more prominent peak has been observed.

To recapitulate, it is necessary to conclude that these leather shavings waste/recycled EVA polymer composites with lower cost can be employed for multi-purpose applications as well as in the reduction of environmental pollution.

6

-In the diagram, Fig.21(b). SEM-EDX analysis in line 1525, the amount of zinc element is repeated twice, to check the accuracy of the graph.

Thank you for your constructive comments. After careful consideration, and contemplating your valuable query in a detailed manner, the zinc element which is reflecting twice in the SEM-EDX spectra analysis (Fig.21(b)) has now been eradicated as illustrated in the revised manuscript.

7

-The title of the article is too long. be shorter and more specialized.

The title of the article has been shortened as per the learnt suggestions, and I hope that the modified article title is now looking more appealing, captivating, descriptive, concise and precise.

8

-If the waste has lost its original properties over time (aging phenomenon), how did you consider this in this research?

Very Thanks.

Thank you for your fruitful insinuations. The authors would really like to discuss regarding the same with the concrete evidences for your thoughtful perspective as indicated below for your kind consideration and perusal,

“Leather” is a “natural-fabric” that is obtained by “tanning animal skin”, or “hides”. The main constituent of leather is collagen, which is a protein that forms the structure of skin and hide. Collagen is a protein composed of “triple-units” of “peptide chains” that are joined by “hydrogen bonds”. During the leather-making process, chromium III is widely used as a tanning agent as it provides several properties to the leather, such as improved physical and mechanical properties, interface-adhesion, thermal resistance, and softness, making it suitable for various industrial applications.

The tanning process makes the leather imputrescible and increases its chemical and physical durability. After “tanning”, leather is composed of “proteins”, “water”, and other “mineral matter”. Scientists are experimenting with other chemical processes to ameliorate the “thermostability of leather”, especially, “leather-waste” from “industrial operations”. Literature findings have evident that Polymer composites filled with leather waste have been reported to have several uses, including “construction materials”, “automobile interior moldings”, “heat”, and “sound insulating boards”, “shoe soles”, and “flooring materials”.

All in all, leather is composed of several layers, including the grain, corium, and flesh layers. The "corium layer", which is comprised of "long fibrous waves" of "collagen fibrils" within the "fiber network layer", is responsible for the substantial contribution to the "leather's thickness", and "strength". The “complex layered-structure” of leather has been studied and reported in several works, and it is known that the “leather” is “highly compatible” with “polar polymer-matrices” due to the presence of “amine”, and “carboxyl groups” in its “collagen structure”. The “natural crosslinking” formed by the “interaction of these groups” through “hydrogen bonds” also contributed to a pivotal part in ameliorating the “mechanical strength” of the fibers.

“Collagen” is a key fibrous protein that contributes to the formation of “skin”, and “hide”. It has a unique structure, with a “triple helix” made up of “polypeptide chains” containing different “amino acids”. This protein can form “hierarchical structures”, such as “microfibrils”, “fibrils”, “fibers”, and “fiber bundles”, when “multiple collagen molecules” aggregate together. These structures, along with other “non-collagenous materials” like “hair”, and “flesh”, eventually combine to form “skin”, and “hide”. The composition of “skin”, and “hide” makes it susceptible to “physical”, “chemical”, and “biological treatments”, which are valuable and utilized in the process of “leather manufacturing”.

Also, EVA is used to make 'inserts/soles and midsoles' throughout the industrial footwear sector, accounting for 18% to 28% of the total. Prior to adding a mixture of Natural and Butadiene rubbers possess better mechanical characteristics like “high capabilities, minimum slippage, shrinkage, slipping, and delivers a ferocious compression-set”, EVA has “low compression-set, abrasion, and tearing characteristics”. These materials have exceptional 'physical and compression' characteristics together with 'least slip-page', when compared to EVA/BR and EVA/NR. In the production of “hiking boots, basketball shoes, and virtually every other type of athletic footwear”, the EVA midsole is an excellent 'vibration-damping' based cushioning material, which is 'light in weight, resists compression set, easily available in any colour, and is made conveniently'.

Furthermore, the Additives employed during the processing of the fabricated leather shaving fiber loaded recycled EVA polymer composites are Zinc Octadecanoate (Zinc Stearate) and Octadecanoic acid (Stearic acid) which have been served as plasticizers while, the lubricants/oils used to process the fabricated polymer composites are paraffins and naphthalene's which resulted in the enhancement of the properties despite the aging phenomenon.

The similar comparable outcomes from the previous literature review in accordance with the utilization of leather fiber loaded polymer composites for value addition applications have been reported with the detailed discussion on the findings in the revised manuscript. With all due respect to the reviewer suggestions, I have shared with you the following referred articles with the similar related findings for your kind reference and consideration,

Referred articles:

[1].      T.J. Madera-Santana, M. Aguilar-Vega, A. Marquez, F. Moreno, M. Richardson, J. Machin, Production of leather-like composites using short leather fibers, II. Mechanical characterization, Polymer Composites 23 (2004).

[2].      M. Parisi, A. Nanni, M. Colonna, Recycling of Chrome-Tanned Leather and Its Utilization as Polymeric Materials and in Polymer-Based Composites: A Review, Polymers 13 (3) (2021) 429–429. doi:10.3390/polym13030429.

[3].      Pan S, Zhao M, Andrawes B, Zhao H, Li L. Compressive behavior of cylindrical rubber buffer confined with fiber reinforced polymer. Journal of Low Frequency Noise, Vibration and Active Control. 2020;39(3):470-484. doi:10.1177/1461348418783570.

[4].      N. Natchimuthu, G. Radhakrishnan, K. Palanivel, K. Ramamurthy, J. S. Anand, Vulcanization characteristics and mechanical properties of nitrile rubber filled with short leather fibres, Polymer International 33 (3) (1994) 329–333. doi:10.1002/pi.1994.210330313.

[5].      R. D. Kale, N. C. Jadhav, Utilization of waste leather for the fabrication of composites and to study its mechanical and thermal properties, SN Applied Sciences 1 (10) (2019) 1–9. doi:10.1007/s42452-019- 1230-9.

[6].      J. D. Ambrósio, A. A. Lucas, H. Otaguro, L. C. Costa, Preparation and characterization of poly (vinyl butyral)-leather fiber composites, Polymer Composites 32 (5) (2011) 776–785. doi:10.1002/pc.21099.

[7].      E.T. Musa, A. Hamza, A.S. Ahmed, Investigation of the Mechanical and Morphological Properties of High-Density Polyethylene (HDPE)/Leather Waste Composites, IOSR Journal of Applied Chemistry 10 (01) (2017) 48–58. doi:10.9790/5736-1001014858.

[8].      K. Ravichandran, N. Natchimuthu, Vulcanization characteristics and mechanical properties of natural rubber-scrap rubber compositions filled with leather particles, Polymer International 54 (3) (2005) 553– 559. doi:10.1002/pi.1725.

[9].      K. Ravichandran, N. Natchimuthu, Natural rubber: leather composites, Polímeros 15 (2) (2005) 102– 108. doi:10.1590/s0104-14282005000200008.

[10].    S. Joseph, T. S. Ambone, A. V. Salvekar, S. N. Jaisankar, P.  Saravanan, E. Deenadayalan, Processing and characterization of waste leather based polycaprolactone biocomposites, Polymer Composites 38 (12) (2017) 2889–2897. doi:10.1002/pc.23891.

[11].    K. Ravichandran, N. Natchimuthu, Natural rubber: leather composites, Polímeros 15 (2) (2005) 102– 108. doi:10.1590/s0104-14282005000200008.

[12].    S. Joseph, T. S. Ambone, A. V. Salvekar, S. N. Jaisankar, P.  Saravanan, E. Deenadayalan, Processing and characterization of waste leather based polycaprolactone biocomposites, Polymer Composites 38 (12) (2017) 2889–2897. doi:10.1002/pc.23891.

[13].    Hang, L.T., Viet, D.Q., Linh, N.P.D., Doan, V.A., Dang, H.L.T., Dao, V.D. and Tuan, P.A., 2021. Utilization of leather waste fibers in polymer matrix composites based on Acrylonitrile-Butadiene rubber. Polymers, 13(1), p.117.

[14].    Ramaraj, B. 2006, Mechanical and thermal properties of ABS and leather waste composites. J. Appl. Polym. Sci., 101: 3062-3066. https://doi.org/10.1002/app.24113

Hence, a scientific explanation of the obtained results has been refined and ameliorated up to a fervent extent. Results are enumerated, test methods are utterly described, interpretation have been corelated with results and previous literature findings. The overall summary should indicate the progress of the research and the limitations. 

Note: All the necessary changes/added sentence has been shown in the RED COLOR.

Thank you very much in advance for taking your time in reviewing this manuscript.

Sincerely, we hope you will find our revision satisfactory.

Round 2

Reviewer 1 Report

ok

Reviewer 2 Report

I do not understand the continuous use in the text of "". Can the authors eliminate this?

The English has been improved but still not very clear and the text still contains repetition but thr paper can be accepted

Figure

Reviewer 3 Report

Dear authors, thank you for your efforts

Thank you for revision the article.

best regards